# Solving Neural Min-Max Games:
# The Role of Architecture, Initialization & Dynamics

**Deep Patel and Emmanouil-Vasileios Vlatakis-Gkaragkounis**
Department of Computer Science
University of Wisconsin-Madison
{dbpatel5,vlatakis}@wisc.edu

## Abstract

Many emerging applications—such as adversarial training, AI alignment, and robust optimization—can be framed as zero-sum games between neural nets, with von Neumann–Nash equilibria (NE) capturing the desirable system behavior. While such games often involve non-convex non-concave objectives, empirical evidence shows that simple gradient methods frequently converge, suggesting a hidden geometric structure. In this paper, we provide a theoretical framework that explains this phenomenon through the lens of *hidden convexity* and *overparameterization*. We identify sufficient conditions—spanning initialization, training dynamics, and network width—that guarantee global convergence to a NE in a broad class of non-convex min-max games. To our knowledge, this is the first such result for games that involve two-layer neural networks. Technically, our approach is twofold: (a) we derive a novel path-length bound for the alternating gradient descent–ascent scheme in min-max games; and (b) we show that the reduction from a hidden convex–concave geometry to two-sided Polyak–Łojasiewicz (PL) min-max condition hold with high probability under overparameterization, using tools from random matrix theory.

## 1 Introduction

At the Nobel Symposium marking the centennial of Game Theory [31, 83], a key challenge was posed:

*the development of a systematic theory for non-convex games*

spurred by the rapid growth of deep learning in incentive-aware multi-agent systems [104, 130].

Indeed, many influential modern AI systems are built upon the fusion of foundational game-theoretic principles—particularly zero-sum games—with the expressive capacity of neural networks. Notable examples include generative adversarial networks (GANs) [51], robust reinforcement learning [89], adversarial attacks [117], domain-invariant representation learning [44], distributionally robust optimization[77, 123], and multi-agent environments featuring natural language interactions, such as AI safety debates between large language models and verifier-prover systems [56, 16]. In these settings, the game-theoretic framework provides a natural and interpretable objective—typically an equilibrium solution endowed with strong normative appeal, such as the celebrated von Neumann minimax points [48] and Nash–Rosen equilibria [80, 94].

At the same time, much of the remarkable progress at the intersection of deep learning and game theory stems from the capacity of deep models to operate effectively in environments with large, often continuous, state and action spaces. Iconic examples include Go [103], autonomous driving [102], Texas Hold'em poker [15], and real-time strategy games such as StarCraft II through AlphaStar [112].

39th Conference on Neural Information Processing Systems (NeurIPS 2025).

Tackling such large-scale decision-making problems has necessitated the combination of expressive architectures with function-approximation-based learning, replacing high-dimensional reward/value functions and strategy/policy spaces with trainable surrogates. Hence, these surrogates act as flexible intermediaries, enabling generalization across complex environments without exhaustive enumeration of action spaces.While theoretical focus has largely remained on linear approximators [121, 26], it is the nonlinear models—such as kernels and deep neural networks—which in practice dramatically expand representational power [67, 58], allowing richer strategic behaviors. Thus, agents' policies are encoded through powerful approximators, and equilibrium learning unfolds through iterative parameter tuning (see Figure 1).

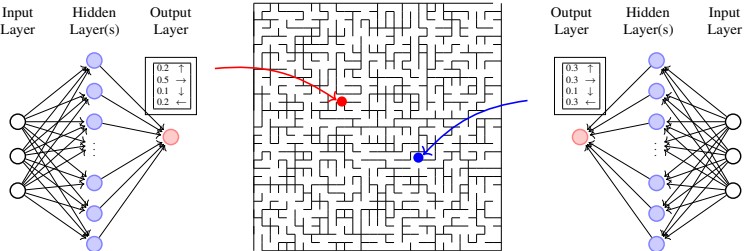

Figure 1: *Illustration of a maze environment where each agent must reason over a vast space of action sequences. Instead of explicitly constructing and searching the full decision tree, a neural network implicitly encodes both the value of paths and the policy for navigation, learning an effective strategy dynamically without ever uncovering the complete structure of the maze.*

Despite the empirical success, algorithms with provable convergence guarantees remain scarce. This is unsurprising given that even in finite games, strong computational hardness results [22, 23, 32, 87] and dynamic impossibility theorems [78, 53, 52, 116, 46, 47] pose significant barriers. Notably, even in two-player zero-sum games—where classical theory guarantees existence and efficient computation of minimax points via LP duality [19] or optimistic first-order methods [120, 5]—these assurances collapse when modern deep-learning architectures, with their inherent non-convexity, are introduced [34, 10, 6]. Specifically:$(i)$ global solution concepts (e.g., von Neumann minimax, Nash equilibria) may fail to exist; $(ii)$ even when they do, tandem gradient-based methods often suffer from *instability*, *cycling*, or *divergence*, resulting in poor solutions. [33, 75, 35, 113].

Thus, the best hope for mitigating the practical impact of these worst-case hardness results lies in focusing on structured subclasses of games. It remains plausible that broad families of non-concave games—rich enough to capture multi-agent interactions—admit tractable local or even global equilibria.

**Hidden convexity: a promising direction.** One compelling approach along this path is the emerging theory of *hidden convex games* [114, 79, 115, 99, 29]. In its simplest form, two players interact via a convex–concave zero-sum game $Loss(\text{Player}_1, \text{Player}_2)$, but control only high-dimensional parameters $\theta, \phi$, through mappings $\text{Player}_1 \leftarrow Map_\theta^1(\cdot)$ and $\text{Player}_2 \leftarrow Map_\phi^2(\cdot)$. These mappings are smooth and known, allowing gradient-based training, but typically not efficiently invertible, reflecting the practical irreversibility of neural architectures. Consequently, while the latent game preserves convex–concave structure, the optimization landscape $Loss(Map_\theta^1, Map_\phi^2)$ over control variables becomes highly non-convex [see 99, p. 26] . Although not every non-convex game admits such a structure, many practical applications naturally fit within this framework (see Appendix B).
**Rank collapse: the fragility of hidden convexity.** A major criticism of the hidden convexity paradigm relies critically on the assumption that the Jacobian of the agents' mappings maintain uniformly bounded singular values throughout training. In practice, such uniform bounds often fail, as real-world architectures may suffer from rank collapse or near-singular behavior during optimization (see, e.g., [101, 43, 37]), undermining theoretical guarantees. When such degeneracies arise, convergence rates can deteriorate exponentially, and worst-case bounds may become vacuous. Even if Jacobian well-conditioning is achieved by a random initialization, there are no assurances that it will be preserved as training evolves.

These limitations underscore the need for explicit, open-box conditions—beyond abstract hidden mappings—that explain the empirical success of efficient training in large-scale min-max settings. Whilst hidden convexity provides significant insights about these systems, it does not answer a fundamental behavioral question:

> *Can appropriate architectural design, initialization protocols, and training dynamics jointly ensure efficient convergence in large-scale neural min-max games?* ($\star$)

## 1.1 Setting and Main Contribution

Motivated by the above challenges, we provide— to the best of our knowledge—the first quantitative convergence guarantees addressing the central question ($\bigstar$) under minimal assumptions. Formally, given input datasets $\mathcal{D}_F$ and $\mathcal{D}_G$, and latent strategy spaces $\mathcal{S}_F$ and $\mathcal{S}_G$, we consider the hidden min-max problem

$$\min_{\theta \in \mathbb{R}^{d_\theta^{(F)}}} \max_{\phi \in \mathbb{R}^{d_\phi^{(G)}}} \mathcal{L}_{\mathcal{D}}(F_\theta, G_\phi), \tag{$\prod$}$$

where $F_\theta : \mathbb{R}^{d_0^{(F)}} \to \mathbb{R}^{\dim(\mathcal{S}_F)}$ and $G_\phi : \mathbb{R}^{d_0^{(G)}} \to \mathbb{R}^{\dim(\mathcal{S}_G)}$ are smooth mappings parameterized by $\theta$ and $\phi$ (e.g., neural network weights). While our results extend beyond, we focus on well-studied [127] *separable* latent minmax objectives of the form

$$\mathcal{L}_{\mathcal{D}}(F, G) = I_1^{\mathcal{D}_F}(F) + I_2^{\mathcal{D}}(F, G) - I_3^{\mathcal{D}_G}(G), \tag{1}$$

where $\mathcal{D} = (\mathcal{D}_F, \mathcal{D}_G)$ and $I_1^{\mathcal{D}_F}, I_3^{\mathcal{D}_G}$—the *individual components*—are strongly convex and smooth, and $I_2^{\mathcal{D}}$—the *coupling component*— is smooth bilinear. As convergence metric, we adopt the *Nash gap* (also known as the *Nikaido–Isoda duality gap* [82]):

$$\mathrm{DG}_{\mathcal{L}_{\mathcal{D}}}(\theta, \phi) := \max_{\phi'} \mathcal{L}_{\mathcal{D}}(F_\theta, G_{\phi'}) - \min_{\theta'} \mathcal{L}_{\mathcal{D}}(F_{\theta'}, G_\phi),$$

and say that $(\hat{\theta}, \hat{\phi})$ is an $\epsilon$-saddle (or $\epsilon$-approximate minimax or Nash equilibrium) if $\mathrm{DG}_{\mathcal{L}_{\mathcal{D}}}(\hat{\theta}, \hat{\phi}) \leq \epsilon$.

**Remarks.** Replacing players' actions with neural nets—i.e., $F_\theta = \mathrm{NN}_\theta(\cdot)$ and $G_\phi = \mathrm{NN}_\phi(\cdot)$— renders the end-to-end landscape highly non-convex, although the latent game $\mathcal{L}$ remains convex–concave. The separable structure naturally unifies several hidden zero-sum regimes: when $I_2$ vanishes, it recovers *separable strongly-convex–concave* games; when $I_1$ and $I_3$ vanish, it reduces to *bilinear games* [114]; and when both components are present, it captures *regularized games* (e.g., Tikhonov- or entropy-regularized settings), recently used in hidden min-max frameworks, including team and zero-sum Markov games [59, 60]. We discuss concrete examples in Section 2 and Appendix B. In these settings, regularization plays a critical role in stabilizing dynamics and mitigating chaotic behaviors, both empirically ([see 99, p. 26]) and theoretically (cf. [115, pp. 7–8], [59]). Before enumerating our techical contributions, we highlight a key result addressing ($\bigstar$):

**Informal Theorem** (Theorem 3.8). *There exists a decentralized, gradient-based method (eq. (Alt-GDA)) that computes, with high probability under suitable Gaussian random initialization, an $\epsilon$-approximate Nash equilibrium for any $\epsilon > 0$ in broad class of hidden convex-concave zero-sum games, where each player's strategy is parameterized by a sufficiently wide two-layer neural network.*

- *The number of iterations required scales as*

$$O\left(\mathrm{poly}\left(\frac{1}{width_1}, \frac{1}{width_2}, \frac{1}{n}, d_{input}\right) \times \frac{L^3}{\mu^3} \times \log\left(\frac{1}{\epsilon}\right)\right),$$

  *where $width_1, width_2$ are the hidden layer widths, $n$ is the number of training samples, $d_{input}$ is the input dimension, $L$ is the smoothness constant, and $\mu$ is the strong convexity modulus of the latent objective.*

- *This guarantee holds provided the network $width_{1,2} = \tilde{\Omega}\left(\mu^2 \frac{n^3}{d_{input}}\right)$.*

**A converse byproduct: input-optimization games.** We also uncover a new convergence guarantee in a related but distinct setting: *optimizing directly over inputs when the neural network mappings are fixed.* This perspective is motivated both by adversarial example generation through min-max formulations (see Section 2, Appendix B & [117]) and by empirical results of [99] for solving normal form zero-sum games using input-optimization at random fixed neural network mappings—without theoretical justification of non-singularity of spectrum trajectory. Formally, the goal is to find input vectors $(x_{\mathrm{Alice}}, x_{\mathrm{Bob}})$ that implement a Nash equilibrium:

$$\min_{x_{\mathrm{Alice}} \in \mathcal{D}_F} \max_{x_{\mathrm{Bob}} \in \mathcal{D}_G} \mathcal{L}\left(F_\theta(x_{\mathrm{Alice}}), G_\phi(x_{\mathrm{Bob}})\right). \tag{$\prod^{-1}$}$$

for some convex-concave function $\mathcal{L}$, typically referred as attack's loss [117]. In this regard, we formally establish that Algorithm AltGDA converges to an $\epsilon$-Nash equilibrium with iteration complexity $\tilde{O}\left(\frac{1}{\epsilon} \log\left(\frac{1}{\epsilon}\right)\right)$ under high-probability guarantees (Theorem 3.5). To the best of our knowledge, this provides the first open-box, provable convergence result for input-optimization attacks based on randomly initialized overparameterized neural networks, matching and theoretically explaining the experimental observations of [99] and [117].

## 1.2 Challenges and Our Approach: Bridging Overparameterization with Strategic Learning

**Back to minimization.** The optimization of min-max objectives—especially convex–concave or structured non-convex games—has been extensively studied (for an appetizer see Appendix A.1-A.2 and references therein). However, the dynamics of gradient-based methods in games where players are parameterized by neural networks remain far less understood. In minimization of training loss, a powerful lens for analyzing the success of gradient descent (GD) is the theory of *overparameterization* and the *Neural Tangent Kernel* (NTK). In the infinite-width limit, GD converges provided the NTK's smallest eigenvalue remains bounded away from zero. For finite-width networks, convergence proofs typically hinge on two ingredients: (i) good NTK conditioning at initialization, and (ii) negligible drift of the NTK during training [85, 27, 13, 106], ensuring that an underlying Polyak–Łojasiewicz (PŁ) condition is maintained.

**Extending to Games: The spectrum path.** Even simple hidden zero-sum games, where players are parameterized by two-layer neural networks with smooth activations, can cause vanilla GDA to diverge arbitrarily [114]. Although PŁ-based convergence for minimization has been understood since the classical works of Polyak and Łojasiewicz [90, 73], analogous results for min-max optimization have only recently emerged [124, 125, 60]. More recently, hidden convexity has been shown to imply a PŁ structure—both in minimization [41] and in min-max games [60]. However, this reduction to PŁ-condition reveals a key technical obstacle: hidden convexity alone cannot safeguard convergence if the Jacobians of the players' mappings suffer from near-singularities—i.e., if the least singular value approaches zero. In this regime, the effective PŁ-modulus degenerates, the gradient dominance property and convergence guarantees break down. Thus, the evolution of singular values under the employed learning dynamics becomes central challenge.

In this work, we adopt the *alternating gradient descent-ascent* (AltGDA) method, which mirrors natural sequential play between agents. From a technical standpoint, alternation proves crucial: simultaneous one-timescale GDA (SimGDA) may diverge both in case of hidden convex–concave games [114] and two-sided-PŁ games [124]. Additionally, alternation has been explored as an acceleration and stabilization tool for min-max optimization [66, 128].

- **AltGDA Path Length:** Hence, our first central technical contributions is a tight control of the *path length* of AltGDA iterates (Lemma 3.3). We show that AltGDA trajectories remain confined within a bounded region around initialization, preventing severe deterioration of hidden convex–concave structure (e.g., Jacobian conditioning). While path-length bounds are relatively straightforward in minimization—by directly unrolling GD iterations—in min-max problems, the alternating structure introduces significant complications for such ad-hoc analysis. To circumvent this, we employ a carefully designed *potential function*—a weighted interpolation between the two players' Nash gaps— by [124], which may be of independent interest.

Beyond bounding the trajectory, two additional challenges arise relative to standard supervised learning:

- **Output Dimension:** In games, neural networks output distributions over actions or more generally higher-dimensional vectors, unlike scalar labels in classification tasks. Estimating the singular value spectrum of such vector-output neural networks is more subtle. To address this, we arrive at Lemma 3.7, by adapting techniques from [106] which essentially combines Hermite expansions of hidden layer outputs, first-order Taylor series expansion and Lipschitzness of Jacobians, and high-probability concentration bounds for random Gaussian matrices.

- **Average-Case Analysis of Input Min-Max Games:** A similar approach is employed for input-optimization games, where the roles of inputs and weights are reversed. From a worst-case perspective, there exist constructions leading to rank-deficient Jacobians and failure of GDA due to convergence to spurious local optima [114], our analysis takes an average-case view. Specifically, we show that min-max input attacks, solved via AltGDA, succeed with high probability when the neural network mappings are randomly sampled with Gaussian initializations (Theorem 3.5).

- **General Loss Structures:** Unlike many prior works, which rely on the non-linear least squares structure of supervised losses to control dynamics [106, 69, 70], we allow general separable latent objectives combining strongly convex regularizers and bilinear couplings. This more general setting requires significantly stronger control on the optimization trajectory and leads to a fundamentally different overparameterization scaling, namely $\Omega(n^3)$ compared to $\Omega(n)$ in pure minimization settings (Theorem 3.8)

## 2  Preliminaries

We begin by introducing the standard notions of smoothness and Lipschitz continuity that will be used throughout this work. All norms are taken to be the Euclidean ($\ell_2$) norm unless otherwise stated.

**Lipschitz Continuity, Smoothness, and Strong Convexity.**  Let $f : \mathbb{R}^d \to \mathbb{R}$ be a differentiable function. We say that $f$ is $L_f$-Lipschitz continuous and $L_{\nabla f}$-smooth if there exist constants $L_f, L_{\nabla f} > 0$ such that

$$|f(u) - f(v)| \le L_f \|u - v\|, \quad \|\nabla f(u) - \nabla f(v)\| \le L_{\nabla f} \|u - v\|, \quad \forall u, v \in \mathbb{R}^d.$$

Moreover, $f$ is $\mu$-strongly convex if there exists $\mu > 0$ such that

$$f(v) \ge f(u) + \langle \nabla f(u), v - u \rangle + \frac{\mu}{2} \|v - u\|^2, \quad \forall u, v \in \mathbb{R}^d.$$

Similarly, for a parametrized mapping (e.g., the neural network) $M_\theta(x) : \mathbb{R}^{d_0} \to \mathbb{R}^{d_2}$ with parameters $\theta \in \mathbb{R}^M$, we say $M_\theta$ is $\beta_M$-smooth (w.r.t. $\theta$) at fixed input $x$ if

$$\sigma_{\max} \left( \nabla_\theta M_\theta(x) - \nabla_\theta M_{\theta'}(x) \right) \le \beta_M \|\theta - \theta'\|, \quad \forall \theta, \theta' \in \mathbb{R}^M, \tag{2}$$

where $\sigma_{\max}(\cdot)$ denotes the largest singular value and $\nabla_\theta M_\theta(x)$ is the Jacobian of $M_\theta(x)$ with respect to $\theta$.

**Finite-Sample Parametrized Min-Max Setting.**  Then, we unroll the general hidden convex–concave model of ([ ]) to the finite-sample empirical risk minimization (ERM) setting, assuming access to a (possibly labeled) dataset $\mathcal{D} = (\mathcal{D}_F, \mathcal{D}_G) = \{(x_i, y_i)\}_{i=1}^n$ of size $n$. Formally, we consider the following optimization problem:

$$\min_{\theta \in \mathbb{R}^{d_\theta^{(F)}}} \max_{\phi \in \mathbb{R}^{d_\phi^{(G)}}} \mathcal{L}_\mathcal{D}(F_\theta, G_\phi) := I_1^{\mathcal{D}_F}(F_\theta) + I_2^{(\mathcal{D}_F, \mathcal{D}_G)}(F_\theta, G_\phi) - I_3^{\mathcal{D}_G}(G_\phi), \tag{$\diamond$}$$

where the mappings $F_\theta : d_0^{(F)} \to \mathbb{R}^{\dim(\mathcal{S}_F)}$ and $G_\phi : d_0^{(G)} \to \mathbb{R}^{\dim(\mathcal{S}_G)}$ are smooth functions parametrized by $\theta$ and $\phi$ (e.g., neural networks). The individual components and the bilinear coupling expand as:

- $I_1^{\mathcal{D}_F}(F_\theta) = \sum_{i \in [|\mathcal{D}_F|]} \ell_i(y_i, F_\theta(x_i))$, $I_3^{\mathcal{D}_G}(G_\phi) = \sum_{j \in [|\mathcal{D}_G|]} \ell_j(y_j, G_\phi(x_j))$,
- $I_2^{(\mathcal{D}_F, \mathcal{D}_G)}(F_\theta, G_\phi) = \sum_{i \in [|\mathcal{D}_F|]} \sum_{j \in [|\mathcal{D}_G|]} F_\theta(x_i)^\top A(x_i, x_j, y_i, y_j) G_\phi(x_j)$,

where for each sample pair $e_{ij}(x_i, x_j, y_i, y_j)$, the coupling matrix $A(x_i, x_j, y_i, y_j) \in \mathbb{R}^{\dim(\mathcal{S}_F) \times \dim(\mathcal{S}_G)}$ encodes interactions between players.

**Blanket Assumptions on the Loss and Coupling Terms.**  We impose the following structural assumptions on the loss components and bilinear couplings appearing in the finite-sample min-max objective ($\diamond$).

**Assumption 2.1** (Smoothness, Hidden Strong Convexity, and Gradient Control)**.**

(i) **Smoothness:** Each sample-wise individual loss $\ell(y, \mathrm{Map}_w(x))$ is differentiable and $L$-smooth with respect to $\mathrm{Map}_w(x)$.

(ii) **Coupling Structure:** Each bilinear coupling matrix $A(x_i, x_j, y_i, y_j)$ is known, fixed, and has bounded operator norm.

(iii) **Hidden Strong Convexity:** Each sample-wise individual loss $\ell(y, h = \mathrm{Map}_w(x))$ is strongly convex with respect to the neural network output $h$.

(iv) **Gradient Growth Condition:** There exist constants $A_1, A_2, A_3 > 0$ such that for all $h \in \mathbb{R}^{d_{\text{out}}}$ and $y \in \mathcal{Y}$, the (latent) gradient of each loss $\ell(y, h)$ satisfies:

$$\|\nabla_h \ell(y, h)\| \le A_1 \|h\| + A_2 \operatorname{diam}(\mathcal{Y}) + A_3.$$

**Remark 2.2.** Item (i) ensures the applicability of gradient-based methods, while Items (i)–(iii) imply that the overall loss $\mathcal{L}_\mathcal{D}$ is $(L_\mathcal{L}, L_{\nabla \mathcal{L}})$-smooth and $(\mu_\theta, \mu_\phi)$-hidden-strongly convex–concave, with constants determined by the structure of $\ell$ and $A(\cdot)$. For standard strongly convex losses (e.g., MSE, logistic loss, cross-entropy with $\ell_2$-regularization), the gradient with respect to the network output

is controlled as in Item (iv) by an affine function of the output norm, with the leading coefficient proportional to the strong convexity modulus, $A_1 = \Theta(\mu)$. [1]

**Neural Network and Training Data Model.**

**Definition 2.3** (Two-layer Neural Network). We consider two-layer neural networks (often referred to as shallow networks). Specifically, such a network $h$ is defined by:

$$h(x) = \mathrm{Map}_{w=(W_1, W_2)}(x) = W_2^{(h)} \psi\left(W_1^{(h)} x\right),$$

where $x \in \mathbb{R}^{d_0^{(h)}}$, $W_1^{(h)} \in \mathbb{R}^{d_1^{(h)} \times d_0^{(h)}}$, $W_2^{(h)} \in \mathbb{R}^{d_2^{(h)} \times d_1^{(h)}}$, and $\psi : \mathbb{R} \to \mathbb{R}$ is an activation function applied coordinate-wise.

**Assumption 2.4** (Properties of the Two-layer Neural Network). We assume:

- $h(\cdot)$ is twice-differentiable and $\beta_h$-smooth with respect to $(W_1^{(h)}, W_2^{(h)})$.
- $\psi$ is twice-differentiable with $\psi(0) = 0$, bounded first and second derivatives ($\dot{\psi}_{\max}, \ddot{\psi}_{\max}$), and finite Hermite norm $\|\psi\|_{\mathcal{H}} < \infty$[2].
- The training data $(X, Y) \in \mathbb{R}^{d_0^{(h)} \times n} \times \mathbb{R}^{d_2^{(h)} \times n}$ satisfies $\|x_i\| = 1 \ \forall i \in [n]$ and $\|Y\| \leq 1$.
- $\sigma_{\max}\left((W_2^{(h)})_k\right) = \mathcal{O}\left(\frac{\dot{\psi}_{\max}}{\ddot{\psi}_{\max}}\right)$ for all $k \in \mathbb{Z}_{\geq 0}$[3].

Although we assume the activation function $\psi$ to be twice differentiable—thereby excluding non-smooth activations such as ReLU—our results naturally extend to smooth approximations like the Gaussian Error Linear Unit (GeLU) [55] and the softplus function [39], which have been shown empirically to perform comparably or even better than ReLU in several settings [4] [12, 40]. The performance of gradient-based training depends critically on the geometry of the training data. A standard proxy for data diversity is the well-conditioning of sample matrix X (with input vectors as rows), under standard random designs such as isotropic or sub-Gaussian inputs [86, 110, 111, 95].

**Assumption 2.5** (Spectral Properties of the Data Matrix). Let $X \in \mathbb{R}^{n \times d}$ denote the data matrix whose rows $x_i$ satisfy $\|x_i\|_2 = 1$ for all $i$. We assume that the number of samples satisfies $n \geq d$, and that X is "*generic*" in the sense that $\sigma_{\min}(X^{*r}) = \Omega(1)$ and $\sigma_{\max}(X) = O\left(\sqrt{n/d}\right)$[5], where $n$ is the number of samples and $d$ is the ambient input dimension.

For a fair comparison with the minimization literature, in the main body of the paper we adopt the data genericity assumption. Interested readers can refer to appendix for fine-grained width bounds.[6]

**Solution concept.** Note that while our min-max objective $\mathcal{L}_{\mathcal{D}}(F_\theta, G_\phi)$ is not convex–concave in $F_\theta, G_\phi$, it is (strongly) convex–concave in the outputs of $F_\theta$ and $G_\phi$, i.e., hidden strongly-convex–concave. Our analysis leverages precisely this *hidden* structure. Specifically, [41, Proposition 2] states that if $\min_\theta f(\theta)$ where $f(\theta) = F(H(\theta))$ and $F$ is strongly convex while $H$ is a smooth map (e.g., a neural net), then $f$ satisfies the PŁ-condition. Thus, hidden strong convexity implies PŁ-condition, even for nonconvex objectives. Utilizing this along with [30, Proposition of 4.1], we can define PŁ-moduli for our min-max objective in terms of the smallest singular values of the neural network Jacobians:

---

[1]Controlling the gradient growth is critical for non-asymptotic overparameterization bounds, as it ensures that iterates remain within regions where hidden convexity persists. A detailed discussion and examples are deferred to Appendix D.

[3]This is needed for upper bound on maximum singular value of $h$'s Jacobian.

[3]Given random Gaussian initialization and ensuring that iterates never leave a finite-radius ball around initialization, we can safely assume the maximum singular value is bounded from above for all iterates $k$.

[4]Moreover, we expect that the smoothness assumption can be relaxed. Since AltGDA includes subgradient variant, our analysis could likely be extended to (almost) smooth activations such as ReLU, by carefully treating the measure-zero set of non-differentiability points. We leave this technical refinement to future work, as the core phenomena should remain qualitatively unchanged.

[5]$\sigma_{\max}(X) = O(\sqrt{n/d})$ w.h.p. when, for e.g., X has i.i.d. $\mathcal{N}(0,1)$ entries [86, Section II.A]. For an arbitrary, fixed $X$, $\sigma_{\max}(X) = \|X\|_2 \leq \|X\|_F = \sqrt{n}$ ($\because \|x_i\|_2 = 1 \ \forall i$). See Remark G.5 in Appendix G.

[6]The assumption $n \gtrsim d$ ensures that $X$ is sufficiently tall to avoid rank deficiency and the minimum singular value of the Khatri-Rao product $\sigma_{\min}(X^{*r})$ serves as natural measures of the dataset's well-conditioning. Intuitively, Assumption 2.5 reflects that the dataset covers the input space sufficiently uniformly, ensuring that no direction is either too collapsed or too amplified. Such a balance is critical for achieving stable optimization dynamics and avoiding pathological trap into lower-dimensional subspaces during training.

**Fact 2.6** (Reduction to Two-Sided PŁ-condition [41, 30]). The loss function $\mathcal{L}_{\mathcal{D}}$ satisfies a two-sided Polyak–Łojasiewicz (PL) condition with parameters $\mu_\theta \sigma_{\min}^2(\nabla_\theta F_\theta)$ and $\mu_\phi \sigma_{\min}^2(\nabla_\phi G_\phi)$, where $\sigma_{\min}(\cdot)$ denotes the smallest singular value of the corresponding Jacobian mappings.

This reduction resolves several challenges inherent to general nonconvex–nonconcave min-max problems. First, it unifies several optimality notions—namely, *global minimax*, *saddle point*, and *gradient stationarity*—which, in general settings, need not coincide. For formal definitions see Appendix E. In the case where the objective satisfies a two-sided Polyak–Łojasiewicz (PŁ) condition, these notions become equivalent even at their $\epsilon$-approximate versions. We formalize this via the following lemma:

**Lemma 2.7** (Lemma 2.1 in [124], Appendix C in [60]). *If the objective function $f$ satisfies the two-sided PŁ-condition, then all three notions in Definition E.1 are equivalent:*

$$\epsilon - \textit{(Saddle Point)} \iff \epsilon - \textit{(Global Min-Max)} \iff \epsilon - \textit{(Stationary Point)} \quad \forall \epsilon \geq 0$$

Second, as discussed in the introduction, saddle points may not exist in general nonconvex–nonconcave problems. Therefore, we explicitly adopt the following benign[7] assumption:

**Assumption 2.8** (Existence of Saddle Points). The objective function $\mathcal{L}(\theta, \phi)$ admits at least one saddle point. Moreover, for any fixed $\phi$, $\min_{\theta \in \mathbb{R}^m} \mathcal{L}(\theta, \phi)$ has a non-empty solution set and a finite minimum value. Similarly, for any fixed $\theta$, $\max_{\phi \in \mathbb{R}^n} \mathcal{L}(\theta, \phi)$ has a non-empty solution set and a finite maximum value.

**Examples of hidden neural min-max optimization.** Due to space limitations, we defer a comprehensive list of examples and references to Appendix B. To build intuition, we present below two representative bilinear examples that highlight the key structural differences. We broadly distinguish two principal types of ML-driven min-max problems

- **Network Optimization:** Problems where optimization is performed over neural network parameters given a fixed dataset (training over weights). *This setting captures tasks such as generative modeling or robust adversarial reinforcement learning.*

$$\text{Example}: \min_\theta \max_\phi F_\theta(x)^\top A G_\phi(x').$$

- **Input Optimization:** Problems where network parameters are fixed (e.g., random initialization), and optimization occurs over the input space (e.g., adversarial perturbations). *This corresponds to input-driven optimization problems such as adversarial attack design.*

$$\text{Example}: \min_{x_{\text{Alice}} \in \mathcal{D}_F} \max_{x_{\text{Bob}} \in \mathcal{D}_G} \left( F_\theta(x_{\text{Alice}})^\top A G_\phi(x_{\text{Bob}}) \right).$$

## 3  Our Results

*Alternating Gradient Descent-Ascent* (AltGDA) proceeds by sequentially updating the parameters of the min-player $\theta$ and the max-player $\phi$, leveraging the most recent gradient information at each step. The updates take the form:

$$\theta^{(t)} = \theta^{(t-1)} - \eta_\theta \nabla_\theta \mathcal{L}_{\mathcal{D}}(\theta^{(t-1)}, \phi^{(t-1)}), \quad \phi^{(t)} = \phi^{(t-1)} + \eta_\phi \nabla_\phi \mathcal{L}_{\mathcal{D}}(\theta^{(t)}, \phi^{(t-1)}) \quad \text{(AltGDA)}$$

where $\eta_\theta, \eta_\phi > 0$ denote the respective step sizes.

Our analysis builds upon the framework of Yang, Kiyavash, and He [124], which guarantees $\log(1/\epsilon)$ convergence under a two-sided PL condition. In our setting, the PL moduli depend on the smallest singular values $\sigma_{\min}$ of the Jacobians $\nabla_\theta F_\theta$ and $\nabla_\phi G_\phi$, which must remain bounded away from zero throughout the optimization trajectory (Fact 2.6). This dependence is critical, as both the PL constants and the step sizes in AltGDA scale inversely with $\sigma_{\min}$.

---

[7]This assumption is mild in our setting for two reasons:
- In generative tasks such as GANs, the existence of a saddle point corresponds to operating in the *realization regime*, where the generator can fully match the data distribution [51].
- Following Gidel et al. [48], if the parameter spaces for $\theta$ and $\phi$ are bounded, saddle point existence can be guaranteed by classical minimax theorems. While we do not explicitly constrain the parameter spaces, our analysis shows that the iterates of the Alternating GDA (AltGDA) method remain confined within a bounded region. Thus, we can effectively assume boundedness without loss of generality.

Hence, we first establish that, under suitable random initialization and sufficient overparameterization, the initialization satisfies $\sigma_{\min}(\nabla_\theta F_\theta), \sigma_{\min}(\nabla_\phi G_\phi) \geq cB$ with high probability (see Lemmas 3.4 and 3.7). Furthermore, by smoothness of the neural mappings, there exists a Euclidean ball $\mathcal{B}((\theta_0, \phi_0), R)$ within which the singular values of the Jacobians remain well-conditioned, i.e., $\sigma_{\min} > B > 0$. The radius is given by $R = \frac{\mu_{\text{Jac}}}{2\beta}$, where $\mu_{\text{Jac}} := \max\left\{\mu_{\text{Jac}}^{(F)}, \mu_{\text{Jac}}^{(G)}\right\}$ and $\beta := \min\{\beta_F, \beta_G\}$. This result parallels Lemma 1 of Song et al. [106], adapted here to the alternating min-max setting.

However, the optimization trajectory could, in principle, leave this region. To prevent this, we analyze the path length of AltGDA using Yang, Kiyavash, and He [124]'s Lyapunov potential function, rather than directly unrolling the iterates—which would be analytically cumbersome due to alternation:

**Definition 3.1** (Lyapunov Potential [124]). For a min-max objective function, $\mathcal{L}(\theta, \phi)$, we define the Lyapunov potential at time $t$ as $P_t = (\max_\phi \mathcal{L}(\theta_t, \phi) - \mathcal{L}(\theta^\star, \phi^\star)) + \lambda(\max_\phi \mathcal{L}(\theta, \phi_t) - \mathcal{L}(\theta_t, \phi_t))$. (Note that the choice of $\lambda$ will not affect our conclusions about overparameterization in this paper.)

**Lemma 3.2** (Theorem 3.2 in [124]). *Suppose the min-max objective function $\mathcal{L}(\theta, \phi)$ is $L_{\nabla\mathcal{L}}$-smooth and satisfies the two-sided PŁ-condition with $(\mu_\theta, \mu_\phi)$. Then if we run AltGDA with $\eta_\theta = \frac{\mu_\phi^2}{18L_{\nabla\mathcal{L}}^3}$ and $\eta_\phi = \frac{1}{L_{\nabla\mathcal{L}}}$, then $\|\theta_{t+1} - \theta_t\| + \|\phi_{t+1} - \phi_t\| \leq \sqrt{\alpha}c^{t/2}\sqrt{P_0}$ where constants $\alpha$ and $c \in (0, 1)$ depend only on $L_{\nabla\mathcal{L}}, \mu_\theta, \mu_\phi$ and $P_0$ is the Lyapunov potential at time $t = 0$. (Please refer to Remark E.5 for the exact expressions for $\alpha$ and $c$.)*

By way of contradiction, let $T$ denote the first iteration such that $(\theta_T, \phi_T) \notin \mathcal{B}((\theta_0, \phi_0), R)$. We will show that, with high probability, the AltGDA trajectory remains within this ball by proving that its total path length is strictly less than $R$.

Indeed, AltGDA path length satisfies: $\ell(T) \triangleq \sum_{t=0}^{T-1} (\|\theta_{t+1} - \theta_t\| + \|\phi_{t+1} - \phi_t\|) \leq \frac{\sqrt{2\alpha_1}}{1 - \sqrt{c}} \cdot \sqrt{P_0}$. Therefore, it suffices to show that $\sqrt{P_0} \leq R/2$ with high probability. The following lemma provides an upper bound on $P_0$ in terms of the gradient norms:

**Lemma 3.3** (Upper Bound on Initial Potential $P_0$). *Suppose the min-max objective $\mathcal{L}(\theta, \phi)$ is $L_\mathcal{L}$-Lipschitz and satisfies a two-sided PŁ condition with constants $(\mu_\theta, \mu_\phi)$. Then the initial Lyapunov potential $P_0 \leq L_\mathcal{L}(C_1 \cdot \|\nabla_\theta\mathcal{L}(\theta_0, \phi_0)\| + C_2 \cdot \|\nabla_\phi\mathcal{L}(\theta_0, \phi_0)\|)$, where $C_1, C_2 = \Theta\left(L_\mathcal{L}/\mu_\theta^3\right)$.*

It is clear that bounding $P_0$ requires controlling the gradient norms at initialization, which—in our neural setting—requires bounding both the output norm and the spectral norm $\sigma_{\max}$ of the Jacobian via the chain rule. Lemmas 3.4, G.2 and 3.7 provide these bounds under standard overparameterization and Lipschitz stability conditions. As a result, we obtain $P_0 \leq \kappa R^2$ for some constant $\kappa < 1$ determined by the network width. Thus, with sufficient overparameterization, the iterates remain confined within the well-conditioned region $\mathcal{B}((\theta_0, \phi_0), R)$.

Interestingly, this analysis not only ensures that the iterates stay within a region where the PŁ-condition holds, but also reveals a beneficial side effect: since the potential function captures a weighted average of Nash gaps and is monotonically decreasing, a small initial value of $P_0$ implies that the initialization is already *mildly close* to equilibrium. Consequently, both convergence and geometric stability are maintained throughout training.

## 3.1 Input-Optimization Min-Max Games

Here, we consider the input-optimization game between two neural networks $F_\theta$ and $G_\phi$ in hidden bilinear objective with $\ell_2$-regularization defined as follows for a given payoff matrix, $A$:

$$\mathcal{L}(\theta, \phi) = F(\theta)^\top AG(\phi) + \frac{\varepsilon}{2}\|F(\theta)\|^2 - \frac{\varepsilon}{2}\|G(\phi)\|^2 \tag{3}$$

This game has been proposed by [114] and experimentally analyzed by [99]. Here, $F_\theta$ and $G_\phi$ are defined similar to Definition 2.3 as $F(\theta) = W_2^{(F)}\psi(W_1^{(F)}\theta)$ and $G(\phi) = W_2^{(G)}\psi(W_1^{(G)}\phi)$ but with parameters $\theta, \phi$ as inputs and randomly initalized $W_k^{(F)} \sim \mathcal{N}(0, \sigma_{k,F}^2), W_k^{(G)} \sim \mathcal{N}(0, (\sigma_{k,G}^2)$, $k \in \{1, 2\}$ along with differentiable activation function $\psi$ (e.g. GeLU). Therefore, the partial derivatives w.r.t. $\theta$ and $\phi$ will be as follows:

$$\nabla_\theta f(\theta, \phi) = (\nabla_\theta F_\theta)^\top AG(\phi) + \varepsilon(\nabla_\theta F_\theta)^\top F(\theta)$$

$$\nabla_\phi f(\theta, \phi) = (\nabla_\phi G_\phi)^\top A^\top F(\theta) - \varepsilon (\nabla_\phi G_\phi)^\top G(\phi) \tag{4}$$

Using these and Lemma 3.3, we can now bound $P_0$ as follows:

$$\implies P_0 \le \|F(\theta_0)\| \cdot (\varepsilon L_{\mathcal{L}} C_1 \sigma_{\max}(\nabla_\theta F_{\theta_0}) + L_{\mathcal{L}} C_2'(1 + \lambda)\sigma_{\max}(A)\sigma_{\max}(\nabla_\phi G_{\phi_0}))$$
$$+ \|G(\phi_0)\| \cdot (L_{\mathcal{L}} C_1 \sigma_{\max}(A)\sigma_{\max}(\nabla_\theta F_{\theta_0}) + \varepsilon C_2'(1 + \lambda)\sigma_{\max}(\nabla_\phi G_{\phi_0})) \tag{5}$$

Since we want to stay within the ball $\mathcal{B}((\theta_0, \phi_0), R)$, we can ensure $P_0 = \kappa R^2$ by controlling each term in Equation (5) accordingly that ultimately yields Theorem 3.5. For this, we would additionally need to prove Lemma 3.4 as stated below. (Please see Appendix F for proof).

**Lemma 3.4.** *Consider a neural network $F_\theta = F(\theta) = W_2^{(F)}\psi(W_1^{(F)}\theta)$ as defined above. Say $\psi$ is the GeLU activation function, $d_1^{(F)} \ge 256 \max\{d_0^{(F)}, d_2^{(F)}\}$ and $\sigma_1^{(F)} = \mathcal{O}\left((d_1^{(F)}\|\theta\|^2)^{-0.5}\right)$. Then, w.p $\ge 1 - e^{-\Omega(d_1^{(F)})}$,*

*(i) the singular values of the Jacobian $\nabla_\theta F_\theta$ are bounded as*

$$\sigma_{\min}(\nabla_\theta F_\theta) = \Omega\left(\sigma_{1,F} \cdot \sigma_{2,F} \cdot d_1^{(F)}\right) \quad and \quad \sigma_{\max}(\nabla_\theta F_\theta) = \mathcal{O}\left(\sigma_{1,F} \cdot \sigma_{2,F} \cdot d_1^{(F)}\right) \tag{6}$$

*(ii) $F(\theta)$ is $\beta_F$-smooth where $\beta_F = \Theta\left(\sigma_{1,F}^2 \cdot \sigma_{2,F} \cdot (d_1^{(F)})^{3/2}\right)$.*

**Theorem 3.5.** *Consider two neural networks $F(\theta), G(\phi)$ as defined in Lemma 3.4 above. For the regularized hidden bilinear min-max objective $\mathcal{L}(\theta, \phi)$ as defined in Equation 3 above, AltGDA reaches $\varepsilon$-saddle point w.p. $\ge 1 - e^{-\Omega(d_1^{(F)})}$ if $(\theta_0, \phi_0)$ and standard deviations $\sigma_{k,F}$ and $\sigma_{k,G}$, $k \in \{1, 2\}$ are chosen such that $\sigma_{k,F/G} = \Theta(\frac{\text{poly}(1/d_1)}{\sigma_{\max}(A)})$:*

To our knowledge, this is the first fine-grained result for overparametrized networks that establishes an $O(\epsilon)$-approximate minimax solution for the hidden bilinear setting originally proposed by Vlatakis-Gkaragkounis, Flokas, and Piliouras [114].

## 3.2 Neural-Parameters Min-Max Games

Now we analyse the case of Neural-Parameters Min-Max Games as described in Section 2. In particular, when both players are two-layer neural networks, through Lemma 3.7, with high probability the Jacobians are non-singular for random Gaussian initializations which ensures that the $\prod$ games with such networks will satisfy 2-sided PŁ-condition with high probability. Consequently, given appropriate initialization conditions for the networks (Assumption 3.6, Equation (9)), just like in Section 3.1, we can show that AltGDA converges to the saddle point by ensuring $P_0 = \kappa R^2$ via requiring both the networks to have at least cubic overparameterization (Theorem 3.8).

**Initialization Scheme 3.6** (Random Initialization). We consider the following initialization scheme for a two-layer neural network, $F$, as defined in Definition 2.3:

$$(W_1^{(F)})_0 \sim \mathcal{N}(0, \sigma_{1,F}^2 I) \qquad (W_2^{(F)})_0 \sim \mathcal{N}(0, \sigma_{2,F}^2 I) \tag{7}$$

**Lemma 3.7** (Lemma 3 & Appendix E.1–E.4 in [106]). *Suppose that a two-layer neural network, $F$, as defined in Definition 2.3, satisfies Assumption 2.4 and $\tau^{r_1}|\psi(a)| \le |\psi(\tau a)| \le \tau^{r_2}|\psi(a)|$, respectively for all $a$, $0 < \tau < 1$, and some constants $r_1, r_2$. Then w.h.p. the neural network*

*(i) Jacobian has following bounds on its singular values*

$$\sigma_{\min}(\nabla_\theta F_\theta) = \tilde{\Omega}\left(\sigma_{1,F}^{r_1}\sqrt{d_1^{(F)}}\right) \quad and \quad \sigma_{\max}(\nabla_\theta F_\theta) = \tilde{\mathcal{O}}\left(\sigma_{1,F}^{r_2}\sqrt{n \cdot d_1^{(F)}}\right) \tag{8}$$

*(ii) is $\beta_F$-smooth with $\beta_F = \sqrt{2}\sigma_{\max}(X)(\dot{\psi}_{\max} + \ddot{\psi}_{\max}\chi_{\max})$ where $\chi_{\max} = \sup_V \sigma_{\max}(V)$.*

**Theorem 3.8** ($\prod$ Games with AltGDA). *Suppose there are two two-layer neural networks, $h_\theta$, $g_\phi$ as defined in Definition 2.3 which satisfy Assumption 2.4 and $\tau^{r_1}|\psi(a)| \le |\psi(\tau a)| \le \tau^{r_2}|\psi(a)|$, respectively for all $a$, $0 < \tau < 1$, and some constants $r_1, r_2$. Suppose the network parameters $\theta_0$ and $\phi_0$ are randomly initialized as in initialization Scheme 3.6 with $(\sigma_{1,F}, \sigma_{2,F})$ and $(\sigma_{1,G}, \sigma_{2,G})$, respectively, which satisfy*

$$\sigma_{1,F} \cdot \sigma_{2,F} \lesssim \frac{1}{\sqrt{d_0^{(F)} d_1^{(F)}}} \quad and \quad \sigma_{1,G} \cdot \sigma_{2,G} \lesssim \frac{1}{\sqrt{d_0^{(G)} d_1^{(G)}}} \tag{9}$$

*and suppose that the hidden layer widths $d_1^{(F)}$ and $d_1^{(G)}$ for the two networks $F$ and $G$ satisfy*

$$d_1^{(F)} = \widetilde{\Omega}\left(\mu_\theta^2 \frac{n^3}{d_0^{(F)}}\right) \quad and \quad d_1^{(G)} = \widetilde{\Omega}\left(\mu_\phi^2 \frac{n^3}{d_0^{(G)}}\right) \tag{10}$$

*where the datasets $(\mathcal{D}_F, \mathcal{D}_G)$ for both the players are assumed to be of size $n$. Then $\prod$ game correspond to an $(\mu_\theta, \mu_\phi)$-HSCSC min-max objective as defined in Equation 1 satisfying Assumption 2.1 and AltGDA with appropriate fixed step-sizes $\eta_\theta, \eta_\phi$ (see Lemma E.7) converges to the saddle point $(\theta^*, \phi^*)$ exponentially fast with high probability.*

We refer the reader to Appendix G for the proof of Theorem 3.8, and exact expressions for failure probabilities and various quantities stated in both Lemma 3.7 and Theorem 3.8

## 4   Conclusion & Future Directions

We provide the first convergence guarantees and overparameterization bounds for alternating gradient methods in input games and hidden (strongly) convex–concave neural games. Our analysis tightly links optimization trajectory control with spectral stability, ensuring convergence to near-equilibrium. If the reader would like to look beyond the technicalities around the non-asymptotic bounds, our proof techniques offer several insights for practitioners:

- **Interpretation of $\sigma_{\min}$ and Exploration:** The smallest singular value of the network Jacobian, $\sigma_{\min}$, controls how well the model explores the strategy space. When $\sigma_{\min} \approx 0$, certain strategies remain unexplored, indicating convergence to spurious subspaces. Our analysis ties this directly to the degree of overparameterization.
- **Data Geometry and Regions of Attraction:** Our results show that overparameterized networks initialized with sufficiently diverse data are more likely to fall into regions where $\sigma_{\min} > 0$, ensuring stable convergence under AltGDA. While computing $\sigma_{\min}$ per iteration is impractical, the connection offers design insights for data and architecture.

Table 1: Comparison between our paper and common practice

|  | Our paper | In practice |
|---|---|---|
| Type of Neural Network | 1-hidden-layer, fully-connected | Typically deep networks, not necessarily fully-connected (e.g., residual or convolutional layers) |
| Training Algorithm | AltGDA | Not necessarily AltGDA / mainly double-loop |
| Network Initialization | Gaussian (with variance constraints) | Similar (e.g., He, Xavier, or LeCun initializations) |

Going beyond the neural networks and training regimes considered in this paper (see Table 1 for a summary) is an important future direction. Among these, the assumption on AltGDA is arguably the most benign. In non-convex/non-concave min-max optimization, stabilization is essential. In practice, double-loop methods (e.g., approximate best-response oracles) are often used for safety, while AltGDA serves as a more parallelizable and simpler single-loop alternative. Similarly, the Gaussian initialization is closely aligned with popular schemes like He or Xavier. The main gap lies in the architecture: practical models are often very deep with fixed-width layers. While recent work has begun to explore overparameterization in deep networks for minimization tasks, our paper focuses on a more analytically tractable setting – explicitly avoiding the NTK regime to provide a non-asymptotic analysis for 1-hidden-layer networks in a game-theoretic context. We view relaxing and extending these assumptions as a promising direction for future work.

Another natural next step is to understand how these techniques extend to non-differentiable activation functions (such as ReLU) or scale to multi-player and non-zero-sum settings – especially in structured environments like polyhedral games, which share connections with extensive-form games. For instance, exploring the analogy between two-sided PŁ-conditions (for two-player games) and hypo-monotonicity in multi-agent operator theory may allow us to transfer and generalize some of the intuition and techniques from our current setting. We hope our work and these possible future directions open up rich and technically deep avenues for developing gradient-based methods tailored for structured, non-monotone multiplayer games. (See also Appendix I.)

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

# Appendix Contents

# A   Further Discussion in Prior Work

## A.1   Optimization with Hidden Structures

Hidden convexity has offered a pathway to global convergence even in otherwise nonconvex problems. Early developments include the use of hidden convexity in the analysis of cubic regularization for Newton's method [81], which achieved global rates under suitable curvature conditions. Extending such frameworks to non-smooth or stochastic optimization, however, remains an open challenge. More recently, hidden convexity has been exploited across a wide range of applications, including reinforcement learning [54, 129, 126], generative modeling [63], supply chain management [42, 24], and neural network training [119]. Further, [11] studied optimization via biased stochastic oracles, developing algorithms with $\tilde{O}(\epsilon^{-2})$ or $\tilde{O}(\epsilon^{-3})$ iteration complexity, depending on the oracle's stability, and recovering new results for hidden convex problems as an application. Prior to its adoption in game theory, policy gradient methods in reinforcement learning have leveraged hidden convexity to establish global convergence guarantees [129, 8], often relying on variance-reduced estimators or large-batch assumptions. Relatedly, hidden convexity also played a crucial role in quadratic optimization problems [9, 107], revealing convex-like structures within certain nonconvex programs.

Turning to min-max optimization, hidden convexity has proven essential for addressing the limitations of classical algorithms. Vlatakis-Gkaragkounis, Flokas, and Piliouras [114] demonstrated that even benign-looking hidden zero-sum games can exhibit cyclic behaviors or divergence when trained via vanilla gradient descent-ascent (GDA). Motivated by such failures, Vlatakis-Gkaragkounis, Flokas, and Piliouras [115] introduced the formal concept of *hidden-convex hidden-concave games*, providing the first convergence guarantees for GDA in nonconvex min-max settings under suitable structural conditions such as strict or strong hidden convexity. Parallel research on non-monotone variational inequalities has developed continuous-time flows that exploit hidden convexity-like structures to ensure global convergence [79]. Extending these ideas, Sakos et al. [99] provided a fully discrete-time algorithm along with provable guarantees for stability and convergence in hidden convex games—filling an important gap compared to previous continuous-time results. Moreover, recent work has investigated preconditioned algorithms for hidden monotone variational inequalities, including Newton-type approaches [29]. Further discussions [29] have clarified under which conditions both the latent variable space and the control space can be bounded, while maintaining uniform lower bounds on the singular values of the players' Jacobians, a critical requirement for robust convergence guarantees.

## A.2   Simultaneous, Alternating, and Extrapolated Dynamics in Convex–Concave Min-Max Optimization

Min-max optimization algorithms have a long-standing history, dating back to the original proximal point methods for variational inequality problems Martinet [74] and Rockafellar [93]. Below we present some

### A.2.1   The classic regime

To set the stage, consider the classical min-max problem of the form $\min_x \max_y f(x, y)$, a fundamental template across optimization and game theory. The most natural extension of gradient descent to such settings is the *gradient descent-ascent* (GDA) method [36], which iteratively updates $x$ to decrease $f$ and $y$ to increase $f$. GDA comes in two flavors: *simultaneous* (Sim-GDA), where $x$ and $y$ are updated in parallel, and *alternating* (Alt-GDA), where updates occur sequentially. However, despite its simplicity, even in convex–concave settings, vanilla GDA fails to guarantee convergence. In basic examples such as the bilinear game $\min_x \max_y xy$, Sim-GDA exhibits unbounded divergence, while Alt-GDA produces bounded but non-convergent iterates that circulate indefinitely [7, 49, 50, 128].

These deficiencies have led to the development of refined algorithms designed to stabilize min-max dynamics, including Extra-Gradient methods [64], Optimistic Gradient Descent [91], and negative momentum variants [50]. Many of these methods achieve accelerated rates compared to plain GDA, particularly under smoothness and convexity assumptions. Nevertheless, the bulk of this literature focuses predominantly on Sim-GDA-type algorithms, largely due to their analytical tractability.

Yet, in many real-world machine learning applications, particularly adversarial training scenarios like GANs, alternating updates more naturally model the dynamics: the generator and discriminator adjust sequentially based on each other's output. Empirical studies [51, 76] suggest that Alt-GDA often converges faster than Sim-GDA. Despite these observations, the theoretical foundations explaining the advantages of alternation remain relatively underdeveloped. A notable advancement toward understanding this phenomenon was provided by Zhang et al. [128], who analyzed local convergence rates under strong convexity–concavity (SCSC) and smoothness assumptions. Their results show that, locally, Alt-GDA achieves iteration complexity $\tilde{O}(\kappa)$ compared to the slower $\tilde{O}(\kappa^2)$ of Sim-GDA, where $\kappa = L/\mu$ denotes the condition number. Nonetheless, these results are limited to *local* convergence—i.e., after the iterates are already sufficiently close to a saddle point—and do not capture the full global behavior. More recently, further refinements have been developed combining extrapolation techniques with alternation to achieve nearly optimal condition number dependence [66].

### A.2.2 The PŁ-Regime and Nonconvex Min-Max Problems

Research has also intensified on extending convergence guarantees beyond convex–concave settings to nonconvex–nonconcave games [105, 21, 92, 109, 68, 1]. The majority of these early works often focused on settings where the objective is nonconvex in $x$ but concave in $y$, and proposed algorithms that either solve inner maximizations exactly or rely on strong assumptions (e.g., Minty Variational Inequalities [68]). Others, such as Abernethy, Lai, and Wibisono [1], introduced Hamiltonian-type methods for nearly bilinear problems but relied on second-order information. Recently, Nouiehed et al. [84] studied a class of minimax problems where the objective only satisfies a one-sided PŁ-condition and introduced the GDmax algorithm, which takes multiple ascent steps at every iteration. However in our case (i) we consider the two-sided PŁ-condition which guarantees global convergence; (ii) we consider AltGDA which takes one ascent step at every iteration. Another closely related work is [18] The authors considered a specific application in generative adversarial imitation learning with linear quadratic regulator dynamics. This is a special example that falls under the two-sided PŁ-condition.

In contrast, a major breakthrough in capturing global convergence via first-order methods was provided by Yang, Kiyavash, and He [124]. They introduced the concept of two-sided Polyak–Łojasiewicz (PŁ) inequalities for nonconvex–nonconcave min-max games, showing that alternating GDA (AltGDA) converges globally at a linear rate under this structure. Furthermore, they designed a variance-reduced stochastic AltGDA method for finite-sum objectives, achieving faster convergence. Building on this, subsequent works have refined the framework: Xu et al. [122] proposed a zeroth-order variant, Liu et al. [71] analyzed randomized stochastic accelerations, and Chen, Yao, and Luo [20] integrated Spider techniques to improve stochastic convergence rates under PŁ-conditions. Extensions to more adaptive and multi-step alternating schemes were explored by Kuruzov et al. [65], aiming to further optimize the dynamics in hidden convex-concave games. At the same time [124] presented a case where simultaneous GDA will diverge from the equilibrium while AltGDA converges to the equilibrium.

### A.3 Overparameterization in Learning Dynamics: From Minimization to Games

### A.3.1 Minimizing training loss

The phenomenon of overparameterization—where neural networks possess far more parameters than the apparent complexity of the target function—has profoundly influenced the theoretical understanding of modern machine learning. Early works rigorously explored how, despite the non-convexity of the training landscape, overparameterization enables gradient-based methods to converge to global optima [98, 38, 3]. Notably, these studies typically required substantial overparameterization, often polynomial in the size of the training data or model complexity. However, empirical observations [72, 97] suggest that even modest increases in network width—sometimes adding just a few neurons—can suffice for successful training, motivating a closer study of *mild overparameterization*.

A pivotal theoretical lens for understanding this success is the *Neural Tangent Kernel* (NTK) framework [57]. In the infinite-width limit, training dynamics linearize, and gradient descent effectively follows a kernelized gradient flow determined by the NTK, which remains nearly constant throughout training. Thus, convergence can be ensured if the NTK is well-conditioned—specifically, if its

minimum eigenvalue remains bounded away from zero. At finite but large widths, convergence analyses typically require proving two properties: (i) the NTK is well-conditioned at initialization, and (ii) it remains stable during training [85, 27, 13].

Recent works have sharpened the quantitative understanding of this regime. For instance, Song et al. [106] demonstrated that a network width of approximately $\tilde{O}(n^{3/2})$ suffices for global convergence at linear rates, improving previous state-of-the-art requirements. In parallel, studies such as [13] established NTK lower bounds that allow optimization with as few as $\Omega(\sqrt{n})$ neurons, bridging optimization and memorization capabilities. Yet, most of these developments operate within minimization frameworks—either fitting labels or adversarially robust losses [17]. The transition from minimization to *games* (e.g., adversarial training, multi-agent learning) brings new challenges.

### A.3.2 Adversarial Losses and MARL

In adversarial learning, recent works have analyzed overparameterized adversarial training primarily from a minimization perspective, focusing on robustness against worst-case perturbations rather than strategic interactions between agents [132, 45, 25]. While robust losses (and closer to our setting) induce non-convexity, the optimization target remains a minimizer rather than a saddle point.

In multi-agent and game-theoretic settings, the literature is comparatively sparser. Policy approximation in multi-agent reinforcement learning (MARL) typically relies on either tabular or linear architectures [118, 108], and extending sample-efficient learning to rich function classes like neural networks remains a frontier. A notable contribution in this direction is the work of Jin, Liu, and Yu [58], which introduced the Multi-Agent Bellman Eluder (BE) dimension as a complexity measure for MARL, enabling sample-efficient learning of Nash equilibria in high-dimensional spaces. In the context of Markov Games, Li et al. [67] studied Nash equilibria computation using kernel-based function approximation, highlighting the difficulties of exploration and generalization in high-dimensional, non-convex settings.

Despite these advances, a comprehensive theory connecting overparameterization, NTK stability, and global convergence in *multi-agent games* remains largely undeveloped. Key questions include:

- How does overparameterization affects the conditioning of multi-agent dynamics?
- whether can alternating optimization (as opposed to simultaneous updates) exploit NTK-like stability, and how regularization or architectural choices influence convergence in strategic environments?

Our work contributes to this growing effort by combining insights from the large-scale network mapppings perspective with recent advances in hidden convexity and PL conditions for nonconvex–nonconcave optimization. In particular, we highlight that under mild overparameterization, even strategic interactions—modeled via hidden convex–concave games—admit global convergence guarantees with simple gradient-based methods, provided the trajectory stays within a controlled neighborhood of initialization where the Jacobians remain well-conditioned.

### A.4 Some Empirical Studies Demonstrating Need of Larger Neural Networks for Zero-Sum Games

Since our work pertains to estimating the amount of overparameterization needed in neural networks for solving various zero-sum games, we highlight some of the works in various applied min-max contexts – adversarial training, GANs, DRO, and neural agents – which show that larger, overparameterized neural networks lead to improved convergence and performance. For instance, in case of adversarial training, Addepalli et al. [2] show improved robustness and performance in adversarial training with larger models. A few other works [61, 14, 100] empirically demonstrate how using larger architectures in GANs improve the training stability, and the quality and variation of generated images. In the realm of LLM Language agents, Karten, Nguyen, and Jin [62] show improved performance using GPT-4.0 as opposed to smaller LLMs. In the case of DRO, Pham et al. [88] show that bigger neural networks may yield better worst-group generalization.

While our main results (Theorem 3.8) concerning neural games may largely be seen to be of theoretical interest without an empirical component attached to it, we refer interested readers to Appendix C for experimental validity of our main results concerning the input games (Theorem 3.5).

# B  Examples/Applications based on hidden min-max games

In the following, we present a series of examples illustrating instances of *hidden convex–concave min-max optimization* in neural network-based settings. We distinguish between two main types of problems:

- Problems where the optimization is performed over the parameters of the neural networks, given a fixed dataset (training over network weights).
- Problems where the network parameters are fixed (randomly initialized), and the optimization is performed over the input space (input-optimization games).

## B.1  Neural Min-Max Games

We begin by illustrating two canonical examples of *input-optimization games* that naturally fall within the hidden convex–concave min-max framework. Both examples demonstrate settings where neural network mappings induce structured, but hidden, convexity and concavity, which can be exploited for provable convergence.

**Example B.1** (Generative Adversarial Networks (GANs)). A *Generative Adversarial Network* (GAN) formulates a two-player minimax game where the generator $G_\theta$ seeks to produce samples that resemble a reference distribution $p_{\text{data}}$, while the discriminator $D_\phi$ attempts to distinguish generated samples from real data. The corresponding min-max problem reads:

$$\min_\theta \max_\phi \quad \Psi(\theta, \phi) := \mathbb{E}_{x \sim p_{\text{data}}} [\log D_\phi(x)] + \mathbb{E}_{x \sim p_\theta} [\log(1 - D_\phi(x))].$$

Assuming that both $p_{\text{data}}$ and $p_\theta$ admit densities and that the support of $p_\theta$ lies within the support of $p_{\text{data}}$, one can reformulate the problem via a latent convex–concave structure. The min-max formulation arises naturally as the generator seeks to minimize the ability of the discriminator to distinguish real from fake samples, while the discriminator simultaneously maximizes its classification performance, thus modeling an adversarial dynamic between two competing objectives. Specifically, considering a distribution $p(x, x')$ that samples either a real or generated point, the loss can be decomposed as:

$$L_{x,x'}(p', D) := \log D(x) + \frac{p'(x')}{p_{\text{data}}(x')} \log(1 - D(x')),$$

which is jointly convex in $p'$ and concave in $D$. Consequently,

$$\Psi(\theta, \phi) = \mathbb{E}_{(x,x') \sim p} [L_{x,x'}(p_\theta(x), D_\phi(x'))],$$

exhibiting the GAN training objective as a *hidden convex–concave game*.

**Example B.2** (Domain-Invariant Representation Learning (DIRL)). Domain adaptation aims to train models that generalize across different domains, despite distribution shifts between training (source) and deployment (target) environments. A popular approach [44] involves learning representations that are: (i) predictive of labels in the source domain, and (ii) invariant to the domain classifier distinguishing source versus target samples. This leads to the following min-max problem:

$$\min_{\theta_f, \theta_g} \max_{\theta_{f'}} \mathbb{E}_{(x,y) \sim P_{\text{source}}} \left[ \ell(f_{\theta_f}(g_{\theta_g}(x)), y) \right] - \lambda \mathbb{E}_{(x,y') \sim P_{\text{mix}}} \left[ \ell(f'_{\theta_{f'}}(g_{\theta_g}(x)), y') \right],$$

where: (i) $g_{\theta_g}$ is the feature extractor, (ii) $f_{\theta_f}$ is the label predictor, (iii) $f'_{\theta_{f'}}$ is the domain classifier, (iv) $\ell$ denotes the classification loss, and (v) $P_{\text{source}}$, $P_{\text{mix}}$ are the source and mixed domain distributions, respectively. When the loss function is convex with respect to the neural mappings, the hidden convex–concave structure becomes apparent, fitting naturally into our theoretical framework.

**Example B.3** (Robust Adversarial Reinforcement Learning (RARL)). One of the major challenges in reinforcement learning (RL) is the difficulty of training agents under realistic conditions, often due to costly data collection or limited availability of real-world environments. To address these challenges, Pinto et al. [89] proposed an adversarial training framework wherein a learner agent and an adversary play against each other by solving the following min-max optimization problem:

$$\min_{\theta_1} \max_{\theta_2} \mathbb{E}_{s_0 \sim \rho, \, a^1 \sim \mu_{\theta_1}(s), \, a^2 \sim \nu_{\theta_2}(s)} \left[ \sum_{t=0}^{T-1} r^1(s_t, a_t^1, a_t^2) \right],$$

where: (i) $\mu_{\theta_1}$ is the learner's policy network, (ii) $\nu_{\theta_2}$ is the adversary's policy network, (iii) $r^1$ denotes the reward function, and (iv) $\rho$ is the initial state distribution.

In this setup, the learner seeks to maximize its expected reward, while the adversary perturbs the environment or dynamics to minimize the learner's performance. Such adversarial modeling captures different sources of uncertainty: either *unknown variations in the underlying Markov Decision Process (MDP)*, or deliberate *adversarial attacks* aiming to degrade the policy (e.g., disturbances in robotic control tasks or adversarial inputs in sensor-based systems). Hidden convex–concave structures can naturally arise when suitable regularizations or smooth policy parameterizations are enforced.

**Example B.4** (Adversarial Example Generation (AEG)). In an *Adversarial Example Generation* (AEG) setting, the goal is to generate adversarial perturbations that cause misclassification by a fixed classifier $f_\phi$. Formally, given clean samples $(x, y) \sim p_{\text{data}}$, a perturbation generator $G_\theta$ seeks to find an adversarial input $x'$ satisfying a distortion constraint (e.g., $\|x - x'\|_\infty \leq \epsilon$) that maximizes the classification loss. The underlying min-max optimization is:

$$\min_\phi \max_\theta \quad \Psi(\theta, \phi) := -\mathbb{E}_{(x',y) \sim p_\theta} \left[ \ell(f_\phi(x'), y) \right],$$

where $\ell$ denotes the cross-entropy loss. Here, the generator (adversary) maximizes the classification loss while the classifier seeks to minimize it. Under appropriate conditions on the neural network mappings and smoothness of the loss, the adversarial game admits a hidden convex–concave structure that can be exploited for convergence analysis.

## B.2 Distributionally Robust Optimization

In many machine learning applications, models are trained under the assumption that data is drawn from a fixed but unknown distribution. However, real-world deployment often leads to distribution shifts (e.g., label noise, adversarial perturbations, or changing environments), which can severely degrade performance. A principled way to address this issue is through *Distributionally Robust Optimization* (DRO) [77, 123], where the model is trained to perform well against the worst-case distribution within a prescribed uncertainty set. The corresponding min-max optimization problem reads:

$$\theta_{\text{DRO}}^* \in \arg \min_\theta \max_{P \in \mathcal{P}} \mathbb{E}_{(x,y) \sim P} \left[ \ell(h_\theta(x), y) \right],$$

where: (i) $h_\theta$ is the predictive model parameterized by $\theta$, (ii) $\ell$ is a loss function (e.g., cross-entropy), (iii) $\mathcal{P}$ is an uncertainty set containing distributions representing expected perturbations. Hidden convexity can emerge when $\ell$ is convex in the model outputs and the uncertainty set $\mathcal{P}$ is suitably structured.

**Example B.5** (Parametric Distributionally Robust Optimization (Parametric-DRO)). While classical DRO assumes that the uncertainty set $\mathcal{P}$ is specified manually, identifying an appropriate $\mathcal{P}$ is often challenging, especially at large deployment scales. To overcome this, Michel, Hashimoto, and Neubig [77] proposed modeling the worst-case distribution using a parameterized generative model $q_\psi$.

The resulting parametric DRO problem reads:

$$\min_\theta \max_{\psi: KL(q_\psi \| q_{\psi_0}) \leq \kappa} \mathbb{E}_{(x,y) \sim p} \left[ \frac{q_\psi(x, y)}{q_{\psi_0}(x, y)} \ell(h_\theta(x), y) \right],$$

where: (i) $h_\theta$ is the predictor network, (ii) $q_\psi$ models the perturbed distribution, (iii) $q_{\psi_0}$ approximates the empirical distribution via maximum likelihood, (iv) $\kappa$ controls the size of the KL-divergence ball around $q_{\psi_0}$. This formulation transforms the DRO problem into a min-max optimization between the model parameters $\theta$ and the perturbation parameters $\psi$, both parameterized via neural networks, fitting naturally into the hidden convex–concave framework under appropriate regularization. Observe that by Pinsker's inequality we get that KL-divergence over $\ell_1$-norm.

## B.3 Input-Optimization Min-Max Games

A recurring structure in adversarial and robust optimization tasks involves optimizing over input spaces rather than model parameters. Based on Wang et al. [117] we describe three prominent examples of such *input-optimization games* below:

**Example B.6** (Ensemble Attack over Multiple Models). Given $K$ machine learning models $\{\mathcal{M}_i\}_{i=1}^{K}$, the goal is to find a universal perturbation $\delta$ that simultaneously fools all models. The corresponding input-optimization game reads:

$$\min_{\delta \in \mathcal{X}} \max_{w \in \mathcal{P}} \sum_{i=1}^{K} w_i f(\delta; x_0, y_0, \mathcal{M}_i) - \frac{\gamma}{2}\|w - 1/K\|_2^2,$$

where $w$ encodes the relative difficulty of attacking each model, and $\gamma$ is a regularization parameter.

**Example B.7** (Universal Perturbation over Multiple Examples). Here, given a set of examples $\{(x_i, y_i)\}_{i=1}^{K}$, the objective is to find a perturbation $\delta$ that simultaneously fools all of them. The optimization problem becomes:

$$\min_{\delta \in \mathcal{X}} \max_{w \in \mathcal{P}} \sum_{i=1}^{K} w_i f(\delta; x_i, y_i, \mathcal{M}) - \frac{\gamma}{2}\|w - 1/K\|_2^2.$$

This mirrors the ensemble attack setup, but focuses on perturbing multiple inputs under a fixed model $\mathcal{M}$.

**Example B.8** (Adversarial Attack over Data Transformations). Consider robustness against transformations (e.g., rotations, translations) applied to the inputs. Given categories of transformations $\{p_i\}$, the optimization reads:

$$\min_{\delta \in \mathcal{X}} \max_{w \in \mathcal{P}} \sum_{i=1}^{K} w_i \mathbb{E}_{t \sim p_i} \left[ f(t(x_0 + \delta); y_0, \mathcal{M}) \right] - \frac{\gamma}{2}\|w - 1/K\|_2^2,$$

where $t$ denotes a random transformation sampled from $p_i$. When $w = 1/K$, this recovers the expectation-over-transformation (EOT) setup.

Observe that under a convex loss function $f$ with respect to the neural mapping $\mathcal{M}(x_0 + \delta)$ (i.e., hidden convexity in $\delta$), and given that $w$ appears linearly in both the bilinear coupling and the individual regularization term of separable framework of Section 1.1, the structure fits naturally within our hidden convex–concave framework. Albeit a careful reader might observe that our main results are stated for unconstrained min-max optimization, there are two standard ways to extend our analysis to constrained settings:

- First, by employing the two-proximal-PL framework developed in Kalogiannis et al. [60]—an improvement over the earlier formulation of Yang, Kiyavash, and He [124]—our convergence guarantees naturally generalize to simple constraint sets, such as $\ell_2$-balls for perturbations $\delta$ and simplices for the mixture weights $w$.

- Alternatively, the constraints can be incorporated directly into the objective through suitable Lagrangian penalty terms, thereby reducing the constrained min-max problem to an unconstrained form amenable to our techniques.

**Unified Perspective.** Across these examples, input-optimization problems naturally exhibit a saddle-point structure, blending adversarial robustness objectives with min-max optimization techniques. They provide concrete and practically motivated instances where hidden convexity can be exploited to ensure convergence guarantees for training and robustness analysis.

## C Demonstrating Empirical Validity of Main Result for Input Games

We consider a hidden game of Rock-Paper-Scissors where two 1-hidden layer neural networks (GeLU activations) are playing the game of the Rock-Paper-Scissors. We will see here (empirically) that, if we use random Gaussian initializations for both the neural network players as described in Theorem 3.5 to define our input game of Rock-Paper-Scissors, and if we use AltGDA for finding min-max optimal strategies for both these neural network players, then the players indeed reach the $\varepsilon$-Nash equilibrium. In particular, both the neural network players control 5-dimensional vectors $\theta, \phi \in \mathbb{R}^5$ and output a latent strategy that lies in the 2-dimensional simplex, $\Delta_2$. Both the players are optimizing the regularized bilinear bilinear objective:

$$\mathcal{L}(\theta, \phi) = F(\theta)^\top A G(\phi) + \varepsilon/2\|F(\theta) - 1/3(1,1,1)^\top\|_2^2 - \varepsilon/2\|G(\phi) - 1/3(1,1,1)^\top\|_2^2$$

where $(1/3, 1/3, 1/3)^\top$ indicates the (unique) $\varepsilon$-mixed strategy Nash equilibrium for both the players and $A = 10 \cdot \begin{pmatrix} 0 & -1 & 1 \\ 1 & 0 & -1 \\ -1 & 1 & 0 \end{pmatrix}$ is the payoff matrix[8] for the Rock-Paper-Scissor game.

Since the actual strategies of the two neural network players $(\theta, \phi)$ lie in $\mathbb{R}^5$, we can't visualize those. However, we can visualize their strategies in the latent space, i.e., in the 2-dimensional simplex instead. Figure 2 below illustrates the AltGDA trajectories for both the players (step-size = 0.01, $\varepsilon = 1$, maximum number of steps = 100,000). As we can see, both the trajectories converge to the $\varepsilon$-Nash equilibrium of the hidden game.

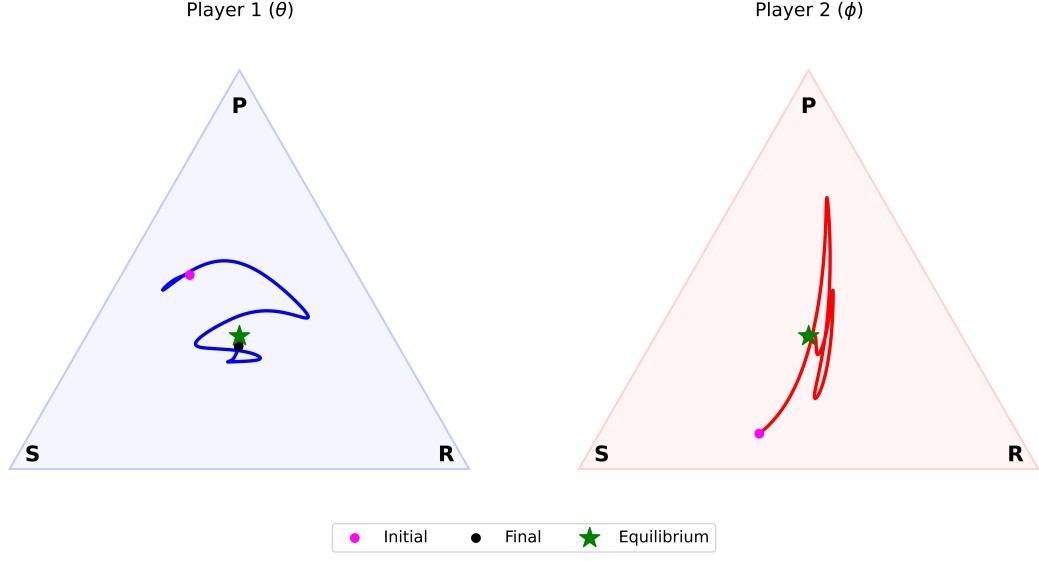

Figure 2: A trajectory of AltGDA in an $\ell_2$-regularized hidden game of Rock-Paper-Scissors. These trajectories correspond to each player's strategies in the latent space (2-dimensional simplex).

---

[8]A scaling factor of 10 was used in the payoff matrix to ensure that the gradients are not too small for the AltGDA updates.

# D Controlling the Gradient Growth: Bounds and Examples for Different Loss Functions

Controlling the gradient growth is critical for non-asymptotic overparameterization bounds, as it ensures that iterates remain within regions where hidden convexity persists. Below, we provide a detailed discussion about this bound for some commonly-used loss functions.

- **Mean-Squared Error (MSE) Loss:** One of the most commonly used losses and is suitable for regression tasks. For a given labelled data point $(x, y)$ where $x \in \mathcal{X}$ and $y \in \mathcal{Y}$, if the predictor function parameterized by parameters $\theta$ is defined as $h := h(x; \theta)$, we see that the MSE loss can be defined as:

$$\ell(y, h) = \frac{1}{2}\|y - h\|_2^2 \tag{11}$$

$$\implies \nabla_h \ell(y, h) = h - y \tag{12}$$

$$\implies \|\nabla_h \ell(y, h)\| \leq \|h\| + \|y\| \tag{13}$$

$$\leq \|h\| + \text{diam}(\mathcal{Y}) \tag{14}$$

$$\implies \|\nabla_h \ell(y, h)\| \leq A_1\|h\| + A_2\text{diam}(\mathcal{Y}) + A_3 \tag{15}$$

where $A_1 = A_2 = 1$ and $A_3 = 0$. More generally, if we had $\ell(y, h) = \frac{\mu}{2}\|y - h\|_2^2$, we would get $A_3 = 0$ and $A_1 = A_2 = \mu$ where $\mu$ is also the hidden-strong convexity modulus.

- **Logistic Loss:** Commonly used in binary classification, where $y \in \{0, 1\}$, and the prediction $h = h(x; \theta) \in \mathbb{R}$ is passed through the sigmoid function. The logistic loss is defined as:

$$\ell(y, h) = -y \log \sigma(h) - (1 - y) \log(1 - \sigma(h)), \quad \text{where } \sigma(h) = \frac{1}{1 + e^{-h}} \tag{16}$$

$$\nabla_h \ell(y, h) = \sigma(h) - y \tag{17}$$

$$\Rightarrow \|\nabla_h \ell(y, h)\| \leq 1 \tag{18}$$

Since the gradient is always in the interval $[-1, 1]$, we can take $A_1 = 0, A_2 = 0, A_3 = 1$. The logistic loss is strongly convex over compact domains or when regularized.

- **Squared Hinge Loss:** Used in support vector machines (SVMs) with a margin-based formulation. For $y \in \{-1, 1\}$ and prediction $h = h(x; \theta) \in \mathbb{R}$:

$$\ell(y, h) = \frac{1}{2} \max(0, 1 - yh)^2 \tag{19}$$

$$\nabla_h \ell(y, h) = \begin{cases} -y(1 - yh), & \text{if } yh < 1 \\ 0, & \text{otherwise} \end{cases} \tag{20}$$

$$\Rightarrow \|\nabla_h \ell(y, h)\| \leq 2|y||h| + 1 \leq 2\|h\| + \text{diam}(\mathcal{Y}) \tag{21}$$

Thus, we can choose constants $A_1 = 2, A_2 = 1, A_3 = 0$. Note: the hidden strong convexity applies within the active region $yh < 1$, and a regularization term often ensures global strong convexity.

- **Cross-Entropy with $\ell_2$-regularization:** For multi-class classification with softmax outputs $h \in \mathbb{R}^K$, target $y \in \Delta^{K-1}$ (probability simplex):

$$\ell(y, h) = -\sum_{k=1}^{K} y_k \log \left( \frac{e^{h_k}}{\sum_{j=1}^{K} e^{h_j}} \right) + \frac{\lambda}{2}\|h\|^2 \tag{22}$$

$$\nabla_h \ell(y, h) = \text{softmax}(h) - y + \lambda h \tag{23}$$

$$\Rightarrow \|\nabla_h \ell(y, h)\| \leq \|\text{softmax}(h) - y\| + \lambda\|h\| \leq 2 + \lambda\|h\| \tag{24}$$

So the gradient norm is bounded by $A_1 = \lambda, A_2 = 0, A_3 = 2$. The $\ell_2$ term ensures strong convexity with modulus $\lambda$.

# E   Proofs about two-sided PŁ & AltGDA

We begin by recalling the definitions of unconstrained continuous min-max optimality conditions of problem:

$$\min_\theta \max_\phi \mathcal{L}(\theta, \phi)$$

**Definition E.1** (Global Optima). We define three equivalent notions of optimality:

1. $(\theta^*, \phi^*)$ is a global minimax, if for any $(\theta, \phi)$: $\mathcal{L}(\theta^*, \phi) \leq \mathcal{L}(\theta^*, \phi^*) \leq \max_{\phi'} \mathcal{L}(\theta, \phi')$

2. $(\theta^*, \phi^*)$ is a saddle point, if for any $(\theta, \phi)$: $\mathcal{L}(\theta^*, y) \leq \mathcal{L}(\theta^*, \phi^*) \leq \mathcal{L}(x, \phi^*)$

3. $(\theta^*, \phi^*)$ is a stationary point, if for any $(\theta, \phi)$: $\nabla_\theta \mathcal{L}(\theta^*, \phi^*) = \nabla_\phi \mathcal{L}(\theta^*, \phi^*) = 0$.

> Thanks to gradient dominance of PŁ conditions, it is possible to prove the equivalence of these notions under two-sided PŁ condition. Observe that this reduction is crucial since gradient-based methods provide safely just a stationary point while the other two notions are global and non-trivially verifiable.

**Lemma E.2** (Lemma 2.1 in [124], Appendix C in [60]). *If the objective function $\mathcal{L}$ satisfies the two-sided PŁ condition, then all three notions in Definition E.1 are equivalent:*

$$\epsilon - (\textit{Saddle Point}) \iff \epsilon - (\textit{Global Min-Max}) \iff \epsilon - (\textit{Stationary Point}) \quad \forall \epsilon \geq 0$$

> Before we present the following lemma, we provide some intuition: This result characterizes the behavior of the optimization trajectory under hidden convex–concave structure, establishing a critical link between parameter dynamics and convergence guarantees. The lemma will formalize how small distortions in the hidden geometry impact the optimization path based on parameters $\alpha, c$.

**Lemma 3.2** (Theorem 3.2 in [124]). *Suppose the min-max objective function $\mathcal{L}(\theta, \phi)$ is $L_{\nabla \mathcal{L}}$-smooth and satisfies the two-sided PŁ-condition with $(\mu_\theta, \mu_\phi)$. Then if we run AltGDA with $\eta_\theta = \frac{\mu_\phi^2}{18 L_{\nabla \mathcal{L}}^3}$ and $\eta_\phi = \frac{1}{L_{\nabla \mathcal{L}}}$, then $\|\theta_{t+1} - \theta_t\| + \|\phi_{t+1} - \phi_t\| \leq \sqrt{\alpha} c^{t/2} \sqrt{P_0}$ where constants $\alpha$ and $c \in (0, 1)$ depend only on $L_{\nabla \mathcal{L}}, \mu_\theta, \mu_\phi$ and $P_0$ is the Lyapunov potential at time $t = 0$. (Please refer to Remark E.5 for the exact expressions for $\alpha$ and $c$.)*

> In most existing works on nonconvex–nonconcave optimization, smoothness is typically defined directly with respect to the optimization parameters. However, given our focus on a fine-grained analysis of neural min–max games, it is equally important to provide a similarly fine-grained treatment of upper bounds involving the geometry of the neural maps.
>
> Although a neural network is, in practice, a highly smooth function, global Lipschitz continuity is incompatible with strong convexity in unconstrained domains. Nonetheless, within a bounded region—such as the ball of radius $R$ where our iterates remain—local Lipschitzness can be rigorously characterized. The following lemma formalizes this bounds for both maximizer & minimizer neural network:

**Lemma E.3** (Local-Lipschitzness for smooth loss function). *Let $F_\theta$ and $G_\phi$ be neural network mappings such that they are $\beta_F$ and $\beta_G$ smooth as defined in Definition 2.3. Now let $(\theta_0, \phi_0)$ be such that Jacobian singular values for both the networks are strictly positive and bounded from above and below, $\mu_{Jac}^{(F)} \leq \sigma(\nabla_\theta F_{\theta_0}) \leq \nu_{Jac}^{(F)}$ and $\mu_{Jac}^{(G)} \leq \sigma(\nabla_\phi G_{\phi_0}) \leq \nu_{Jac}^{(G)}$. Suppose the stationary point for min-max objective $\mathcal{L}(F_\theta, G_\phi)$ as defined in Assumption 2.1 also lies in the ball, $(\theta^*, \phi^*) \in \mathcal{B}((\theta_0, \phi_0), R)$. Then there exists an $R > 0$ such that $\forall (\theta, \phi) \in \mathcal{B}((\theta_0, \phi_0), R)$, we have*

$$\max_{\phi \in \mathcal{B}(\phi_0, R)} \|\nabla_{G_\phi} \mathcal{L}(F_\theta, G_\phi)\|, \max_{\theta \in \mathcal{B}(\theta_0, R)} \|\nabla_{F_\theta} \mathcal{L}(F_\theta, G_\phi)\| \leq L_{\mathcal{L}}^{act} \tag{25}$$

*where we denote the upper bound as an 'active' Lipschitz constant $L_{\mathcal{L}}^{act}$:*

$$L_{\mathcal{L}}^{act} = 4.5 L_{\nabla \mathcal{L}} R \max\{\nu_{Jac}^{(F)}, \nu_{Jac}^{(G)}\} \tag{26}$$

*Proof.* Using Lemma 1 of Song et al. [106] as discussed in Section 3, if we choose $R = \frac{\mu_{\text{Jac}}}{2\beta}$, where $\mu_{\text{Jac}} := \max\left\{\mu_{\text{Jac}}^{(F)}, \mu_{\text{Jac}}^{(G)}\right\}$ and $\beta := \min\left\{\beta_F, \beta_G\right\}$, then we see that for $\forall(\theta, \phi) \in \mathcal{B}((\theta_0, \phi_0), R)$, we have

$$\frac{\mu_{\text{Jac}}^{(F)}}{2} \leq \sigma_{\min}(\nabla_\theta F_\theta) \leq \sigma_{\max}(\nabla_\theta F_\theta) \leq \frac{3\nu_{\text{Jac}}^{(F)}}{2}$$

$$\frac{\mu_{\text{Jac}}^{(G)}}{2} \leq \sigma_{\min}(\nabla_\phi G_\phi) \leq \sigma_{\max}(\nabla_\phi G_\phi) \leq \frac{3\nu_{\text{Jac}}^{(G)}}{2}$$

And by Lemma 4 (Appendix C) of Song et al. [106], we see that both the mapping $F_\theta$ and $G_\phi$ are Lipschitz-continuous in the ball $\mathcal{B}((\theta_0, \phi_0), R)$. That is, $\forall(\theta, \phi), (\theta', \phi') \in \mathcal{B}((\theta_0, \phi_0), R)$, we have the following:

$$\|F_\theta - F_{\theta'}\| \leq \frac{3\nu_{\text{Jac}}^{(F)}}{2}\|\theta - \theta'\| \tag{27}$$

$$\|G_\phi - G_{\phi'}\| \leq \frac{3\nu_{\text{Jac}}^{(G)}}{2}\|\phi - \phi'\| \tag{28}$$

where $\sigma_{\max}(\nabla_\theta F_{\theta_0}) \leq \nu_{\text{Jac}}^{(F)}$ and $\sigma_{\max}(\nabla_\phi G_{\phi_0}) \leq \nu_{\text{Jac}}^{(G)}$.

Without loss of generality, we prove for the case of $F_\theta$ mapping as the same argument works for the $G_\phi$ mapping as well. We can observe the following for any $(\theta, \phi) \in \mathcal{B}((\theta_0, \phi_0), R)$:

$$\max_{\theta \in \mathcal{B}(\theta_0, R)} \|\nabla_{F_\theta}\mathcal{L}(F_\theta, G_\phi)\| \leq \|\nabla_{F_\theta}\mathcal{L}(F_{\theta_0}, G_\phi)\| + \max_{\theta' \in \mathcal{B}(\theta_0, R)} \|\nabla_{F_\theta}\mathcal{L}(F_{\theta'}, G_\phi) - \nabla_{F_\theta}\mathcal{L}(F_{\theta_0}, G_\phi)\|$$

$$\leq \|\nabla_{F_\theta}\mathcal{L}(F_{\theta_0}, G_\phi)\| + L_{\nabla\mathcal{L}} \cdot \max_{\theta' \in \mathcal{B}(\theta_0, R)} \|F_{\theta'} - F_{\theta_0}\|$$

$$\leq \|\nabla_{F_\theta}\mathcal{L}(F_{\theta_0}, G_\phi)\| + \frac{3L_{\nabla\mathcal{L}}\nu_{\text{Jac}}^{(F)}}{2} \max_{\theta' \in \mathcal{B}(\theta_0, R)} \|\theta_0 - \theta'\|$$

$$(\because \text{By Eq.}(27))$$

$$\leq \|\nabla_{F_\theta}\mathcal{L}(F_{\theta_0}, G_\phi)\| + \frac{3L_{\nabla\mathcal{L}}\nu_{\text{Jac}}^{(F)}R}{2}$$

$$= \|\nabla_{F_\theta}\mathcal{L}(F_{\theta_0}, G_\phi) - \underbrace{\nabla_{F_\theta}\mathcal{L}(F_{\theta^*}, G_{\phi^*})}_{=0}\| + \frac{3L_{\nabla\mathcal{L}}\nu_{\text{Jac}}^{(F)}R}{2} \tag{29}$$

where the last equality holds true because $(\theta^*, \phi^*)$ is a stationary point for the min-max objective $\mathcal{L}(F_\theta, G_\phi)$ and the Jacobian for $F$ is non-singular as $(\theta^*, \phi^*) \in \mathcal{B}((\theta_0, \phi_0), R)$ by assumption. That is,

$$0 = \nabla_\theta \mathcal{L}(F_{\theta^*}, G_{\phi^*}) = (\nabla_\theta F_{\theta^*})^\top \nabla_{F_\theta}\mathcal{L}(F_{\theta_0}, G_\phi) \tag{30}$$

$$\implies 0 = \|(\nabla_\theta F_{\theta^*})^\top \nabla_{F_\theta}\mathcal{L}(F_{\theta^*}, G_{\phi^*})\| = \|(\nabla_\theta F_{\theta^*})^\top\|\|\nabla_{F_\theta}\mathcal{L}(F_{\theta^*}, G_{\phi^*})\| \tag{31}$$

$$\implies 0 = \nabla_{F_\theta}\mathcal{L}(F_{\theta^*}, G_{\phi^*}) \quad (\because \nabla_\theta F_{\theta^*} \text{ non-singular}) \tag{32}$$

Thus, continuing from Equation (29), we have that that for any $(\theta, \phi) \in \mathcal{B}((\theta_0, \phi_0), R)$:

$$\max_{\theta \in \mathcal{B}(\theta_0, R)} \|\nabla_{F_\theta}\mathcal{L}(F_\theta, G_\phi)\| \leq \|\nabla_{F_\theta}\mathcal{L}(F_{\theta_0}, G_\phi) - \nabla_{F_\theta}\mathcal{L}(F_{\theta^*}, G_{\phi^*})\| + \frac{3L_{\nabla\mathcal{L}}\nu_{\text{Jac}}^{(F)}R}{2}$$

$$\leq L_{\nabla\mathcal{L}}(\|F_{\theta_0} - F_{\theta^*}\| + \|G_{\phi_0} - G_{\phi^*}\|) + \frac{3L_{\nabla\mathcal{L}}\nu_{\text{Jac}}^{(F)}R}{2}$$

$$(\because \mathcal{L} \text{ is } L_{\nabla\mathcal{L}}\text{-smooth}) \tag{33}$$

$$\leq \frac{3L_{\nabla\mathcal{L}}\nu_{\text{Jac}}^{(F)}}{2}\|\theta_0 - \theta^*\| + \frac{3L_{\nabla\mathcal{L}}\nu_{\text{Jac}}^{(G)}}{2}\|\phi_0 - \phi^*\| + \frac{3L_{\nabla\mathcal{L}}\nu_{\text{Jac}}^{(F)}R}{2}$$

$$(\because \text{By Equations } (27), (28)) \tag{34}$$

$$\leq 3L_{\nabla\mathcal{L}}\max\{\nu_{\text{Jac}}^{(F)}, \nu_{\text{Jac}}^{(G)}\}R + \frac{3L_{\nabla\mathcal{L}}\max\{\nu_{\text{Jac}}^{(F)}, \nu_{\text{Jac}}^{(G)}\}R}{2}$$

$$= 4.5L_{\nabla\mathcal{L}}\max\{\nu_{\text{Jac}}^{(F)}, \nu_{\text{Jac}}^{(G)}\}R \tag{35}$$

$$\square$$

To establish convergence guarantees under hidden convex–concave structure, it is essential to demonstrate that sufficient overparameterization yields a favorable initialization. Specifically, we show that: (i) the initialization lies close to a saddle point (in terms of gradient norm), and (ii) the optimization path remains within a region where the neural Jacobians have well-conditioned singular spectra. The latter will be ensured via a path length argument.

The following lemma provides a key component in this direction: it connects the initial value of the Lyapunov potential used in AltGDA (a weighted version of the Nash gap) to the gradient norms of both neural players. In subsequent lemmas, we will show that with high probability, and under sufficient width, this leads to the iterates remaining inside a well-conditioned-Jacobians' manifold that preserves PŁ-condition throughout training.

**Lemma E.4** (Upper Bound on Initial Potential $P_0$; Lemma 3.3 in Main Text). *Let $F_\theta$ and $G_\phi$ be neural network mappings such that they are $\beta_F$ and $\beta_G$ smooth as defined in Definition 2.3. Now let $(\theta_0, \phi_0)$ be such that Jacobian singular values for both the networks are strictly positive and bounded from above and below, $\mu_{Jac}^{(F)} \leq \sigma(\nabla_\theta F_{\theta_0}) \leq \nu_{Jac}^{(F)}$ and $\mu_{Jac}^{(G)} \leq \sigma(\nabla_\phi G_{\phi_0}) \leq \nu_{Jac}^{(G)}$. Suppose the min-max objective $\mathcal{L}(\theta, \phi)$ is $(\mu_\theta, \mu_\phi)$-HSCSC. Then the initial Lyapunov potential $P_0$ can be bounded from above as:*

$$
\begin{aligned}
P_0 \leq{} & \|(\nabla_{F_\theta}\mathcal{L}(F_{\theta_0}, G_{\phi_0}))^\top\| \cdot \nu_{Jac}^{(F)} \cdot \frac{1}{\mu_\theta(\mu_{Jac}^{(F)})^2} \cdot \|\nabla_\theta\mathcal{L}(\theta_0, \phi_0)\| \\
& + \|(\nabla_{F_\theta}\mathcal{L}(F_{\theta_0}, G_{\phi_0}))^\top\| \cdot \beta_F \frac{1}{2\mu_\theta^2(\mu_{Jac}^{(F)})^4} \cdot \|\nabla_\theta\mathcal{L}(\theta_0, \phi_0)\|^2 \\
& + (1+\lambda) \cdot \|(\nabla_{G_\phi}\mathcal{L}(F_{\theta_0}, G_{\phi_0}))^\top\| \cdot \nu_{Jac}^{(G)} \cdot \frac{1}{\mu_\phi(\mu_{Jac}^{(G)})^2} \cdot \|\nabla_\phi\mathcal{L}(\theta_0, \phi_0)\| \\
& + (1+\lambda) \cdot \|(\nabla_{G_\phi}\mathcal{L}(F_{\theta_0}, G_{\phi_0}))^\top\| \cdot \beta_G \frac{1}{2\mu_\phi^2(\mu_{Jac}^{(G)})^4} \cdot \|\nabla_\phi\mathcal{L}(\theta_0, \phi_0)\|^2
\end{aligned}
\tag{36}
$$

*Proof.* Using Lemma 1 of Song et al. [106] as discussed in Section 3, if we choose $R = \frac{\mu_{Jac}}{2\beta}$, where $\mu_{Jac} := \max\left\{\mu_{Jac}^{(F)}, \mu_{Jac}^{(G)}\right\}$ and $\beta := \min\{\beta_F, \beta_G\}$, then we see that for $\forall (\theta, \phi) \in \mathcal{B}((\theta_0, \phi_0), R)$, we have

$$
\frac{\mu_{Jac}^{(F)}}{2} \leq \sigma_{\min}(\nabla_\theta F_\theta) \leq \sigma_{\max}(\nabla_\theta F_\theta) \leq \frac{3\nu_{Jac}^{(F)}}{2}
$$
$$
\frac{\mu_{Jac}^{(G)}}{2} \leq \sigma_{\min}(\nabla_\phi G_\phi) \leq \sigma_{\max}(\nabla_\phi G_\phi) \leq \frac{3\nu_{Jac}^{(G)}}{2}
$$

And by Lemma 4 (Appendix C) of Song et al. [106], we see that both the mapping $F_\theta$ and $G_\phi$ are Lipschitz-continuous in the ball $\mathcal{B}((\theta_0, \phi_0), R)$. That is, $\forall (\theta, \phi), (\theta', \phi') \in \mathcal{B}((\theta_0, \phi_0), R)$, we have the following:

$$
\|F_\theta - F_{\theta'}\| \leq \frac{3\nu_{Jac}^{(F)}}{2}\|\theta - \theta'\|
\tag{37}
$$

$$
\|G_\phi - G_{\phi'}\| \leq \frac{3\nu_{Jac}^{(G)}}{2}\|\phi - \phi'\|
\tag{38}
$$

where $\sigma_{\max}(\nabla_\theta F_{\theta_0}) \leq \nu_{Jac}^{(F)}$ and $\sigma_{\max}(\nabla_\phi G_{\phi_0}) \leq \nu_{Jac}^{(G)}$.

Since $\mathcal{L}$ is $(\mu_\theta, \mu_\phi)$-HSCSC, it satisfies 2-sided PŁ-condition with PŁ-moduli as per Fact 2.6. Thus, we can obtain the following bound for $W_0$:

$$
W_0 = \mathcal{L}(\theta_0, \phi^*(\theta_0)) - \mathcal{L}(\theta_0, \phi_0)
\tag{39}
$$
$$
\leq \left(\nabla_{G_\phi}\mathcal{L}(F_{\theta_0}, G_{\phi_0})\right)^\top \left(G_{\phi^*(\theta_0)} - G(\phi_0)\right) \ (\because \text{hidden concavity})
\tag{40}
$$
$$
\leq \|\left(\nabla_{G_\phi}\mathcal{L}(F_{\theta_0}, G_{\phi_0})\right)^\top\| \cdot \|G_{\phi^*(\theta_0)} - G_{\phi_0}\|
\tag{41}
$$
$$
\leq \|\left(\nabla_{G_\phi}\mathcal{L}(F_{\theta_0}, G_{\phi_0})\right)^\top\| \cdot \left(\|\nabla_\phi G_{\phi_0}\|\|\phi^*(\theta_0) - \phi_0\| + \frac{\beta_G}{2}\|\phi^*(\theta_0) - \phi_0\|^2\right)
\tag{42}
$$

$(\because G_\phi$ is $\beta_G$-smooth$)$

$$\leq \|\left(\nabla_{G_\phi}\mathcal{L}(F_{\theta_0}, G_{\phi_0})\right)^\top\| \cdot \left(\nu_{\text{Jac}}^{(G)} \cdot \|\phi^*(\theta_0) - \phi_0\| + \frac{\beta_G}{2}\|\phi^*(\theta_0) - \phi_0\|^2\right) \tag{43}$$

$(\because \sigma(\nabla_\theta F_{\theta_0}) \leq \nu_{\text{Jac}}^{(F)})$

$$\leq \|\left(\nabla_{G_\phi}\mathcal{L}(F_{\theta_0}, G_{\phi_0})\right)^\top\| \cdot \left(\nu_{\text{Jac}}^{(G)} \cdot \sqrt{\frac{2}{\mu_\phi(\mu_{\text{Jac}}^{(G)})^2}} \cdot \sqrt{\mathcal{L}(\theta_0, \phi^*(\theta_0)) - \mathcal{L}(\theta_0, \phi_0)}\right.$$

$$\left.+ \frac{\beta_G}{\mu_\phi(\mu_{\text{Jac}}^{(G)})^2}(\mathcal{L}(\theta_0, \phi^*(\theta_0)) - \mathcal{L}(\theta_0, \phi_0))\right) \tag{44}$$

$(\because$ quadratic growth condition and Fact 2.6$)$

$$\implies W_0 \leq \|\left(\nabla_{G_\phi}\mathcal{L}(F_{\theta_0}, G_{\phi_0})\right)^\top\| \cdot \nu_{\text{Jac}}^{(G)} \cdot \frac{1}{\mu_\phi(\mu_{\text{Jac}}^{(G)})^2} \cdot \|\nabla_\phi \mathcal{L}(\theta_0, \phi_0)\|$$

$$+ \|\left(\nabla_{G_\phi}\mathcal{L}(F_{\theta_0}, G_{\phi_0})\right)^\top\| \cdot \beta_G \frac{1}{2\mu_\phi^2(\mu_{\text{Jac}}^{(G)})^4} \cdot \|\nabla_\phi \mathcal{L}(\theta_0, \phi_0)\|^2 \tag{45}$$

(using PŁ-condition and Fact 2.6)

Similarly, for $U_0$, we can say that:

$$U_0 = \mathcal{L}(\theta_0, \phi^*(\theta_0)) - \mathcal{L}(\theta^*, \phi^*) \tag{46}$$

$$\leq \mathcal{L}(\theta_0, \phi^*(\theta_0)) - \mathcal{L}(\theta^*, \phi_0) \quad (\because \mathcal{L}(\theta^*, \phi) \leq \mathcal{L}(\theta^*, \phi^*)\ \forall \phi) \tag{47}$$

$$\leq \mathcal{L}(\theta_0, \phi^*(\theta_0)) - \mathcal{L}(\theta^*(\phi_0), \phi_0) \quad (\because \mathcal{L}(\theta^*, \phi_0) \geq \mathcal{L}(\theta^*(\phi_0), \phi_0)) \tag{48}$$

$$= \underbrace{\mathcal{L}(\theta_0, \phi^*(\theta_0)) - \mathcal{L}(\theta_0, \phi_0)}_{\text{①}} + \underbrace{\mathcal{L}(\theta_0, \phi_0) - \mathcal{L}(\theta^*(\phi_0), \phi_0)}_{\text{②}} \tag{49}$$

Noticing that ① is exactly equal to $W_0$ and following similar arguments we used for $W_0$ to get a upper bound for ② but with hidden convexity instead, we obtain the following for $U_0$:

$$U_0 \leq \|\left(\nabla_{G_\phi}\mathcal{L}(F_{\theta_0}, G_{\phi_0})\right)^\top\| \cdot \nu_{\text{Jac}}^{(G)} \cdot \frac{1}{\mu_\phi(\mu_{\text{Jac}}^{(G)})^2} \cdot \|\nabla_\phi \mathcal{L}(\theta_0, \phi_0)\|$$

$$+ \|\left(\nabla_{G_\phi}\mathcal{L}(F_{\theta_0}, G_{\phi_0})\right)^\top\| \cdot \beta_G \frac{1}{2\mu_\phi^2(\mu_{\text{Jac}}^{(G)})^4} \cdot \|\nabla_\phi \mathcal{L}(\theta_0, \phi_0)\|^2 \tag{50}$$

$$+ \|\left(\nabla_{F_\theta}\mathcal{L}(F_{\theta_0}, G_{\phi_0})\right)^\top\| \cdot \nu_{\text{Jac}}^{(F)} \cdot \frac{1}{\mu_\theta(\mu_{\text{Jac}}^{(F)})^2} \cdot \|\nabla_\theta \mathcal{L}(\theta_0, \phi_0)\|$$

$$+ \|\left(\nabla_{F_\theta}\mathcal{L}(F_{\theta_0}, G_{\phi_0})\right)^\top\| \cdot \beta_F \frac{1}{2\mu_\theta^2(\mu_{\text{Jac}}^{(F)})^4} \cdot \|\nabla_\theta \mathcal{L}(\theta_0, \phi_0)\|^2$$

By combining the upper bounds obtained for $U_0$ and $W_0$, we get the following upper bound on the initial potential $P_0$:

$$P_0 \leq \|\left(\nabla_{F_\theta}\mathcal{L}(F_{\theta_0}, G_{\phi_0})\right)^\top\| \cdot \nu_{\text{Jac}}^{(F)} \cdot \frac{1}{\mu_\theta(\mu_{\text{Jac}}^{(F)})^2} \cdot \|\nabla_\theta \mathcal{L}(\theta_0, \phi_0)\|$$

$$+ \|\left(\nabla_{F_\theta}\mathcal{L}(F_{\theta_0}, G_{\phi_0})\right)^\top\| \cdot \beta_F \frac{1}{2\mu_\theta^2(\mu_{\text{Jac}}^{(F)})^4} \cdot \|\nabla_\theta \mathcal{L}(\theta_0, \phi_0)\|^2$$

$$+ (1+\lambda) \cdot \|\left(\nabla_{G_\phi}\mathcal{L}(F_{\theta_0}, G_{\phi_0})\right)^\top\| \cdot \nu_{\text{Jac}}^{(G)} \cdot \frac{1}{\mu_\phi(\mu_{\text{Jac}}^{(G)})^2} \cdot \|\nabla_\phi \mathcal{L}(\theta_0, \phi_0)\|$$

$$+ (1+\lambda) \cdot \|\left(\nabla_{G_\phi}\mathcal{L}(F_{\theta_0}, G_{\phi_0})\right)^\top\| \cdot \beta_G \frac{1}{2\mu_\phi^2(\mu_{\text{Jac}}^{(G)})^4} \cdot \|\nabla_\phi \mathcal{L}(\theta_0, \phi_0)\|^2$$

$\square$

**Remark E.5** (Path length bound for AltGDA). We know from Lemma 3.2 that the AltGDA path length satisfies:

$$\ell(T) \triangleq \sum_{t=0}^{T-1} \left( \|\theta_{t+1} - \theta_t\| + \|\phi_{t+1} - \phi_t\| \right) \leq \frac{\sqrt{2\alpha_1}}{1 - \sqrt{c}} \cdot \sqrt{P_0} \tag{51}$$

$$\leq \sqrt{2\alpha} \frac{1 + \sqrt{c}}{1 - c} \sqrt{P_0} < \frac{2\sqrt{2\alpha_1}}{1 - c} \cdot \sqrt{P_0} \ \ (\because c \in (0,1)) \tag{52}$$

where $\alpha_1$, $L$, and $c$ are defined as

$$\alpha_1 = \frac{2(1 + \eta_\phi^2 L_{\nabla\mathcal{L}}^2)\eta_\theta^2 L^2}{\mu_\theta} + \frac{20(1 + \eta_\phi^2 L_{\nabla\mathcal{L}}^2)\eta_\theta^2 L_{\nabla\mathcal{L}}^2 + 20L_{\nabla\mathcal{L}}^2 \eta_\phi^2}{\mu_\phi} \tag{53}$$

$$L = L_{\nabla\mathcal{L}} + \frac{L_{\nabla\mathcal{L}}^2}{\mu_\phi} \tag{54}$$

$$c = 1 - \frac{\mu_\theta \mu_\phi^2}{36(L_{\nabla\mathcal{L}})^3} \tag{55}$$

Now, say we define $T$ to be the first time instant when $(\theta_T, \phi_T) \notin \mathcal{B}((\theta_0, \phi_0), R)$ for an appropriate $R > 0$. And now, say, we start with appropriate $(\theta_0, \phi_0)$ such that $\ell(T) < R$. A sufficient condition to ensure this would be to find $(\theta_0, \phi_0)$ and a corresponding radius $R > 0$ such that

$$\frac{2\sqrt{2\alpha_1}}{1 - c}\sqrt{P_0} < R \tag{56}$$

$$\iff \frac{\alpha_1}{(1 - c)^2} P_0 < \frac{R^2}{8} \tag{57}$$

$$\iff \left( \frac{1296 L_{\nabla\mathcal{L}}^6}{\mu_\theta^2 \mu_\phi^4} \cdot \alpha_1 \right) \cdot P_0 < \frac{R^2}{8} \tag{58}$$

**Remark E.6** (Simplification of $\alpha_1$ for AltGDA Path Length Bound). For the case of $(\mu_\theta, \mu_\phi)$-HSCSC and $L_{\nabla\mathcal{L}}$-smooth (w.r.t. $(\theta, \phi)$) min-max objective function $\mathcal{L}$ and for learning rates $\eta_\theta = \frac{c_\theta \mu_\phi^2 \sigma_{\min}^4(\nabla_\phi G_{\phi_0})}{18 L_{\nabla\mathcal{L}}^3}$ and $\eta_\phi = \frac{c_\phi}{L_{\nabla\mathcal{L}}}$ with $0 < c_\theta, c_\phi \leq 1$, we can simplify $\alpha_1$ defined above in Equation (53) for AltGDA path length bound as follows:

$$\alpha_1 = \frac{2(1 + \eta_\phi^2 L_{\nabla\mathcal{L}}^2)\eta_\theta^2 L^2}{\mu_\theta \sigma_{\min}^2(\nabla_\theta F_{\theta_0})} + \frac{20(1 + \eta_\phi^2 L_{\nabla\mathcal{L}}^2)\eta_\theta^2 L_{\nabla\mathcal{L}}^2 + 20L_{\nabla\mathcal{L}}^2 \eta_\phi^2}{\mu_\phi \sigma_{\min}^2(\nabla_\phi G_{\phi_0})} \tag{59}$$

$$\lesssim \frac{2c_\phi^2 c_\theta^2 \mu_\phi^2 \sigma_{\min}^4(\nabla_\phi G_{\phi_0})}{(18)^2 L_{\nabla\mathcal{L}}^2 \mu_\theta \sigma_{\min}^2(\nabla_\theta F_{\theta_0})} + \frac{20c_\phi^2 c_\theta^2 \mu_\phi^3 \sigma_{\min}^6(\nabla_\phi G_{\phi_0})}{18^2 L_{\nabla\mathcal{L}}^4} \tag{60}$$

In fact, $\frac{1296 L_{\nabla\mathcal{L}}^6}{\mu_\theta^2 \mu_\phi^4} \cdot \alpha_1$ simplifies to the following:

$$\frac{1296 L_{\nabla\mathcal{L}}^6}{\mu_\theta^2 \mu_\phi^4} \cdot \alpha_1 = \frac{8 L_{\nabla\mathcal{L}}^4 c_\theta^2 c_\phi^2}{\mu_\theta^3 \mu_\phi^2 \sigma_{\min}^6(\nabla_\theta F_{\theta_0})\sigma_{\min}^4(\nabla_\phi G_{\phi_0})}$$

$$+ \frac{80 c_\theta^2 c_\phi^2 L_{\nabla\mathcal{L}}^2}{\mu_\theta^2 \mu_\phi \sigma_{\min}^4(\nabla_\theta F_{\theta_0})\sigma_{\min}^2(\nabla_\phi G_{\phi_0})} \tag{61}$$

**Lemma E.7** (Simplified AltGDA path length bound condition). *Consider the premise as defined in Lemma 3.2 for $(\mu_\theta, \mu_\phi)$-HSCSC and $L_{\nabla\mathcal{L}}$-smooth min-max objective with learning rates $\eta_\theta = c_\theta \mu_\phi^2 / 18 L_{\nabla\mathcal{L}}^2$ and $\eta_\phi = c_\phi / L_{\nabla\mathcal{L}}$ with $c_\theta, c_\phi \in (0, 1]$ along with the premise of Lemma E.4. Let $T$ be the first time instant when $(\theta_T, \phi_T) \notin \mathcal{B}((\theta_0, \phi_0), R)$ for an appropriate $R > 0$. Then to ensure $(\theta_t, \phi_t) \in \mathcal{B}((\theta_0, \phi_0), R) \ \forall t \leq T$ as discussed in Remark E.5 (Equation (58)), the following is a sufficient condition:*

$$\|\nabla_\theta \mathcal{L}(F_{\theta_0}, G_{\phi_0})\| + \|\nabla_\phi \mathcal{L}(F_{\theta_0}, G_{\phi_0})\| \lesssim \frac{R^2}{8} \tag{62}$$

*given that $c_\theta$ and $c_\phi$ are chosen such that:*

$$c_\theta = \sqrt{\frac{1}{4 \cdot A_\theta \cdot B_\phi}} \quad and \quad c_\phi = 1 \tag{63}$$

*where*

$$A_\theta = \Bigg( (1 + \|\nabla_{F_\theta}\mathcal{L}(F_{\theta_0}, G_{\phi_0})\|) \cdot \Big(1 + 8L_{\nabla\mathcal{L}}^4 \nu_{Jac}^{(F)}\Big) \cdot \Bigg(1 + \frac{1}{\mu_\theta^3(\mu_{Jac}^{(F)})^6}\Bigg) \cdot (1 + \|\nabla_\theta\mathcal{L}(F_{\theta_0}, G_{\phi_0})\|)$$

$$\cdot \Big(1 + 80L_{\nabla\mathcal{L}}^2 \nu_{Jac}^{(F)}\Big) \cdot \Bigg(1 + \frac{1}{\mu_\theta^3(\mu_{Jac}^{(F)})^4}\Bigg) \cdot \Big(1 + 8L_{\nabla\mathcal{L}}^4 \beta_F\Big) \cdot \Bigg(1 + \frac{1}{2\mu_\theta^5(\mu_{Jac}^{(F)})^{10}}\Bigg) \cdot \Big(1 + 80L_{\nabla\mathcal{L}}^2 \beta_F\Big)$$

$$\cdot \Bigg(1 + \frac{1}{2\mu_\theta^4(\mu_{Jac}^{(F)})^8}\Bigg) \cdot \Bigg(1 + \frac{1}{\mu_\theta^2(\mu_{Jac}^{(F)})^4}\Bigg) \cdot \Bigg(1 + \frac{1}{2\mu_\theta^3(\mu_{Jac}^{(F)})^6}\Bigg) \Bigg)$$

*and*

$$B_\phi = \Bigg( (1 + \|\nabla_{G_\phi}\mathcal{L}(F_{\theta_0}, G_{\phi_0})\|) \cdot \Bigg(1 + \frac{1}{\mu_\phi^3(\mu_{Jac}^{(G)})^6}\Bigg) \cdot \Bigg(1 + \frac{1}{\mu_\phi(\mu_{Jac}^{(G)})^4}\Bigg) \cdot \Bigg(1 + \frac{1}{\mu_\phi^2(\mu_{Jac}^{(G)})^4}\Bigg)$$

$$\cdot \Bigg(1 + \frac{1}{\mu_\phi(\mu_{Jac}^{(G)})^2}\Bigg) \cdot (1 + \lambda) \cdot (1 + \|\nabla_\phi\mathcal{L}(F_{\theta_0}, G_{\phi_0})\|) \cdot \Big(1 + 80L_{\nabla\mathcal{L}}^2 \nu_{Jac}^{(G)}\Big) \cdot \Big(1 + 8L_{\nabla\mathcal{L}}^4 \beta_G\Big)$$

$$\cdot \Bigg(1 + \frac{1}{\mu_\phi^4(\mu_{Jac}^{(G)})^8}\Bigg) \cdot \Big(1 + 80L_{\nabla\mathcal{L}}^2 \beta_G\Big) \cdot \Bigg(1 + \frac{1}{\mu_\phi^3(\mu_{Jac}^{(G)})^8}\Bigg) \Bigg)$$

*Proof.* By choosing $c_\theta$ and $c_\phi$ as specified above and using observations from Remark E.6 (Equation (61)), a sufficient condition to ensure the iterates never leave as derived in Remark E.5 (Equation (58)) can be further simplified as follows:

$$\Bigg( \frac{1296L_{\nabla\mathcal{L}}^6}{\mu_\theta^2\mu_\phi^4} \cdot \alpha_1 \Bigg) \cdot P_0 < \frac{R^2}{8}$$

$$\therefore \Bigg( \frac{8L_{\nabla\mathcal{L}}^4 c_\theta^2 c_\phi^2}{\mu_\theta^3\mu_\phi^2\sigma_{\min}^6(\nabla_\theta F_{\theta_0})\sigma_{\min}^4(\nabla_\phi G_{\phi_0})} + \frac{80c_\theta^2 c_\phi^2 L_{\nabla\mathcal{L}}^2}{\mu_\theta^2\mu_\phi\sigma_{\min}^4(\nabla_\theta F_{\theta_0})\sigma_{\min}^2(\nabla_\phi G_{\phi_0})} \Bigg) \cdot P_0 < \frac{R^2}{8}$$

$$\therefore T_1\|\nabla_\theta\mathcal{L}(\theta_0, \phi_0)\| + T_2\|\nabla_\theta\mathcal{L}(\theta_0, \phi_0)\|^2 + T_3\|\nabla_\phi\mathcal{L}(\theta_0, \phi_0)\| + T_4\|\nabla_\phi\mathcal{L}(\theta_0, \phi_0)\|^2 < \frac{R^2}{8}$$

where $T_j$'s are coefficients of the gradient norms as per Equations (61) and (58). Notice that both $T_2$ and $T_4$ contain $c_\theta^2$ and as per our choice of $c_\theta$, $c_\theta^2$ contains $\frac{1}{1+\|\nabla_\theta\mathcal{L}(F_{\theta_0},G_{\phi_0})\|}$ and $\frac{1}{1+\|\nabla_\phi\mathcal{L}(F_{\theta_0},G_{\phi_0})\|}$ terms which can help absorb one of the powers of the gradient norms $\|\nabla_\theta\mathcal{L}(F_{\theta_0}, G_{\phi_0})\|$ or $\|\nabla_\phi\mathcal{L}(F_{\theta_0}, G_{\phi_0})\|$ in terms corresponding to $T_2$ and $T_4$. This will help constructing (as shown below) a simpler sufficient condition that would require only the norm of the loss' gradient w.r.t. parameters $\theta, \phi$ to be small. Furthermore, by construction, $c_\theta \leq 1/2, c_\phi = 1$. Therefore, we have that $T_i \leq 1 \; \forall i \in [4]$. Combining all of the above, what we require for ensuring iterates never leave the ball of radius $R$ is as follows:

$$\|\nabla_\theta\mathcal{L}(F_{\theta_0}, G_{\phi_0})\| + \|\nabla_\phi\mathcal{L}(F_{\theta_0}, G_{\phi_0})\| \lesssim \frac{R^2}{8} \tag{64}$$

$\square$

**Remark E.8** (Local-smoothness constant). If the iterates $(\theta_t, \phi_t) \; \forall t$ stay within the ball $\mathcal{B}((\theta_0, \phi_0), R)$, by invoking H.3, we can use the local-Lipschitz constant derived Lemma E.3 along with the smoothness of the neural network maps $(\beta_F, \beta_G)$ to obtain the exact value of the smoothness constant for the HSCSC and smooth loss as $L_{\nabla\mathcal{L}} = L_\mathcal{L}^{\text{act}} \max\{\beta_F, \beta_G\}$.

# F Proofs for Input-Optimization Min-Max Games

> The following result characterizes the lower and upper bounds on the singular values of the Jacobian for the case of Input-Optimization Min-Max Games. In particular, we bring out the dependence of these lower and upper bounds on the size of the hidden layer, $d_1^{(F)}$ and $d_1^{(G)}$, and variances for random Gaussian initializations of the neural network layers, $\{\sigma_k^{(F)}\}_{k \in [2]}$ and $\{\sigma_k^{(G)}\}_{k \in [2]}$, for both the players $F$ and $G$. Computing the lower bound on singular values is important as it's used in defining the radius around the initial parameters to ensure the neural network Jacobian remains non-singular within the entire ball. Moreover, these lower and upper bounds will be helpful in determining a set of sufficient conditions on the variances for ensuring the AltGDA trajectory reaches the saddle point.

**Lemma F.1** (Lemma 3.4 in Main Paper). *Consider a neural network $F$ with parameters $\theta \in \mathbb{R}^{d_0^{(F)}}$ defined as $F(\theta) = W_2^{(F)} \psi(W_1^{(F)} \theta)$ where $W_k^{(F)} \in \mathbb{R}^{d_k^{(F)} \times d_{k-1}^{(F)}}$ ($k \in \{1,2\}$), $(W_1^{(F)})_{i,j} \sim \mathcal{N}(0, (\sigma_1^{(F)})^2) \; \forall i, j$, $(W_2^{(F)})_{k,l} \sim \mathcal{N}(0, (\sigma_2^{(F)})^2) \; \forall k, l$ and $\psi$ is the GeLU activation function with $d_1^{(F)} \geq 256 d_0^{(F)}$, $d_1^{(F)} \geq 256 d_2^{(F)}$ and $(\sigma_1^{(F)})^2 < \frac{\pi}{4Cd_1^{(F)}\|\theta\|^2}$. Then the minimum singular value of the Jacobian $\nabla_\theta F_\theta$ is lower bounded as*

$$\sigma_{\min}(\nabla_\theta F_\theta) > \frac{\sigma_1^{(F)} \sigma_2^{(F)} d_1^{(F)}}{16} \cdot \left( \frac{1}{2} - \sigma_1^{(F)}\|\theta\|\sqrt{\frac{Cd_1^{(F)}}{\pi}} \right) \tag{65}$$

*w.p $\geq 1 - 2e^{-\frac{d_1^{(F)}}{64}} - e^{-Cd_1^{(F)}}$. The maximum singular value of $\nabla_\theta F_\theta$ is upper bounded as*

$$\sigma_{\max}(\nabla_\theta F_\theta) < 3.47 \sigma_1^{(F)} \sigma_2^{(F)} d_1^{(F)} \tag{66}$$

*w.p. $\geq 1 - 2e^{-\frac{d_1^{(F)}}{64}}$.*

*Proof.* By Theorem 4.6.1 in [110], we get w.p. $\geq 1 - 2e^{-t^2}$:

$$\sigma_1\left(\sqrt{d_1^{(F)}} - 4\left(\sqrt{d_0^{(F)}} + t\right)\right) \leq \sigma_{\min}(W_1^{(F)}) \leq \sigma_{\max}(W_1^{(F)}) \leq \sigma_1^{(F)}\left(\sqrt{d_1^{(F)}} + 4\left(\sqrt{d_0^{(F)}} + t\right)\right). \tag{67}$$

By choosing $t = \frac{1}{8}\sqrt{d_1^{(F)}}$, we get w.p. $\geq 1 - 2e^{-d_1^{(F)}/64}$

$$\sigma_1^{(F)}\left(\frac{1}{2}\sqrt{d_1^{(F)}} - 4\sqrt{d_0^{(F)}}\right) \leq \sigma_{\min}(W_1^{(F)}) \leq \sigma_{\max}(W_1^{(F)}) \leq \sigma_1^{(F)}\left(\frac{3}{2}\sqrt{d_1^{(F)}} + 4\sqrt{d_0^{(F)}}\right). \tag{68}$$

If we set $d_1^{(F)} \geq 256 d_0^{(F)}$, we get

$$\left(\frac{1}{2}\sqrt{d_1^{(F)}} - 4\sqrt{d_0^{(F)}}\right) \geq \frac{1}{4}\sqrt{d_1^{(F)}} \tag{69}$$

and

$$\frac{3}{2}\sqrt{d_1^{(F)}} + 4\sqrt{d_0^{(F)}} \leq \frac{7}{4}\sqrt{d_1^{(F)}}. \tag{70}$$

Therefore, we get that w.p. $\geq 1 - 2e^{-\frac{d_1^{(F)}}{64}}$,

$$\frac{1}{4}\sigma_1^{(F)}\sqrt{d_1^{(F)}} \leq \sigma_{\min}(W_1^{(F)}) \leq \sigma_{\max}(W_1^{(F)}) \leq \frac{7}{4}\sigma_1^{(F)}\sqrt{d_1^{(F)}}. \tag{71}$$

For the case of $W_2^{(F)}$, because we have $d_2^{(F)} < d_1^{(F)}$, we will use analogous reasoning as that for $W_1^{(F)}$ above for $(W_2^{(F)})^\top$ with $t = \frac{1}{8}\sqrt{d_1^{(F)}}$ which yields w.p. $\geq 1 - 2e^{-\frac{d_1^{(F)}}{64}}$,

$$\sigma_2^{(F)}\left(\frac{1}{2}\sqrt{d_1^{(F)}} - 4\sqrt{d_2^{(F)}}\right) \leq \sigma_{\min}((W_2^{(F)})^\top) = \sigma_{\min}(W_2^{(F)}) \tag{72}$$

$$\leq \sigma_{\max}((W_2^{(F)})^\top) = \sigma_{\max}(W_2^{(F)}) \leq \sigma_2^{(F)}(\frac{3}{2}\sqrt{d_1^{(F)}} + 4\sqrt{d_2^{(F)}}) \tag{73}$$

Then, by setting $d_1^{(F)} \geq 256 d_2^{(F)}$, we get

$$\frac{1}{4}\sigma_2^{(F)}\sqrt{d_1^{(F)}} \leq \sigma_{\min}(W_2^{(F)}) \leq \sigma_{\max}(W_2^{(F)}) \leq \frac{7}{4}\sigma_2^{(F)}\sqrt{d_1^{(F)}}$$

Note that the Jacobian for the neural network $F(\theta)$ can be computed as

$$\nabla_\theta F_\theta = \frac{\partial F(\theta)}{\partial \theta} = W_2^{(F)} \frac{\partial}{\partial z}\psi(z)W_1^{(F)} = W_2^{(F)}\Psi(W_1^{(F)}\theta)W_1^{(F)} \tag{74}$$

where $z := W_1^{(F)}\theta$ and $\Psi(W_1^{(F)}\theta) = \text{diag}(\psi'((W_1^{(F)}\theta)_1), \dots, \psi'((W_1^{(F)}\theta)_{d_1^{(F)}}))$. Therefore, we can compute the minimum and maximum singular values for this Jacobian $\nabla_\theta F_\theta$ by looking at its operator norm: $\|\nabla_\theta F_\theta\| = \|W_2^{(F)}\Psi(W_1^{(F)}\theta)W_1^{(F)}\|$. Then, by properties of the operator norm, we see that

$$\|\nabla_\theta F_\theta\| \geq \sigma_{\min}(W_2^{(F)})\|\Psi(W_1^{(F)}\theta)\|\sigma_{\min}(W_1^{(F)}) \tag{75}$$

$$\|\nabla_\theta F_\theta\| \leq \sigma_{\max}(W_2^{(F)})\|\Psi(W_1^{(F)}\theta)\|\sigma_{\max}(W_1^{(F)}) \tag{76}$$

This further tells us that

$$\sigma_{\min}(\nabla_\theta F_\theta) \geq \sigma_{\min}(W_2^{(F)})\sigma_{\min}(\Psi(W_1^{(F)}\theta))\sigma_{\min}(W_1^{(F)}) \tag{77}$$

$$\sigma_{\max}(\nabla_\theta F_\theta) \leq \sigma_{\max}(W_2^{(F)})\sigma_{\max}(\Psi(W_1^{(F)}\theta))\sigma_{\max}(W_1^{(F)}) \tag{78}$$

We can further simplify these lower and upper bounds for the Jacobian singular values by noting that since $\Psi(W_1^{(F)}\theta)$ is a diagonal matrix, $\sigma_{\min}(\Psi(W_1^{(F)}\theta)) = \min_{1 \leq i \leq d_1^{(F)}} \psi'(z_i)$ and

$\sigma_{\max}(\Psi(W_1^{(F)}\theta)) = \max_{1 \leq i \leq d_1^{(F)}} \psi'(z_i)$ where $z_i = (W_1^{(F)}\theta)_i \ \forall i \in [d_1^{(F)}]$.

Now, notice that $z = W_1^{(F)}\theta \sim \mathcal{N}(0, (\sigma_1^{(F)})^2\|\theta\|_2^2 I_{d_1^{(F)} \times d_1^{(F)}})$. Using the fact that GeLU's derivative, $\psi'$, is $L_{\psi'}$-Lipschitz with $L_{\psi'} = \sup_{x \in \mathbb{R}} |\psi''(x)| = \varphi(0) = \frac{2}{\sqrt{\pi}}$ ($\varphi$ is standard normal PDF), we can appeal to concentration inequality for Lipschitz functions of Gaussian random variables and infer for $0 < \epsilon < 1/2$:

$$P[\sigma_{\min}(\Psi(W_1^{(F)}\theta)) > \epsilon] = P[\forall i : \psi'((W_1^{(F)}\theta)_i) > \epsilon] = (P[\psi'((W_1^{(F)}\theta)_1) > \epsilon])^{d_1^{(F)}} \tag{79}$$

$$= (P[\psi'((W_1^{(F)}\theta)_1) - 1/2 > \epsilon - 1/2])^{d_1^{(F)}} \tag{80}$$

$$= (P[\psi'((W_1^{(F)}\theta)_1) - E[\psi'((W_1^{(F)}\theta)_1)] > \epsilon - 1/2])^{d_1^{(F)}} \tag{81}$$

$$(\because E[\psi'((W_1^{(F)}\theta)_1)] = E[(W_1^{(F)}\theta)_1\varphi((W_1^{(F)}\theta)_1)] + E[\Phi((W_1^{(F)}\theta)_1)] = 0 + 1/2) \tag{82}$$

$$(\varphi, \Phi \text{ are standard normal PDF \& CDF, resp.}) \tag{83}$$

$$\geq 1 - e^{-\frac{(1/2-\epsilon)^2}{2L_{\psi'}^2(\sigma_1^{(F)})^2\|\theta\|^2}} \tag{84}$$

$$(\because P[\psi'(z) \leq 1/2 - t] \leq e^{-\frac{\pi t^2}{(\sigma_1^{(F)})^2\|\theta\|^2}}; \text{ choose } t = 1/2 - \epsilon) \tag{85}$$

$$= 1 - e^{-Cd_1^{(F)}} \text{ (if we set } \epsilon = 1/2 - (\sigma_1^{(F)}\|\theta\|\sqrt{Cd_1^{(F)}/\pi})) \tag{86}$$

Therefore, as long as we have $(\sigma_1^{(F)})^2 < \frac{\pi}{4Cd_1^{(F)}\|\theta\|^2}$, we have w.p. $\geq 1 - e^{-Cd_1^{(F)}}$ that

$$\sigma_{\min}(\Psi(W_1^{(F)}\theta)) > \frac{1}{2} - \sigma_1^{(F)}\|\theta\|\sqrt{\frac{Cd_1^{(F)}}{\pi}} \tag{87}$$

Combining all of this, we can say w.p. $\geq 1 - 2e^{-\frac{d_1^{(F)}}{64}} - e^{-Cd_1^{(F)}}$, we have:

$$\sigma_{\min}(\nabla_\theta F_\theta) > \frac{\sigma_1^{(F)}\sigma_2^{(F)}d_1^{(F)}}{16} \cdot \left(\frac{1}{2} - \sigma_1^{(F)}\|\theta\|\sqrt{\frac{Cd_1^{(F)}}{\pi}}\right). \tag{88}$$

And w.p. $\geq 1 - 2e^{-\frac{d_1^{(F)}}{64}}$, we have: $\sigma_{\max}(\nabla_\theta F_\theta) < \left(\frac{7\sqrt{d_1^{(F)}}}{4}\right)^2 \cdot 1.13\sigma_1^{(F)}\sigma_2^{(F)} < 3.47\sigma_1^{(F)}\sigma_2^{(F)}d_1^{(F)}$

where we used the fact that $\max_{x \in \mathbb{R}} \psi'(x) = \psi'(\sqrt{2}) \approx 1.1289$.  □

> Before we present the following lemma, we provide some intuition: In addition to the previous lemma for high probability lower and upper bounds on singular values of the neural network Jacobian, we also need to check whether the networks in input-optimization games are smooth or not. Towards that, the following result proves that it is so and provides the exact smoothness constant. Computing this smoothness constant is important for another reason: It is used in defining the radius around the initial parameters to ensure the neural network Jacobian remains non-singular within the entire ball.

**Lemma F.2** (Lemma 3.4 in Main Paper). *Consider a neural network $F$ with parameters $\theta \in \mathbb{R}^{d_0^{(F)}}$ as defined in Lemma F.1 above. Then, w.p. $\geq 1 - 2e^{-\frac{d_1^{(F)}}{64}}$ the neural network $F$ is $\beta_F$-smooth where $\beta_F = \frac{1}{\sqrt{2\pi}} \cdot \frac{343(\sigma_1^{(F)})^2\sigma_2^{(F)}(d_1^{(F)})^{3/2}}{32}$.*

*Proof.* In order to prove that $F$ is $\beta_F$-smooth, we need to show that Equation 2 holds true. For $\theta, \theta' \in \mathbb{R}^{d_0^{(F)}}$, we write

$$\|\nabla_\theta F_\theta - \nabla_\theta F_{\theta'}\| \leq \|W_2^{(F)}\|\|\Psi(W_1^{(F)}\theta) - \Psi(W_1^{(F)}\theta')\|\|W_1^{(F)}\| \tag{89}$$

$$\leq \|W_2^{(F)}\|L_{\psi'}\|W_1^{(F)}\theta - W_1^{(F)}\theta'\|\|W_1^{(F)}\| \quad (\because \psi' \text{ is } L_{\psi'} - \text{Lipschitz}) \tag{90}$$

$$= L_{\psi'}\|W_2^{(F)}\|\|W_1^{(F)}\|\|\theta - \theta'\|\|W_1^{(F)}\| \tag{91}$$

$$\implies \sigma_{\max}(\nabla_\theta F_\theta - \nabla_\theta F_{\theta'}) \leq \frac{2}{\sqrt{2\pi}}\sigma_{\max}(W_2^{(F)}) \cdot \sigma_{\max}^2(W_1^{(F)})\|\theta - \theta'\| \tag{92}$$

where $L_{\psi'} = \frac{2}{\sqrt{2\pi}}$ for derivative of GeLU activation function. Using Lemma F.1 results for singular values of the random matrices $W_1^{(F)}$ and $W_2^{(F)}$, we get that w.p. $\geq 1 - 2e^{-\frac{d_1^{(F)}}{64}}$

$$\sigma_{\max}(\nabla_\theta F_\theta - \nabla_\theta F_{\theta'}) \leq \frac{343(\sigma_1^{(F)})^2\sigma_2^{(F)}(d_1^{(F)})^{3/2}}{32\sqrt{2\pi}}\|\theta - \theta'\| \tag{93}$$

□

> The result in the main paper offers an average-case analysis under Gaussian sampling with variance scaled as $\text{poly}(1/d_1)$. Here, we provide a more detailed version that explicitly states the assumptions and technical conditions required to ensure convergence to equilibrium. While the high-level complexity perspective remains unchanged, we believe this finer analysis may be of independent interest, particularly for applications in adversarial attack design.

**Theorem F.3** (Theorem 3.5 in Main Paper). *Consider two neural networks $F, G$ with parameters $\theta \in \mathbb{R}^{d_0^{(F)}}$ and $\phi \in \mathbb{R}^{d_0^{(G)}}$, respectively, as defined in Lemma F.1 above. Then for the $\varepsilon$-regularized bilinear min-max objective $\mathcal{L}(\theta, \phi)$ as defined in Equation (3) with the neural networks $F$ and $G$ defined above, alternating gradient-descent-ascent with appropriate fixed learning rates $\eta_\theta, \eta_\phi$ (see Lemma E.7) reaches the desired saddle point w.p. $\geq 1 - 4e^{-\frac{d_1^{(F)}}{64}} - 4e^{-\frac{d_1^{(G)}}{64}} - e^{-Cd_1^{(F)}} - e^{-Cd_1^{(G)}}$ ($C$ are some universal constants) if the initial parameters $(\theta_0, \phi_0)$ and standard deviations $\sigma_k^{(F)}$ and $\sigma_k^{(G)}, k \in \{1, 2\}$ are chosen such that:*

$$(\sigma_1^{(F)})^2 < \frac{\pi}{4Cd_1^{(F)}\|\theta_0\|^2} \quad \& \quad (\sigma_1^{(G)})^2 < \frac{\pi}{4Cd_1^{(G)}\|\phi_0\|^2} \tag{94}$$

$$\frac{(\sigma_1^{(F)})^4 \cdot (\sigma_2^{(F)})^2 \cdot \|\theta_0\|}{\left(\frac{1}{2} - (\sigma_1^{(F)})\|\theta_0\|\sqrt{\frac{Cd_1^{(F)}}{\pi}}\right)^2} \lesssim \frac{\pi}{\varepsilon \cdot (d_1^{(F)})^{3.5}} \tag{95}$$

$$\frac{(\sigma_1^{(G)})^4 \cdot (\sigma_2^{(G)})^2 \cdot \|\phi_0\|}{\left(\frac{1}{2} - (\sigma_1^{(G)})\|\phi_0\|\sqrt{\frac{Cd_1^{(G)}}{\pi}}\right)^2} \lesssim \frac{\pi}{\varepsilon \cdot (d_1^{(G)})^{3.5}} \tag{96}$$

$$\frac{(\sigma_1^{(G)}\sigma_2^{(G)})((\sigma_1^{(F)})^3\sigma_2^{(F)})\|\theta_0\|}{\left(\frac{1}{2} - (\sigma_1^{(F)})\|\theta_0\|\sqrt{\frac{Cd_1^{(F)}}{\pi}}\right)^2} \lesssim \frac{\pi}{\sigma_{\max}(A)(d_1^{(F)})^{2.5}} \tag{97}$$

$$\frac{(\sigma_1^{(F)}\sigma_2^{(F)})((\sigma_1^{(G)})^3\sigma_2^{(G)})\|\phi_0\|}{\left(\frac{1}{2} - (\sigma_1^{(G)})\|\phi_0\|\sqrt{\frac{Cd_1^{(G)}}{\pi}}\right)^2} \lesssim \frac{\pi}{\sigma_{\max}(A)(d_1^{(G)})^{2.5}} \tag{98}$$

*Proof.* Since $\mathcal{L}(\theta, \phi)$ is $\varepsilon$-hidden-strongly-convex-strongly-concave, by Fact 2.6 it also satisfies the 2-sided PŁ-condition with $(\mu_\theta = \varepsilon \cdot \sigma_{\min}^2(\nabla_\theta F_\theta), \mu_\phi = \varepsilon \cdot \sigma_{\min}^2(\nabla_\phi G_\phi))$ w.p. $\geq 1 - 2e^{-\frac{d_1}{64}} - e^{-Cd_1}$, we can utilise path-length bound as derived in Lemma 3.2 which leaves us with controlling the potential $P_0$ for ensuring convergence to the saddle point. Thus, computing loss gradients (Equation (4)) and using the sufficient condition for ensuring iterates do not leave the ball $\mathcal{B}((\theta_0, \phi_0), R)$ (where $R = \frac{\max\{\mu_{\text{Jac}}^{(F)}, \mu_{\text{Jac}}^{(G)}\}}{\min\{\beta_F, \beta_G\}}$ as defined in Section 3) thus ensuring non-singular Jacobian inside the ball for both the neural networks (Lemma E.7), we require the following:

$$\left( \|F(\theta_0)\| \cdot (\varepsilon\sigma_{\max}(\nabla_\theta F_{\theta_0}) + \sigma_{\max}(\nabla_\phi G_{\phi_0})\sigma_{\max}(A)) \right.$$

$$\left. + \|G(\phi_0)\| \cdot (\sigma_{\max}(\nabla_\theta F_{\theta_0})\sigma_{\max}(A) + \varepsilon\sigma_{\max}(\nabla_\phi G_{\phi_0})) \right) \lesssim \frac{R^2}{8} \tag{99}$$

Thus, we want

$$\left( (3.47\varepsilon\sigma_1^{(F)}\sigma_2^{(F)}d_1^{(F)})\|F(\theta_0)\| + (3.47\sigma_1^{(G)}\sigma_2^{(G)}d_1^{(G)}\sigma_{\max}(A))\|F(\theta_0)\| \right.$$

$$\left. + (3.47\sigma_1^{(F)}\sigma_2^{(F)}d_1^{(F)}\sigma_{\max}(A)) \cdot \|G(\phi_0)\| + (3.47\varepsilon\sigma_1^{(G)}\sigma_2^{(G)}d_1^{(G)})\|G(\phi_0)\| \right) \lesssim \frac{R^2}{8} \tag{100}$$

$$(\because \sigma_{\max}(\nabla_\theta F_{\theta_0}) < 3.47\sigma_1^{(F)}\sigma_2^{(F)}d_1^{(F)}, \sigma_{\max}(\nabla_\phi G_{\phi_0}) < 3.47\sigma_1^{(G)}\sigma_2^{(G)}d_1^{(G)})$$

Therefore, we want the following to hold true:

1. $(3.47\varepsilon\sigma_1^{(F)}\sigma_2^{(F)}d_1^{(F)})\|F(\theta_0)\| \lesssim \frac{R^2}{32}$

2. $(3.47\sigma_{\max}(A)\sigma_1^{(G)}\sigma_2^{(G)}d_1^{(G)})\|F(\theta_0)\| \lesssim \frac{R^2}{32}$

3. $(3.47\sigma_{\max}(A)\sigma_1^{(F)}\sigma_2^{(F)}d_1^{(F)})\|G(\phi_0)\| \lesssim \frac{R^2}{32}$

4. $(3.47\varepsilon\sigma_1^{(G)}\sigma_2^{(G)}d_1^{(G)})\|G(\phi_0)\| \lesssim \frac{R^2}{32}$

Since $\|F(\theta_0)\| \leq \sigma_{\max}(W_2^{(F)})\sigma_{\max}(\psi(W_1^{(F)}\theta_0))$, we can use Lemma F.1 for maximum singular values of $W_2^{(F)}$ and the following calculation for upper bounding $\psi(W_1^{(F)}\theta)$:

$$\|\psi((W_1^{(F)}\theta))\| \leq \sqrt{d_1^{(F)}} \max_i |\psi((W_1^{(F)}\theta)_i)| \tag{101}$$

$$\implies P[\|\psi(W_1^{(F)}\theta)\| > t] \leq P[\sqrt{d_1^{(F)}} \max_i |(W_1^{(F)}\theta)_i| > t] \tag{102}$$

$$\text{Also, } P[\max_i |(W_1^{(F)}\theta)_i| > t] \le 2d_1 e^{-t^2/(2\sigma_{1,F}^2 \|\theta\|^2)} \tag{103}$$

$$(\because \psi(x) \le x \ \forall x, W_1^{(F)}\theta \sim \mathcal{N}(0, \sigma_{1,F}^2 \|\theta\|^2 I_{d_1^{(F)}})) \tag{104}$$

$$\implies P[\max_i |(W_1^{(F)}\theta)_i| > t] \le \delta \tag{105}$$

$$(\text{Set } \delta = 2d_1 e^{-t^2/(2\sigma_{1,F}^2 \|\theta_0\|^2)} = e^{-Cd_1}) \tag{106}$$

$$\implies P[\sqrt{d_1^{(F)}} \max_i |(W_1^{(F)}\theta)_i| \le C'\sigma_1^{(F)} \|\theta\| d_1^{(F)}] > 1 - e^{-Cd_1} \tag{107}$$

Thus, for ensuring 1.) holds, we will demand the following: $(3.47\varepsilon\sigma_1^{(F)}\sigma_2^{(F)}d_1^{(F)}) \cdot \left(\frac{7}{4}\sigma_2^{(F)}\sqrt{d_1^{(F)}}\right) \cdot \left(C'\sigma_1^{(F)}\|\theta_0\|d_1^{(F)}\right) \lesssim \frac{R^2}{32}$. This will yield the following sufficient condition on $\sigma_1^{(F)}$ and $\sigma_2^{(F)}$:

$$(\sigma_1^{(F)})^2 \cdot (\sigma_2^{(F)})^2 \lesssim \frac{R^2}{195\varepsilon C'\|\theta_0\|(d_1^{(F)})^{2.5}} \tag{108}$$

By definition the radius $R \ge \frac{\mu_{\text{Jac}}^{(F)}}{2\beta_F}$. Substituting the lower bound for minimum singular value and Lipschitzness constant for Jacobian of network $F$ from Lemmas F.1-F.2, we obtain:

$$\frac{(\sigma_1^{(F)})^4 \cdot (\sigma_2^{(F)})^2 \cdot \|\theta_0\|}{\left(\frac{1}{2} - (\sigma_1^{(F)})\|\theta_0\|\sqrt{\frac{Cd_1^{(F)}}{\pi}}\right)^2} \lesssim \frac{\pi}{\varepsilon \cdot (d_1^{(F)})^{3.5}} \tag{109}$$

Analogous reasoning with $R \ge \frac{\mu_{\text{Jac}}^{(G)}}{2\beta_G}$ for ensuring 4.) holds true in case of $G(\phi_0)$ gives us a similar condition on $\sigma_1^{(G)}$ and $\sigma_2^{(G)}$:

$$\frac{(\sigma_1^{(G)})^4 \cdot (\sigma_2^{(G)})^2 \cdot \|\phi_0\|}{\left(\frac{1}{2} - (\sigma_1^{(G)})\|\phi_0\|\sqrt{\frac{Cd_1^{(G)}}{\pi}}\right)^2} \lesssim \frac{\pi}{\varepsilon \cdot (d_1^{(G)})^{3.5}} \tag{110}$$

For ensuring 2.), by Lemmas F.1-F.2 and using $R \ge \frac{\mu_{\text{Jac}}^{(F)}}{2\beta_F}$, we see that we need $(3.47\sigma_{\max}(A)\sigma_1^{(G)}\sigma_2^{(G)}d_1^{(G)}) \cdot \left(\frac{7}{4}\sigma_2^{(F)}\sqrt{d_1^{(F)}}\right) \cdot \left(C'\sigma_1^{(F)}\|\theta_0\|d_1^{(F)}\right) \lesssim \frac{R^2}{32}$ which yields the following sufficient condition:

$$(3.47\sigma_1^{(G)}\sigma_2^{(G)}d_1^{(G)})\left(\frac{7}{4}\sigma_2^{(F)}\sqrt{d_1^{(F)}}\right)\left(C'\sigma_1^{(F)}\|\theta_0\|d_1^{(F)}\right) \lesssim \frac{R^2}{32\sigma_{\max}(A))} \tag{111}$$

$$= \frac{(\mu_{\text{Jac}}^{(F)})^2}{128\sigma_{\max}(A)\beta_F^2} \tag{112}$$

$$\lesssim \frac{\left(\frac{1}{2} - (\sigma_1^{(F)})\|\theta_0\|\sqrt{\frac{Cd_1^{(F)}}{\pi}}\right)^2 \cdot \pi}{\sigma_{\max}(A)(\sigma_1^{(F)})^2 d_1^{(F)}} \tag{113}$$

$$(\text{By Lemma } F.1 - F.2)$$

$$\implies \frac{(\sigma_1^{(G)}\sigma_2^{(G)})((\sigma_1^{(F)})^3\sigma_2^{(F)})\|\theta_0\|}{\left(\frac{1}{2} - (\sigma_1^{(F)})\|\theta_0\|\sqrt{\frac{Cd_1^{(F)}}{\pi}}\right)^2} \lesssim \frac{\pi}{\sigma_{\max}(A)(d_1^{(F)})^{2.5}} \tag{114}$$

Similarly, for ensuring 3.), we see that we need $(3.47\sigma_{\max}(A)\sigma_1^{(F)}\sigma_2^{(F)}d_1^{(F)}) \cdot \left(\frac{7}{4}\sigma_2^{(G)}\sqrt{d_1^{(G)}}\right) \cdot$
$\left(C'\sigma_1^{(G)}\|\phi_0\|d_1^{(G)}\right) \lesssim \frac{R^2}{32}$ which yields the following sufficient conditions when we use $R \geq \frac{\mu_{\text{Jac}}^{(G)}}{2\beta_g}$:

$$\frac{(\sigma_1^{(F)}\sigma_2^{(F)})((\sigma_1^{(G)})^3\sigma_2^{(G)})\|\phi_0\|}{\left(\frac{1}{2} - (\sigma_1^{(G)})\|\phi_0\|\sqrt{\frac{Cd_1^{(G)}}{\pi}}\right)^2} \lesssim \frac{\pi}{\sigma_{\max}(A)(d_1^{(G)})^{2.5}} \tag{115}$$

$$\tag{116}$$

Thus, w.p. $\geq 1 - 4e^{-\frac{d_1^{(F)}}{64}} - 4e^{-\frac{d_1^{(G)}}{64}} - e^{-Cd_1^{(F)}} - e^{-Cd_1^{(G)}}$, we stay within a ball around the random initializations $\mathcal{B}((\theta_0, \phi_0), R)$ thereby ensuring min-max objective satisfies 2-sided PŁ-condition. By Lemma 3.2, given that we have chosen appropriate fixed learning rates as per Lemma E.7, we are now guaranteed to reach the saddle point. $\qquad\square$

# G Proofs for Neural-Parameters Min-Max Games

> The following lemma establishes a sharp connection between the spectral properties of the input data and the initial conditioning of the neural network's Jacobian, highlighting how data diversity directly influences the minimum singular value at initialization.

**Lemma G.1** (Lemma 3.7 in Main Paper; Lemma 3 & Appendix E.1–E.4 in [106]). *Suppose that a two-layer neural network, $F_\theta$, as defined in Definition 2.3, satisfies Assumption 2.4 and $\tau^{r_1}|\psi(a)| \leq |\psi(\tau a)| \leq \tau^{r_2}|\psi(a)|$, respectively for all $a$, $0 < \tau < 1$, and some constants $r_1, r_2$. Then the neural network Jacobian for a random Gaussian initialization $\theta_0 = ((W_1^{(F)})_0, (W_2^{(G)})_0)$ has the following lower bounds on its smallest singular value w.p. $\geq 1 - (p_1 + p_2)$:*

$$\mu_{Jac}^{(F)} \geq (\sigma_1^{(F)})^{r_1}\sqrt{(1-\delta_1)\frac{c_t^2}{t!}d_1^{(F)}} \cdot \sigma_{\min}(X^{*t}) \tag{117}$$

*We have the following upper bound on its largest singular value w.p. $\geq 1 - (p_1 + p_3 + p_4)$:*

$$\nu_{Jac}^{(F)} \lesssim \sigma_2^{(F)}\dot{\psi}_{\max}\sigma_{\max}(X)\sqrt{d_1^{(F)}} + (\sigma_1^{(F)})^{r_1}\sqrt{(1+\delta_2)(c_1^2 + c_\infty^2)d_1^{(F)}}\sigma_{\max}(X)$$

$$+ (\sigma_1^{(F)})^{r_2}|c_0|\sqrt{(1+\delta_2)d_1^{(F)}n} \tag{118}$$

*And the smoothness constant for the neural network can be computed as*

$$\beta_F = \sqrt{2}\sigma_{\max}(X)(\dot{\psi}_{\max} + \ddot{\psi}_{\max}\chi_{\max}) \tag{119}$$

*where $\chi_{\max} = \sup_{W_2^{(F)}} \sigma_{\max}(W_2^{(F)})$. Here $\{c_i\}_i$ denote Hermite expansion coefficients corresponding to $\psi((W_1^{(F)})_0 X)$, $\delta_j > 0\ \forall j \in [4]$, $p_1 = (d_1^{(F)})^{-Ck_1 d_0^{(F)}} + (d_1^{(F)})^{-Ck_2 d_2^{(F)}}$ for universal constant $C$ with sufficiently large $k_1, k_2$, $p_2 = \exp\left(-\left(\frac{\delta_1\sigma_{\min}(\mathbb{E}[M_0])}{4\dot{\psi}_{\max}^2\sigma_{\max}^2(X)k_1\sigma_1^{(F)}\sqrt{d_0^{(F)}\log d_1^{(F)}}}\right)^2\right)$ where $M_0 = \psi(X^\top((W_1^{(F)})_0^\top)\psi((W_1^{(F)})_0 X)$, $p_3 = \exp\left(-\left(\frac{\delta_2\sigma_{\max}(\mathbb{E}[M_0])}{4\dot{\psi}_{\max}^2\sigma_{\max}^2(X)k_1\sigma_1^{(F)}\sqrt{d_0^{(F)}\log d_1^{(F)}}}\right)^2\right)$ & $p_4 = e^{-C'd_1^{(F)}}$ for a universal constant $C'$.*

> Before we present the following lemma, we offer some context and motivation: In input-optimization games, the gradient norm structure naturally arises from the formulation of hidden bilinear zero-sum games. In more general settings, the relevant properties are detailed in Appendix D. The lemma below demonstrates that, under appropriate initialization and sufficient overparameterization, the neural network output remains bounded from above in terms of spectral properties of the data matrix with high probability. This property will play a critical role in ensuring that the optimization trajectory remains confined within the well-conditioned region (the ball).

**Lemma G.2** (Lemma 3.7 in Main Paper; Neural network output is bounded w.h.p.; Appendix E.5 in [106]). *Consider a neural network $F_\theta$ with parameters $\theta = (W_1^{(F)}, W_2^{(F)})$ as defined in Lemma G.1 above. Suppose we randomly initialize the neural network at $\theta_0$ by choosing $\sigma_1^{(F)}$ and $\sigma_2^{(G)}$ such that*

$$\sigma_1^{(F)}\sigma_2^{(F)} \lesssim \frac{1}{\sqrt{d_0^{(F)}d_1^{(F)}}}$$

*Then w.p. $\geq 1 - p_1 - p_5$, the neural network output at this random initialization $\theta_0$ for the given training data $\mathcal{D}_F$ (as described in Assumption 2.4) is bounded from above as follows:*

$$\|F_{\theta_0}(\mathcal{D}_F)\| \lesssim \delta_3 k_1 k_2 \sigma_{\max}(X) \tag{120}$$

*where $p_1 = (d_1^{(F)})^{-Ck_1 d_0^{(F)}} + (d_1^{(F)})^{-Ck_2 d_2^{(F)}}$ for universal constant $C$ with sufficiently large $k_1, k_2$, $\delta_3 > 0$, and $p_5 = e^{-C\delta_3^2}$ for some universal constant $C$.*

**Theorem G.3** (Theorem 3.8 in Main Paper; HSCSC Games with AltGDA). *Suppose there are two two-layer neural networks, $F_\theta$, $G_\phi$ as defined in Definition 2.3 which satisfy Assumption 2.4 and $\tau^{r_1}|\psi(a)| \leq |\psi(\tau a)| \leq \tau^{r_2}|\psi(a)|$, respectively for all $a$, $0 < \tau < 1$, and some constants $r_1, r_2$. Suppose the network parameters $\theta_0$ and $\phi_0$ are randomly initialized as in Assumption 3.6 with $(\sigma_1^{(F)}, \sigma_2^{(F)})$ and $(\sigma_1^{(G)}, \sigma_2^{(G)})$, respectively, which satisfy*

$$\sigma_1^{(F)}\sigma_2^{(F)} \lesssim \frac{1}{\sqrt{d_0^{(F)}d_1^{(F)}}} \quad \text{and} \quad \sigma_1^{(G)}\sigma_2^{(G)} \lesssim \frac{1}{\sqrt{d_0^{(G)}d_1^{(G)}}} \tag{121}$$

*and suppose that the hidden layer widths $d_1^{(F)}$ and $d_1^{(G)}$ for the two networks F and G satisfy*

$$d_1^{(F)} = \widetilde{\Omega}\left(\mu_\theta^2 \frac{n^3}{d_0^{(F)}}\right) \ \& \ d_1^{(G)} = \widetilde{\Omega}\left(\mu_\phi^2 \frac{n^3}{d_0^{(G)}}\right) \tag{122}$$

*where the datasets $(\mathcal{D}_F, \mathcal{D}_G)$ for both the players are assumed to be of size $n$. Then Alternating Gradient-Descent-Ascent procedure with appropriate fixed learning rates $\eta_\theta, \eta_\phi$ (see Lemma E.7) for an $(\mu_\theta, \mu_\phi)$-HSCSC and $L_{\nabla\mathcal{L}}$-smooth min-max objective $\mathcal{L}_\mathcal{D}(F_\theta, G_\phi)$ as defined in Equation 1 satisfying Assumption 2.1 converges to the saddle point $(\theta^*, \phi^*)$ exponentially fast with probability at least $1 - (p_1 + p_2 + p_3 + p_4 + p_5) - (p_1 + p_2' + p_3' + p_4' + p_5')$ (Here, the failure probabilities $p_j$'s and $p_j'$'s are defined for networks $F_\theta$ and $G_\phi$, respectively, as per Lemmas G.1-G.2).*

*Proof.* If we randomly initialize the neural network at $\theta_0 = ((W_1^{(F)})_0, (W_2^{(G)})_0)$ as per the stated initialization scheme in Assumption 3.6 with Equation (121), we have by Lemma G.2 that w.p. $\geq 1 - p_1 - p_5$:

$$\|F_{\theta_0}(\mathcal{D}_F)\| \lesssim \delta_3 k_1 k_2 \sigma_{\max}(X) \tag{123}$$

where $p_1, p_5$ are as defined in Lemma G.2. Analogous reasoning gives a similar bound for the output of initialization condition for the neural network $G_\phi$ with data $\mathcal{D}_G$.

Since our min-max objective $\mathcal{L}_\mathcal{D}(F_\theta, G_\phi)$ is separable as defined in Equation 1, we can start by rewriting the bilinear component $I_2^\mathcal{D}(F_\theta, G_\phi) = (F_\theta(\mathcal{D}_F))^\top A(G_\phi(\mathcal{D}_G))$. Firstly, we can compute the gradients for the min-max objective as follows given data $\mathcal{D} = (\mathcal{D}_F, \mathcal{D}_G)$:

$$\nabla_\theta L_\mathcal{D}(F_\theta, G_\phi) = \nabla_\theta(I_1^{\mathcal{D}_F}(F_\theta) + I_2^\mathcal{D}(F_\theta, G_\phi)) \tag{124}$$

$$= (\nabla_\theta F_\theta(\mathcal{D}_F))^\top(\nabla_{z=F_\theta(\mathcal{D}_F)}I_1^{\mathcal{D}_F}(z)) + (\nabla_\theta F_\theta(\mathcal{D}_F))^\top A G_\phi(\mathcal{D}_G) \tag{125}$$

$$\nabla_\phi L_\mathcal{D}(F_\theta, G_\phi) = \nabla_\phi(-I_3^{\mathcal{D}_G}(G_\phi) + I_2^\mathcal{D}(F_\theta, G_\phi)) \tag{126}$$

$$= -(\nabla_\phi G_\phi(\mathcal{D}_G))^\top(\nabla_{z=G_\phi(\mathcal{D}_G)}I_3^{\mathcal{D}_G}(z)) + (\nabla_\phi G_\phi(\mathcal{D}_G))^\top A F_\theta(\mathcal{D}_F) \tag{127}$$

By Assumption 2.1(iv) on the gradient norm of strongly-convex functions, triangle inequality, and submultiplicativity of operator norm, we can say that the gradient norms of our separable min-max objective can be upper bounded as follows:

$$\|\nabla_\theta L_\mathcal{D}(F_\theta, G_\phi)\| \leq \|(\nabla_\theta F_\theta(\mathcal{D}_F))^\top\|\left(A_1^{(F)}\|F_{\theta_0}(\mathcal{D}_F)\| + A_2^{(F)}\text{diam}(\mathcal{Y}^{(F)}) + A_3^{(F)}\right)$$
$$+ \|(\nabla_\theta F_\theta(\mathcal{D}_F))^\top\|\|A\|\|G_\phi(\mathcal{D}_G)\| \tag{128}$$
$$\|\nabla_\phi L_\mathcal{D}(F_\theta, G_\phi)\| \leq \|(\nabla_\phi G_\phi(\mathcal{D}_G))^\top\|\left(A_1^{(G)}\|G_{\phi_0}(\mathcal{D}_G)\| + A_2^{(G)}\text{diam}(\mathcal{Y}^{(G)}) + A_3^{(G)}\right)$$
$$+ \|(\nabla_\phi G_\phi(\mathcal{D}_G))^\top\|\|A\|\|F_\theta(\mathcal{D}_F)\|$$

Since $\mathcal{L}(\theta, \phi)$ is $(\mu_\theta, \mu_\phi)$-hidden-strongly-convex-strongly-concave, by Fact 2.6 it also satisfies the 2-sided PŁ-condition with $(\mu_\theta \cdot \sigma_{\min}^2(\nabla_\theta F_\theta), \mu_\phi \cdot \sigma_{\min}^2(\nabla_\phi G_\phi))$ PŁ-moduli w.p. $\geq 1 - p_1 - p_2$ (where $p_1, p_2$ as defined in Lemma G.1). Thus we can utilise path-length bound as derived in Lemma 3.2 which leaves us with controlling the potential $P_0$ for ensuring convergence to the saddle point. Given the loss gradients above (Equation (128)) and using the sufficient condition for ensuring iterates do not leave the ball $\mathcal{B}((\theta_0, \phi_0), R)$ (where $R = \frac{\max\{\mu_{\text{Jac}}^{(F)}, \mu_{\text{Jac}}^{(G)}\}}{\min\{\beta_F, \beta_G\}}$ as defined in Section 3) thus

ensuring non-singular Jacobian inside the ball for both the neural networks (Lemma E.7), we require the following:

$$\left( \underbrace{\|(\nabla_\theta F_{\theta_0}(\mathcal{D}_F))^\top\| \cdot \left( A_1^{(F)} \|F_{\theta_0}(\mathcal{D}_F)\| + A_2^{(F)} \mathrm{diam}(\mathcal{Y}^{(F)}) + A_3^{(F)} \right)}_{T_1} \right.$$

$$+ \underbrace{\|(\nabla_\theta F_{\theta_0}(\mathcal{D}_F))^\top\| \|A\| \|G_{\phi_0}(\mathcal{D}_G)\|}_{T_2}$$

$$+ \underbrace{\|(\nabla_\phi G_{\phi_0}(\mathcal{D}_G))^\top\| \cdot \left( A_1^{(G)} \|G_{\phi_0}(\mathcal{D}_G)\| + A_2^{(G)} \mathrm{diam}(\mathcal{Y}^{(G)}) + A_3^{(G)} \right)}_{T_3}$$

$$\left. + \underbrace{\|(\nabla_\phi G_{\phi_0}(\mathcal{D}_G))^\top\| \|A\| \|F_{\theta_0}(\mathcal{D}_F)\|}_{T_4} \right) \lesssim \frac{R^2}{8}$$

In order to ensure the above, we will demand the following:

1. $T_1 \lesssim \frac{R^2}{32}$

2. $T_2 \lesssim \frac{R^2}{32}$

3. $T_3 \lesssim \frac{R^2}{32}$

4. $T_4 \lesssim \frac{R^2}{32}$

We will show arguments for the case of the 'min' player (neural network $F_\theta$) here. Exactly the same arguments provide the analogous result for the 'max' player (neural network $G_\phi$).

For ensuring (1.)–(4.), we will use the fact that $\sigma_{\max}(\nabla_\theta F_{\theta_0}(\mathcal{D}_F)) \leq \nu_{\mathrm{Jac}}^{(F)}$, $\sigma_{\max}(\nabla_\phi G_{\phi_0}(\mathcal{D}_G)) \leq \nu_{\mathrm{Jac}}^{(G)}$, Lemma G.1 for upper bounds on $\nu_{\mathrm{Jac}}^{(F)}$, and $\nu_{\mathrm{Jac}}^{(G)}$ and smoothness constants for two-layer neural networks along with the upper bounds on neural network outputs for $F$ and $G$ as derived above in Lemma G.2.

Thus, a sufficient condition for ensuring 1.) would be to use $R \geq \frac{\mu_{\mathrm{Jac}}^{(F)}}{2\beta_F}$ and see that w.p. $\geq 1 - p_1 - p_2 - p_3 - p_4 - p_5$ the following holds (assuming $|c_0|$ is sufficiently large s.t. $|c_0|\sqrt{(1+\delta_2)d_1^{(F)}n}$ becomes the dominating term in $\nu_{\mathrm{Jac}}^{(F)}$):

$$\underbrace{C_1 \cdot \left( |c_0|\sqrt{(1+\delta_2)d_1^{(F)}n} \right) \cdot \left( A_1^{(F)}\delta_3 k_1 k_2 \sigma_{\max}(X) + A_2^{(F)}\mathrm{diam}(\mathcal{Y}^{(F)}) + A_3^{(F)} \right)}_{(LHS)} \lesssim \frac{1}{32}\left( \frac{\mu_{\mathrm{Jac}}^{(F)}}{2\beta_F} \right)^2$$

$$\implies (LHS) \lesssim \frac{(\sigma_1^{(F)})^{2r_1}(1-\delta_1)(c_t^2/t!)d_1^{(F)}\sigma_{\min}^2(X^{*t})}{2(\sigma_{\max}(X))^2(\dot\psi_{\max} + \ddot\psi_{\max}\chi_{\max})^2} \quad \text{(By Lemma G.1)} \tag{129}$$

$$\implies d_1^{(F)} \gtrsim \frac{(|c_0|^2(1+\delta_2)n) \cdot (\sigma_{\max}(X))^4(\dot\psi_{\max} + \ddot\psi_{\max}\chi_{\max})^4 \cdot (A_1^{(F)}\delta_3 k_1 k_2 \sigma_{\max}(X))^2}{(\sigma_1^{(F)})^{2+2r_1}(1-\delta_1)^2(c_t^4/(t!)^2)\sigma_{\min}^4(X^{*t})} \tag{130}$$

$$\implies d_1^{(F)} \gtrsim \frac{(|c_0|^2(1+\delta_2)n) \cdot (\sigma_{\max}(X))^4(\dot\psi_{\max} + \ddot\psi_{\max}\chi_{\max})^4 \cdot (A_1^{(F)}\delta_3 k_1 k_2 \sigma_{\max}(X))^2}{(\sigma_1^{(F)})^{2+2r_1}(1-\delta_1)^2(c_t^4/(t!)^2)\sigma_{\min}^4(X^{*t})} \tag{131}$$

$$\therefore d_1^{(F)} \gtrsim \xi(C_\delta, t, \psi, \{c_i\}_{i \geq 0})\frac{n(A_1^{(F)})^2\sigma_{\max}^6(X)}{\sigma_{\min}^4(X^{*t})} \tag{132}$$

where $\delta_4 = \max\{k_1, k_2\}$, $C_\delta = \{\delta_1, \delta_2, \delta_3, \delta_4\}$, and

$$\xi(C_\delta, t, \psi, \{c_i\}_{i\geq 0}) = \frac{\left(|c_0|^2(1+\delta_2)\right) \cdot (\dot{\psi}_{\max} + \ddot{\psi}_{\max}\chi_{\max})^4 \cdot (\delta_3 k_1 k_2)^2}{(\sigma_1^{(F)})^{2+2r_1}(1-\delta_1)^2(c_t^4/(t!)^2)}$$

Here, as per our Assumption 2.4 on the data and arguing along the lines of Section 2.1 in Oymak and Soltanolkotabi [86], when we use the fact that $\sigma_{\max}(X) \simeq \sqrt{\frac{n}{d_0^{(F)}}}$, $\sigma_{\min}(X^{*t}) \simeq \sqrt{\frac{n}{(d_0^{(F)})^t}} \simeq 1$[9], and $n \simeq (d_0^{(F)})^t$ where $t \geq 2$[10], we get that:

$$d_1^{(F)} = \tilde{\Omega}\left(\frac{n \cdot n^3 \cdot (A_1^{(F)})^2}{(d_0^{(F)})^3 \cdot 1}\right) = \tilde{\Omega}\left(\frac{n^3 \mu_\theta^2}{d_0^{(F)}}\right) \tag{133}$$

In the second equality above, we used the observation from Appendix D that $A_1 = \theta(\mu)$ where $\mu$ is the strong-convexity modulus. Thus, the amount of overparameterization we need for network $F_\theta$ is as follows:

$$d_1^{(F)} d_0^{(F)} = \tilde{\Omega}(n^3 \mu_\theta^2) \tag{134}$$

Analogous reasoning for ensuring (3.) with $R \geq \frac{\mu_{\mathrm{Jac}}^{(G)}}{2\beta_G}$ gives us w.p. $1 - p_1' - p_2' - p_3' - p_4' - p_5'$:

$$d_1^{(G)} \gtrsim \xi(L_{\mathcal{L}}, C_\delta', t', \psi, \{c_i'\}_{i\geq 0}) \frac{n(A_1^{(G)})^2 \sigma_{\max}^6(X)}{\sigma_{\min}^4(X^{*t})} \tag{135}$$

Using arguments from Oymak and Soltanolkotabi [86] as done above for the case of 1.), we get a similar cubic overparameterization bound for the 'max' player, $G_\phi$, as well:

$$d_1^{(G)} d_0^{(G)} = \tilde{\Omega}(n^3 \mu_\phi^2) \tag{136}$$

We get the following w.p. $\geq 1 - (p_1' + p_5') - (p_1 + p_2 + p_3 + p_4)$ by similar reasoning as above for ensuring 2.) holds along with using $R \geq \frac{\mu_{\mathrm{Jac}}^{(F)}}{2\beta_F}$:

$$\underbrace{\left(|c_0|\sqrt{(1+\delta_2)d_1^{(F)}n}\right) \cdot \sigma_{\max}(A) \cdot (\delta_3' k_1' k_2' \sigma_{\max}(X))}_{(LHS)} \lesssim \frac{1}{32}\left(\frac{\mu_{\mathrm{Jac}}^{(F)}}{2\beta_F}\right)^2$$

$$\implies (LHS) \lesssim \frac{(\sigma_1^{(F)})^{2r_1}(1-\delta_1)(c_t^2/t!)d_1^{(F)}\sigma_{\min}^2(X^{*t})}{2(\sigma_{\max}(X))^2(\dot{\psi}_{\max} + \ddot{\psi}_{\max}\chi_{\max})^2} \quad \text{(By Lemma G.1)} \tag{137}$$

$$\therefore d_1^{(F)} \gtrsim \frac{\left(|c_0|^2(1+\delta_2)n\right) \cdot (\sigma_{\max}(X))^4(\dot{\psi}_{\max} + \ddot{\psi}_{\max}\chi_{\max})^4 \cdot \sigma_{\max}^2(A) \cdot (\delta_3' k_1' k_2' \sigma_{\max}(X))^2}{(\sigma_1^{(F)})^{2+2r_1}(1-\delta_1)^2(c_t^4/(t!)^2)\sigma_{\min}^4(X^{*t})}$$
$$\tag{138}$$

$$\implies d_1^{(F)} \gtrsim \xi(A, C_{(\delta,\delta')}, t, \psi, \{c_i\}_{i\geq 0})\frac{n\sigma_{\max}^6(X)}{\sigma_{\min}^4(X^{*t})} \tag{139}$$

Again, using arguments from Oymak and Soltanolkotabi [86] above, we get a similar cubic overparameterization bound for the 'min' player, $F_\theta$, as above:

$$d_1^{(F)} d_0^{(F)} = \tilde{\Omega}(n^3) \tag{140}$$

Applying reasoning analogous to that for ensuring (2.) to the case of ensuring (4.) holds, by using $R \geq \frac{\mu_{\mathrm{Jac}}^{(G)}}{2\beta_G}$ we get w.p. $\geq 1 - (p_1 + p_5) - (p_1' + p_2' + p_3' + p_4')$:

$$d_1^{(G)} \gtrsim \xi(A, L_{\mathcal{L}}, C_\delta', t', \psi, \{c_i'\}_{i\geq 0})\frac{n\sigma_{\max}^6(X)}{\sigma_{\min}^4(X^{*t})} \tag{141}$$

---

[9]For a matrix $W \in \mathbb{R}^{m \times n}$ and $t \in Z_{\geq 1}$, the Khatri-Rao product is denoted as $W^{*t} \in \mathbb{R}^{m^t \times n}$ with its $j$-th column defined as vector$(w_j \otimes \cdots \otimes w_j) \in \mathbb{R}^{m^t}$ where $\otimes$ denotes Kronecker product.

[10]In practice, one typically has $n \simeq (d_0^{(F)})^t$ for $t \geq 2$.

Finally, as was done for the case of ensuring 1.), 3.) and 4.) above, we once again arguments from Oymak and Soltanolkotabi [86] for spectral properties of the training data $\mathcal{D}$ and get a similar cubic overparameterization bound for the 'max' player, $G_\phi$, as well:

$$d_1^{(G)} d_0^{(G)} = \tilde{\Omega}(n^3) \tag{142}$$

Thus, w.p. $\geq 1 - (p_1' + p_5') - (p_1 + p_2 + p_3 + p_4)$, we stay within a ball around the random initializations $\mathcal{B}((\theta_0, \phi_0), R)$ thereby ensuring min-max objective satisfies 2-sided PŁ-condition. By Lemma 3.2, given that we have chosen appropriate fixed learning rates as per Lemma E.7, we are now guaranteed to reach the saddle point.

$\square$

**Remark G.4.** The proof above requires activation function to not be an odd function for ensuring $c_0 \neq 0$.

**Remark G.5** (Effects of assumption about $\sigma_{\max}(X)$ on amount of overparameterization). As noted in the footnote pertaining to $\sigma_{\max}(X)$ for the data matrix $X$ in Assumption 2.5, if all we know about the data matrix is that it's row-normalized and that it's *not* a random matrix (e.g. random Gaussian matrix), then we can conclude that $\sigma_{\max}(X) = O(\sqrt{n})$. Using this bound on the maximum singular value of the data matrix instead, we can conclude from Equations (132), (135), (139), and (141) that

$$d_1^{(F)} = \tilde{\Omega}(n^4) \tag{143}$$

$$d_1^{(G)} = \tilde{\Omega}(n^4) \tag{144}$$

Since we have $n \simeq (d_0^{(F)})^t$ $(t \geq 2)$ in practice for the MIN player $F_\theta$ (analogously, $n \simeq (d_0^{(G)})^t$ for the MAX player $G_\phi$), we can conclude that the amount of overparameterization needed for both the players is more than the cubic overparameterization when the data matrix is, for example, an i.i.d. random Gaussian matrix:

$$d_1^{(F)} d_0^{(F)} = \tilde{\Omega}(n^{4+1/t}) \tag{145}$$

$$d_1^{(G)} d_0^{(G)} = \tilde{\Omega}(n^{4+1/t}) \tag{146}$$

# H   Clarifications

## H.1   Smoothness on variables or Map

In the main paper, we state the following assumption regarding the objective function:

**Assumption H.1** (Smoothness, Hidden Strong Convexity, and Gradient Control).

(i) **Smoothness:** Each sample-wise loss $\ell(y, h = \mathrm{Map}_w(x))$ is differentiable and $L$-smooth with respect to $h$.
(ii) **Coupling Structure:** Each bilinear coupling matrix $A(x_i, x_j, y_i, y_j)$ is known, fixed, and has bounded operator norm.
(iii) **Hidden Strong Convexity:** Each sample-wise loss $\ell(y, h = \mathrm{Map}_w(x))$ is strongly convex with respect to the neural network output $h$.
(iv) **Gradient Growth Condition:** There exist constants $A_1, A_2, A_3 > 0$ such that for all $h \in \mathbb{R}^{d_{\mathrm{out}}}$ and $y \in \mathcal{Y}$, the latent gradient of each loss satisfies:

$$\|\nabla_h \ell(y, h)\| \le A_1 \|h\| + A_2 \operatorname{diam}(\mathcal{Y}) + A_3.$$

To compute the smoothness of the composition $\ell(y, \mathrm{Map}_w(x))$ with respect to $w$, we invoke the following classical result[11]:

**Lemma H.3** (Composition Smoothness, adapted from Proposition 2(c) in [131]). *Let $f : \mathbb{R}^d \to \mathbb{R}$ be a closed, convex function that is $L_f$-locally-Lipschitz on a Euclidean ball $\mathcal{B}(x_0, R)$ for some fixed $x_0 \in \mathbb{R}^d$ and $R > 0$ and let $g : \mathbb{R}^n \to \mathbb{R}^d$ be an $L_g$-smooth function. Then, the composition $f \circ g$ is $L_f L_g$-smooth in the Euclidean ball $\mathcal{B}(x_0, R)$.*

## H.2   Refined Upper Bound Expression for Lyapunov Potential $P_0$

For clarity and presentation purposes, the main paper presents a simplified—yet qualitatively accurate—upper bound on the initial potential $P_0$. In appendix, we provide the precise formulation that captures the correct dependence on constants and exponents. For completeness, we restate both results below:

The key distinction in the refined expression lies in the appearance of additional quadratic terms. As we show in the accompanying proof, these higher-order contributions can be effectively controlled through suitable initialization and appropriately chosen step sizes. Thus, the improved bound yields tighter theoretical insight without compromising practical applicability.

---

11

**Remark H.2.** This adaptation is necessary, as a function cannot simultaneously be globally Lipschitz (bounded gradients) and strongly convex (increasing gradient norm) in the unconstrained setting. However, under sufficient overparameterization and appropriate random initialization, we show that the AltGDA iterates remain within a bounded region where these properties hold locally. Thus, our assumption of local Lipschitzness of latent loss, the induced smoothness and the hidden strong convexity within a Euclidean ball is both theoretically consistent and empirically valid.

*Main Paper Version:*

**Lemma 3.3** (Upper Bound on Initial Potential $P_0$). *Suppose the min-max objective $\mathcal{L}(\theta, \phi)$ is $L_{\mathcal{L}}$-Lipschitz and satisfies a two-sided PŁ condition with constants $(\mu_\theta, \mu_\phi)$. Then the initial Lyapunov potential $P_0 \leq L_{\mathcal{L}} \left( C_1 \cdot \|\nabla_\theta \mathcal{L}(\theta_0, \phi_0)\| + C_2 \cdot \|\nabla_\phi \mathcal{L}(\theta_0, \phi_0)\| \right)$, where $C_1, C_2 = \Theta \left( L_{\mathcal{L}}/\mu_\theta^3 \right)$.*

*Appendix Version (Refined):*

**Lemma E.4** (Upper Bound on Initial Potential $P_0$; Lemma 3.3 in Main Text). *Let $F_\theta$ and $G_\phi$ be neural network mappings such that they are $\beta_F$ and $\beta_G$ smooth as defined in Definition 2.3. Now let $(\theta_0, \phi_0)$ be such that Jacobian singular values for both the networks are strictly positive and bounded from above and below, $\mu_{Jac}^{(F)} \leq \sigma(\nabla_\theta F_{\theta_0}) \leq \nu_{Jac}^{(F)}$ and $\mu_{Jac}^{(G)} \leq \sigma(\nabla_\phi G_{\phi_0}) \leq \nu_{Jac}^{(G)}$. Suppose the min-max objective $\mathcal{L}(\theta, \phi)$ is $(\mu_\theta, \mu_\phi)$-HSCSC. Then the initial Lyapunov potential $P_0$ can be bounded from above as:*

$$
\begin{aligned}
P_0 \leq & \; \|(\nabla_{F_\theta} \mathcal{L}(F_{\theta_0}, G_{\phi_0}))^\top\| \cdot \nu_{Jac}^{(F)} \cdot \frac{1}{\mu_\theta (\mu_{Jac}^{(F)})^2} \cdot \|\nabla_\theta \mathcal{L}(\theta_0, \phi_0)\| \\
& + \|(\nabla_{F_\theta} \mathcal{L}(F_{\theta_0}, G_{\phi_0}))^\top\| \cdot \beta_F \frac{1}{2\mu_\theta^2 (\mu_{Jac}^{(F)})^4} \cdot \|\nabla_\theta \mathcal{L}(\theta_0, \phi_0)\|^2 \quad\quad (36) \\
& + (1 + \lambda) \cdot \|\left(\nabla_{G_\phi} \mathcal{L}(F_{\theta_0}, G_{\phi_0})\right)^\top\| \cdot \nu_{Jac}^{(G)} \cdot \frac{1}{\mu_\phi (\mu_{Jac}^{(G)})^2} \cdot \|\nabla_\phi \mathcal{L}(\theta_0, \phi_0)\| \\
& + (1 + \lambda) \cdot \|\left(\nabla_{G_\phi} \mathcal{L}(F_{\theta_0}, G_{\phi_0})\right)^\top\| \cdot \beta_G \frac{1}{2\mu_\phi^2 (\mu_{Jac}^{(G)})^4} \cdot \|\nabla_\phi \mathcal{L}(\theta_0, \phi_0)\|^2
\end{aligned}
$$

## H.3 Random Initialization: How restrictive are our results?

We note the following for both the input games and neural games regarding random initializations:

1. **Neural Games:** Our main result (Theorem 3.8) assumes a commonly used initialization scheme (e.g., He or LeCun), with the only additional requirement being that the variances satisfy Equation (9) in the main text. Of course, bridging the gap between practice and theory, it remains an interesting open question whether the current polynomial overparameterization requirement can be reduced to linear. However, we note that – prior to our work – there were no existing theoretical results connecting initialization schemes with convergence guarantees in neural min-max games.

2. **Input Games:** In this setting, our results are even less restrictive regarding initialization. Specifically, we show that if the neural networks are randomly initialized from a standard Gaussian distribution, then AltGDA can compute the min-max optimal inputs. As discussed above, this is an *average-case* result, in the spirit of smoothed analysis. While it is theoretically expected that there exist neural architectures encoding hard min-max landscapes—potentially close to PPAD-hard instances—in practice, our result suggests that a randomly initialized neural min-max game is tractable under AltGDA.

# I Discussion & Future Work

This work initiates a principled framework for understanding optimization in hidden convex–concave min-max games, a setting central to the theory and practice of modern machine learning. By bridging overparameterization with spectral geometry and alternating dynamics, we show how convergence and equilibrium stability can emerge from architectural design and initialization. Beyond offering the first non-asymptotic guarantees for such hidden structures, our analysis reveals the potential of min-max learning as a structured alternative to unconstrained overfitting. We hope these insights inspire a broader rethinking of how optimization, architecture, and strategic interaction coalesce in scalable intelligent systems.

In this last section, we reflect on our overparameterization bounds in comparison to known results from single-agent minimization, and outline promising directions for future research.

While our analysis shares surface-level similarities with techniques from classical minimization problems, there are crucial structural differences. In the single-agent case, convergence is often guaranteed by showing that the Neural Tangent Kernel (NTK) remains well-conditioned near initialization, and that the loss is already small—a zero-order property.

By contrast, computing a Nash equilibrium in a min-max setting is inherently more subtle: the solution concept is first & second-order. Rather than merely minimizing a loss, we seek to simultaneously drive the gradients of both players to zero while respecting their opposing incentives—capturing the saddle-point nature of equilibrium. Consequently, our notion of being "close to optimality" requires the initialization to yield not only small gradient norms but also a geometric alignment between the descent and ascent directions. This structural gap underpins the difference in overparameterization requirements between the single-agent and multi-agent cases.

These insights naturally give rise to several open questions from deep learning and game theory point of view:

## I.1 Deep Learning Perspective:

- **Sharper Overparameterization Bounds:**
  - Can we tighten the width–depth trade-offs for hidden min-max games, especially in non-bilinear or partially observable regimes?
  - Our results suggest that overparameterization smooths the optimization landscape in simple bilinear games, but the precise scaling with respect to hidden-layer width and depth in *nonlinear* architectures remains unclear. A possible direction is to characterize phase transitions in convergence when width exceeds a critical threshold that depends on the spectral complexity or curvature of the game operator.

- **Beyond Width and Smoothness:**
  - How does the depth of the architecture influence convergence in hidden games? Can we extend current results to networks with non-smooth activations such as ReLU, using tools beyond gradient-Lipschitz analysis?
  - Smoothness assumptions simplify the analysis but obscure the behavior of realistic neural dynamics. We could instead exploit techniques from *non-smooth dynamical systems* and *proximal envelope* theory to handle non-smooth losses.
  - Another open question is whether depth induces implicit averaging effects similar to stochastic smoothing, thereby stabilizing the dynamics in min–max training.

- **Structural Guarantees in Multi-Agent PŁ Games:**
  - What are the necessary structural properties of multi-agent PŁ-type games that ensure convergence to a unique Nash equilibrium? The challenge lies in extending single-agent PŁ conditions to multi-agent systems where gradients interact. Are there generalized PŁ inequalities that capture cross-agent monotonicity?
  - Investigating block-wise or coupled PŁ structures could reveal when independent gradient updates mimic joint gradient descent–ascent in strongly monotone regimes.

- **Computational Complexity:**
  - What is the inherent hardness of solving hidden convex–concave games with overparameterized models? Can we characterize tractable subclasses?
  - The inversion or injectivity verification of neural networks is known to be NP- or coNP-complete even for ReLU architectures (see, e.g., the COLT'25 paper *"Complexity of Injectivity and Verification of ReLU Neural Networks"*). How do such barriers propagate to the equilibrium computation problem when game payoffs are defined implicitly through network mappings?
  - The overparameterized setting often introduces implicit convexification. Can we formalize when this leads to *provable polynomial-time convergence* versus when training remains PPAD- or NP-hard?

- **From Neural Networks to Transformers:**
  - What explains the observed differences in scaling laws between overparameterized feedforward networks and attention-based architectures in game-theoretic learning? Could these insights inform AI alignment and debate frameworks?
  - Transformers introduce dynamic reweighting of information through attention, which may alter the effective conditioning of the game Jacobian.
  - Understanding how self-attention layers modify optimization stability and expressivity could illuminate why Transformers exhibit faster equilibration or better robustness to adversarial perturbations.

- **Beyond Full Gradient Feedback:**
  - Much of the current analysis, including ours, assumes full gradient information. It remains an open and critical question whether similar benefits of overparameterization persist under bandit or partial-information settings.
  - A natural direction is to extend existing results to *stochastic feedback* or *bandit-gradient estimators* using variance-controlled or mirror-descent methods.
  - Another fundamental question: does overparameterization implicitly reduce the variance of policy-gradient estimators by averaging across redundant feature paths?

## I.2  Game Theory Perspective:

- **CCE with Neural Parametrizations (Normal-Form):**
  - *Representational question:* For a class of neural correlating devices $g_\theta(z)$ that map public randomness $z$ to joint actions, characterize when the induced set of implementable distributions is dense in the CCE polytope. What overparameterization (width/depth) suffices for $\varepsilon$-dense coverage uniformly across $n$-player games with bounded payoffs?
  - *Optimization vs. Calibration:* Standard no-regret dynamics imply convergence to CCE in the tabular case. With function approximation (shared neural critics/policies), give conditions (e.g., uniform stability, gradient-calibration bounds) under which the averaged joint play converges to an $\varepsilon$-CCE at rates that improve with network width (via better optimization landscapes) without blowing up statistical complexity.
  - *Implicit bias:* In the overparameterized (NTK) regime, training to minimize regret surrogates biases the joint distribution toward low-complexity mixtures. Can we quantify the *implicit regularization* toward "simple" CCEs (few extreme points in support) as a function of width, depth, and training dynamics?

- **CCE in Extensive-Form Games (EFG): EFCE, CEFCE, NFCCE via Nets.**
  - *Sequential structure:* Compare neural parameterizations for (i) NFCCE (normal-form coarse CCE), (ii) EFCE (extensive-form CE), and (iii) CEFCE (coarse EFCE). Give width/depth conditions under which sequence-form constraints (flow and realization-plan consistency) can be enforced by differentiable layers with projection or Lagrangian penalties.
  - *Counterfactual losses:* Can counterfactual regret minimization with neural policies/critics be shown to converge to EFCE/CEFCE when critics are overparameterized but trained with regularized Bellman residuals? Identify structural conditions (perfect recall, bounded branching, Lipschitz counterfactual values) that guarantee $\tilde{O}(1/\sqrt{T})$ exploitability while using batched, partial counterfactual feedback.

- *Abstraction without tears:* Overparameterized policies can represent rich information-set strategies directly. Develop "learned abstraction" bounds: when does a deep policy with attention over histories match the exploitability of hand-crafted abstractions, and what width/depth yields $\varepsilon$-EFCE support recovery?

- **Markov (Stochastic) Games: Stationary CCE and Overparameterized Critics.**
  - *Stationary CCE:* Define a stationary CCE as a joint policy $\pi$ with a correlating signal that is time-consistent across states. Give conditions under which joint no-regret learning with overparameterized $Q$-critics converges (in Cesàro average) to an $\varepsilon$-stationary CCE, with rates that depend on mixing/concentrability constants and the critic class capacity.
  - *Depth helps bootstrapping:* Hypothesis: deeper critics reduce the Bellman residual floor achievable by gradient methods, improving optimization error; width controls realizability. Provide a decomposition of the CCE gap into Approximation$(\mathcal{F}_Q)$ + Optimization$(\theta)$ + Statistical$(\mathcal{F}_Q, \text{data})$, and bound each term as a function of network depth/width and exploration coverage.
  - *Temporal correlation:* Analyze how correlating devices that are *state-dependent* (learned correlation) interact with Markovian dynamics. When does limited-bandwidth correlation (few bits per step) suffice for $\varepsilon$-CCE in ergodic MGs?

- **Overparameterization vs. Bellman–Eluder (BE) / Bellman Rank Complexity.**
  - *Function-class lens:* Let $\mathcal{F}_Q$ be the $Q$-function class realized by a network in the linearized (NTK) regime. Its eluder dimension equals the feature dimension for linear classes; for deep nets, the *effective* eluder dimension depends on width, depth, and implicit regularization. Conjecture: with proper norm control (weight decay/early stopping), the *effective* BE dimension scales like the *stable rank* of the induced features, not the raw parameter count.
  - *Sample complexity for $\varepsilon$-CCE:* In two-player zero-sum Markov games with realizable $Q^* \in \mathcal{F}_Q$, the sample complexity to reach $\varepsilon$-CCE under fitted Q-type updates should scale as

  $$\tilde{O}\Big(\frac{\text{BE}-\dim(\mathcal{F}_Q)\ +\ \log(1/\delta)}{(1-\gamma)^p\, \varepsilon^2}\Big),$$

  for a small integer $p$ depending on the algorithm (e.g., $p \in \{2, 3\}$), assuming standard concentrability/mixing. Overparameterization *per se* does not hurt if the BE dimension is controlled by implicit/explicit regularization.
  - *Width–complexity trade:* Identify regimes where increasing width improves optimization (faster approach to $\theta^*$) while keeping BE-dimension nearly constant via margin-based or path-norm constraints, yielding *strictly better* time–sample trade-offs.
  - *Transformers vs. MLPs:* Attention layers can implement dynamic state-action feature selection, potentially lowering the BE dimension for tasks with sparse predictive structure (long-range but low "intrinsic" rank). Formalize when attention reduces BE-dim relative to equally wide MLPs, explaining empirical scaling differences in Markov game benchmarks.

- **Bandit/Partial Information Extensions:**
  - *Bandit CCE:* For normal-form/Markov games with bandit feedback, derive $\varepsilon$-CCE rates when both policies and correlating devices are neural. Key ingredient: variance-controlled gradient surrogates (e.g., doubly-robust or control-variates) to keep estimation error compatible with a bounded BE dimension.
  - *Information complexity of correlation:* Quantify the *information budget* (bits of correlation, episodes of exploration) required to learn $\varepsilon$-CCE as a function of BE dimension and mixing; relate to *communication complexity* of implementing the equilibrium.

**Conclusion.**    Our study sheds new light on the geometry of overparameterized min-max optimization by establishing the first precise convergence guarantees in hidden bilinear games. We demonstrate how overparameterization not only facilitates convergence but also implicitly regularizes the optimization landscape, ensuring robustness through well-conditioned spectral structure. Crucially, we bridge the gap between neural initialization and equilibrium computation, revealing that convergence in adversarial training is governed by deeper geometric principles. These insights open new avenues for understanding and designing scalable multi-agent learning systems, and we believe they mark a significant step toward principled foundations for modern generative and strategic AI.

