# OpenReview forum: "Solving Neural Min-Max Games: The Role of Architecture, Initialization & Dynamics"
_NeurIPS.cc/2025/Conference — NeurIPS 2025 spotlight_

### Official Review · Reviewer_njLz · 2025-06-05

**Clarity:** 3
**Significance:** 2
**Originality:** 3
**Rating:** 4
**Confidence:** 3

**Summary:**

This paper investigates the problem of solving *neural min-max games* where the players are parameterized by neural networks (Network Optimization) or the strategies are the input of data (input Optimization). Specifically, the authors prove that if the players are parameterized by sufficiently wide two-layer neural networks, then under suitable initialization, the alternating Gradient Descent Ascent algorithm converges to the Nash equilibrium of the neural min-max games.

**Questions:**

My questions primarily concern how the settings and theoretical results in this paper reflect real-world training dynamics in min-max games. While the paper clearly focuses on theoretical contributions, I believe demonstrating that the theoretical phenomena (e.g., **overparameterization enabling convergence** and the **Gaussian initialization placing parameters near game equilibria**) have observable counterparts in practical applications would significantly enhance the paper's impact.

In particular, my questions include:

* Beyond mathematical tractability, are there other reasons to focus on two-layer neural networks in neural min-max games? For example, is it possible to prove an analogy of universial approximation theorem for two-layer neural networks in min-max games?

* How do the theoretical phenomena discussed above align with practical applications? If incorporating additional experiments is currently infeasible, could the authors provide references demonstrating these phenomena are indeed observed in real-world applications?

* I also have a mild question regarding the claim in Lines 3-5: "While such games often involve non-convex non-concave objectives, empirical evidence shows that simple gradient methods frequently converge..." Could the authors provide references to specific empirical studies where gradient methods were observed to converge to equilibria in comparable experimental settings? To the best of my knowledge, unlike gradient methods in minimization problems, gradient descent-ascent is generally understood to have convergence limitations in real-world min-max games like GANs.


**I would consider increasing my evaluation score if the paper included a substantive discussion of how the theoretical phenomena discovered in this work manifest in practical applications.** This could be achieved through either empirical validation demonstrating these phenomena in realistic training scenarios, or provide evidence of existing literature showing analogous behaviors in applied min-max optimization.

**Ethical Concerns:**

["NO or VERY MINOR ethics concerns only"]

**Final Justification:**

This paper provides some interesting and novel theoretical results on the training dynamics of neural min-max games. However, there are no experiments to show whether these results are reflected in real-world scenarios, which makes the value of these theoretical results unclear. Thus, my final justification is borderline acceptable.

**Limitations:**

yes

**Paper Formatting Concerns:**

I believe this paper does not have major formatting issues.

**Quality:**

3

**Strengths And Weaknesses:**

## Strengths:

* The problem of solving neural min-max games is novel and interesting.
* The paper is well-written, particularly the introduction, which effectively motivates the general notation of neural min-max games.
* The results establish an intriguing connection between overparameterization in minimization problems and min-max games, offering valuable theoretical insights.

---
## Weakness:

* The current title suggests a broad investigation into how different architectures, initialization schemes, and training dynamics affect neural min-max game solving. However, the paper focuses narrowly on the convergence of alternating gradient descent ascent for two-layer neural networks. This discrepancy makes the title an inaccurate reflection of the paper’s actual contributions.


* The motivation of studying two-layer neural networks is not very clear. In single-objective optimization, two-layer neural networks are well-motivated by the universal approximation theorem. However, the paper does not adequately explain why this architecture is particularly meaningful or relevant in the context of min-max games. A stronger theoretical or empirical justification would improve the motivation for this choice.

* The paper lacks experimental results, making it unclear whether the theoretical findings align with practical scenarios.

* The proof of the main result reveals that convergence relies critically on initializing the two-layer neural networks with Gaussian distributions, which places the parameters near equilibrium from the outset (Lines 311–315). This may restrictive and fails to reflect real-world scenarios. Moreover, the absence of numerical experiments further undermines the practical relevance of the theoretical findings. Without empirical validation, it remains unclear whether the proven convergence properties hold under more realistic conditions or whether they are merely an artifact of the idealized assumptions.

---

> ### Author Rebuttal · Authors · 2025-07-31
>
> We sincerely thank the reviewer for their time and constructive feedback. We hope our response clarifies the contributions of the paper and addresses the concerns raised. We especially appreciate the reviewer's willingness to see the value in the theoretical study of overparameterization—even while raising thoughtful points about practical impact.
>
> ---
>
> ### Strengths
>
> > - The problem of solving neural min-max games is novel and interesting.
> > - The paper is well-written, particularly the introduction, which effectively motivates the general notion of neural min-max games.
> > - The results establish an intriguing connection between overparameterization in minimization problems and min-max games, offering valuable theoretical insights.
>
> We are grateful for the positive assessment and appreciation of our presentation.
>
> ---
>
> ### On Title Suggestion
>
> The reviewer suggested a more focused title. While we believe that both the abstract and introduction clearly frame the scope and avoid any overselling, we are happy to adopt a more targeted title such as:
>
> - *Solving Shallow Neural Min-Max Games: The Role of Overparameterization...*
> - *The Role of Overparameterization in 1-Hidden-Layer Zero-Sum Neural Games...*
>
> We note that such changes are typically accepted during the camera-ready/proceeding phase of the NeurIPS process.
>
> ---
>
> ### On the Use of Two-Layer Networks
> The motivation for using two-layer neural networks stems indeed from their **universal approximation property (UAP)**. Informally, UAP asserts that two-layer networks can approximate any continuous function on a compact domain to arbitrary precision—purely as a matter of representational power. This property is **independent of the optimization algorithm** used and applies regardless of the learning task.
>
> Thus, UAP holds in min-max games; two-layer networks can implement the Nash strategies of the hidden convex–concave game—*if* the right parameters can be found.
> This leads to our main question:
> > **How hard is it, from an optimization perspective, to find the parameters that realize such equilibrium strategies using a simple method like AltGDA?**
>
> While UAP ensures *existence*, our focus is on **computational tractability** and **algorithmic convergence**. We are not concerned with whether such a network *can represent* the solution (it can), but whether a natural learning dynamic—AltGDA—can **discover** it efficiently in practice.
>
> This is particularly non-trivial since AltGDA is adversarial in nature (unlike cooperative optimization schemes), and even in structured games with bilinear coupling, the dynamics can become unstable. Hence, even for such structured games, we require $\widetilde{\Omega}(n^3)$ overparameterization to guarantee convergence to the Nash equilibrium under AltGDA.
>
> In summary:
> - UAP ensures representational power—two-layer networks can implement Nash strategies.
> - Our contribution analyzes whether AltGDA can *find* those parameters, given large but practical overparameterization.
> - This bridges expressivity with optimization, offering the first steps toward understanding learning in neural min-max games.
>
> ---
>
> ## Main Concern: Practical Impact of Overparameterization in Min-Max Games
>
> > **I would consider increasing my evaluation score if the paper included a substantive discussion of how the theoretical phenomena discovered in this work manifest in practical applications.**
>
> We understand and value this concern. While our focus is theoretical (the primary area in our submission is THEORY), we aim below to provide both empirical context and intuitive takeaways from our results.
>
> ---
>
> ### Literature Showing Benefits of Larger Networks in Min-Max Settings
>
> In several applied min-max contexts—adversarial training, GANs, DRO, and neural agents—larger networks have been shown to lead to improved convergence and performance:
>
> - **Adversarial training:** Addepalli et al. [1] show improved robustness and performance with larger models.
> - **GANs:** Karras et al. [2], Brock et al. [3], and Sauer et al. [4] demonstrate that larger architectures stabilize training and enhance image quality.
> - **LLM Language agents:** Karten et al. [5] (PokéChamp) show improved performance using GPT-4.0 versus smaller LLMs.
> - **DRO:** Pham et al. [6] show that bigger models may yield better worst-group generalization.
>
> These results exemplify that while game dynamics can be reasonably unstable, if the model is enough large, this burden can be overpassed.
> Interestingly if the reader would like to sacrifice the technicalities around the non-asymptotic bounds our proof techniques offer several insights for practitioners:
>
> 1. **Interpretation of $\sigma_{\min}$ and Exploration**
>    The smallest singular value of the network Jacobian, $\sigma_{\min}$, controls how well the model explores the strategy space. When $\sigma_{\min} \approx 0$, certain strategies remain unexplored, indicating convergence to spurious subspaces. Our analysis ties this directly to the degree of overparameterization.
>
> 2. **Data Geometry and Regions of Attraction**
>    Our results show that overparameterized networks initialized with sufficiently diverse data are more likely to fall into regions where $\sigma_{\min} > 0$, ensuring stable convergence under AltGDA. While computing $\sigma_{\min}$ per iteration is impractical, the connection offers design insights for data and architecture.
>
> **These insights are, of course, intuitive and can stand on their own.** What makes our contribution notable is that our proof *relies critically* on these properties to establish formal guarantees.
>
> ---
>
> ### On Experiments
> Since our submission was under the `THEORY` track, we focused the main paper on foundational results. We have conducted supporting experiments to include in the final version. Per NeurIPS policy, we exclude them here but will include them in the final version.
>
> **In response to the reviewer's request**, we also note that similar experiments appear in prior work ([1']–[6'] below) either in shallow or deep NNs, which provide visual and empirical support for our framework.
>
> We hope this collective evidence serves as a strong preliminary justification of the practical relevance of our results.
>
> ---
>
> ### References (Literature on Overparameterization & Experiments)
>
> **Empirical Benefits of Bigger Models in Min-Max Learning**
> [1] Addepalli et al., *Scaling Adversarial Training to Large Perturbation Bounds*, ECCV 2022
>
> [2] Karras et al., *Progressive Growing of GANs for Improved Quality, Stability, and Variation*, ICLR 2018
>
> [3] Brock et al., *Large Scale GAN Training for High Fidelity Natural Image Synthesis*, ICLR 2019
>
> [4] Sauer et al., *StyleGAN-XL: Scaling StyleGAN to Large Diverse Datasets*, SIGGRAPH 2022
>
> [5] Karten et al., *PokéChamp: an Expert-level Minimax Language Agent*, 2025
>
> [6] Pham et al., *The Effect of Model Size on Worst-Group Generalization*, NeurIPS 2021
>
> **Successful Gradient-Based Training in Shallow or Large Neural Zero-Sum Games**
> [1'] Mladenovic et al., *Generalized Natural Gradient Flows in Hidden Convex–Concave Games and GANs*, ICLR 2022
>
> [2'] Vlatakis-Gkaragkounis et al., *Solving Min-Max Optimization with Hidden Structure via Gradient Descent Ascent*, NeurIPS 2021
>
> [3'] Sakos et al., *Exploiting Hidden Structures in Non-Convex Games for Convergence to Nash Equilibrium*, NeurIPS 2023
>
> [4'] Song et al., *NN-Based Synchronous Iteration Learning for Multi-Player Zero-Sum Games*, Neurocomputing 2017
>
> [5'] Lei et al., *SGD Learns One-Layer Networks in WGANs*, ICML 2020
>
> [6'] Wu et al., *Using CNNs for Solving Two-Player Zero-Sum Games*, Expert Systems with Applications 2022
>
> ---
>
> ### Random Initialization – How Restrictive Are Our Results?
>
> 1. **Network Optimization Min-Max Games**
> Our main result (Theorem 3.7) assumes a commonly used initialization scheme (e.g., He or LeCun), with the only additional requirement being that the variances satisfy Equation (9) in our paper.
> *Footnote:*
> He uses ($\mathcal{N}(0, \frac{2}{n_{\text{in}}})$) for ReLU; LeCun uses $(\mathcal{N}(0, \frac{1}{n_{\text{in}}}))$ for Sigmoid—both preserve variance and avoid vanishing gradients.
>
> Of course, bridging the gap between practice and theory, it remains an interesting open question whether the current polynomial overparameterization requirement can be reduced to linear. However, we note that—prior to our work—there were no existing theoretical results connecting initialization schemes with convergence guarantees in neural min-max games.
>
> 2. **Input Optimization Min-Max Games**
> In this setting, our results are even **less restrictive** regarding initialization. Specifically, we show that if the neural networks are randomly initialized from a standard Gaussian distribution, then **AltGDA can compute the min-max optimal inputs**. As discussed in the main text, this is an *average-case result*, in the spirit of smoothed analysis. While it is theoretically expected that there exist neural architectures encoding hard min-max landscapes—potentially close to PPAD-hard instances—in practice, our result suggests that a **randomly initialized neural min-max game is tractable under AltGDA**. In fact, from our proof, one can further deduce that the **mean of the Gaussian initialization can be arbitrary**, thanks to the convergence properties of AltGDA. Thus, this result can be interpreted through the lens of **smoothed analysis**:
> > Given a hard input neural min-max game, if one perturbs it with Gaussian noise (with inverse-polynomial variance), then AltGDA is provably able to find the optimal min-max input. The smaller the variance, the more brittle/rare are the hard instances for AltGDA.
>
> We hope these clarifications address your concerns and provide a broader view of the theoretical and practical relevance of our work. Thank you again for your thoughtful review and engagement.

---

> > ### Comment · Reviewer_njLz · 2025-08-02
> >
> > I would like to sincerely thank the authors for their thorough and thoughtful rebuttal. As a result, I have raised my score to 4. My decision reflects my appreciation for the paper’s interesting and novel theoretical contributions in the area of neural min-max games.
> >
> > I also recognize that several reviewers have expressed a shared concern that the absence of experimental results makes the practical relevance of the theoretical findings unclear, which is why I am unable to assign an even higher score. I highly encourage the authors to consider including experimental evaluations in the next version of the paper to further strengthen their work.

---

### Official Review · Reviewer_Xm4L · 2025-06-30

**Clarity:** 3
**Significance:** 2
**Originality:** 3
**Rating:** 4
**Confidence:** 2

**Summary:**

This paper investigates the training dynamics of two-player, zero-sum games where players' strategies are represented by two-layer neural networks. The work aims to explain the empirical success of gradient-based methods in these non-convex non-concave settings. The authors provide a theoretical framework based on "hidden convexity" and overparameterization to establish convergence guarantees for the AltGDA algorithm. The core of their analysis involves deriving a path-length bound for the algorithm's iterates and showing that a two-sided PL condition holds with high probability under sufficient overparameterization.

**Questions:**

- The assumptions O(n^3) widh requirement appear to distance the paper's findings from current deep learning practices. Could you provide a more compelling argument for the relevance of these results to practitioners?

- The paper focuses exclusively on two-player zero-sum games. Do you see a clear path to extending your framework to more complex settings, or are they fundamental issues?

**Ethical Concerns:**

["NO or VERY MINOR ethics concerns only"]

**Final Justification:**

- My initial criticism of the bound was likely too harsh without the context the authors have provided from the literature.
- They also provided insights from the theoretical analysis.
- As mentioned in the original review there is a gap between the theory and current practice but I have a more positive view of the paper's contribution.

**Limitations:**

The paper does acknowledge its limitations and potential societal impacts, which is good. However, the combination of requiring very wide two-layer networks and smooth activations, means the results are quite far from how neural networks are used in practice.

**Paper Formatting Concerns:**

no concern.

**Quality:**

2

**Strengths And Weaknesses:**

**Strengths**

- The paper is well-written. The authors do a good job of motivating the problem and outlining their approach.
- The contribuations of this work could have impact on modern machine learning, with relevance to GANs, adversarial robustness, and multi-agent systems.

**Weakness**

- The required network width of O(n^3) is extremely high.  This cubic scaling with the number of samples makes the result largely of theoretical interest.

- The fact that the paper’s guarantees hold only for two-layer networks is a major practical limitation and extending these results to deep networks remains a challenge for future research.

- The paper is purely theoretical, and there’s a big gap betwen its assumptions and how things actually work in practice. It really needs some experiments to show whether the overparmeterize networks are required, or to check if things break down when those conditions aren’t met. Without any empirical evidence, the results are abstract.

---

> ### Author Rebuttal · Authors · 2025-07-31
>
> We would like to sincerely thank the reviewer for engaging with our submission. Although the overall assessment was borderline-negative, we appreciate that you acknowledged both the relevance of the problem and the clarity of the draft.
>
> While one might note that this paper was **submitted to the theory track**—and thus expect more tolerance for foundational results—we recognize the importance of bridging theoretical insights with practical implications. We would like to highlight that, to the best of our knowledge, this work provides the **first overparameterization condition** for solving a broad class of **unconstrained min-max optimization problems** that arise in modern machine learning. Importantly, our result gives a *sufficient* condition, not a necessary one. Investigating whether the $\widetilde{\Omega}(n^3)$ bound is tight or improvable remains a compelling direction for future work.
>
> ### Comparison with Previous Work
>
> It’s worth recalling that the first overparameterization bounds for minimization problems were of the order $\Omega(n^8 L^{12})$ (see table at the end of our resposne). It is natural that the complexity of the problem increases in game settings (with more than one player), and we currently lack a complete understanding of how such complexity scales. Hence, we see our work as the initial stepping stone in this research journey. As for experiments, we do plan to include them in the final version, but also note that several empirical results already exist in related literature (please see our detailed response to Reviewer `F1FN`).
>
> That said, we understand that a practitioner might find it difficult to extract immediate actionable insights from the current theoretical guarantees alone. Therefore, beyond the asymptotic bounds and technical derivations, we outline below a few **practical takeaways** that our proof techniques reveal:
>
> ---
>
> ### Practical Insights for Practitioners
>
>
> 1. **Interpretation of $\sigma_{\min}$ and Exploration**:
>    The smallest singular value of the network's Jacobian, $\sigma_{\min}$, governs how well the model explores the strategy space. If $\sigma_{\min}$ is near zero, certain regions of the space may be entirely unexplored—indicating convergence to spurious subspaces. This provides a novel geometric interpretation of poor learning dynamics in games.
>
> 2. **Data Geometry and Regions of Attraction**:
>    The region where $\sigma_{\min} > 0$ is critical for guaranteeing convergence of AltGDA. Our results offer intuition into how *data selection and spread* affect optimization. While computing $\sigma_{\min}$ at each iteration for each candidate point is impractical, our findings imply that *overparameterization*, together with sufficiently diverse data, increases the likelihood of initialization inside a favorable region of attraction.
>
> ---
>
> These insights are, of course, intuitive and can stand on their own. What makes our contribution notable is that our proof *relies critically* on these properties to establish formal guarantees.
>
>
>
>
> ### Beyond Two-Player Zero-Sum Games
>
> > **Do you see a clear path to extending your framework to more complex settings, or are there fundamental issues?**
>
> This is an excellent and very insightful question. As is implicit throughout the paper, our results build upon several recent tools from the *hidden convex–concave* and *two-sided PŁ* literature, which we combined with insights from overparameterization theory to address general ML tasks within the two-player zero-sum setting.
>
> The natural next step is to understand how these techniques scale to **multi-player** and **non-zero-sum** settings—especially in structured environments like **polyhedral games**, which share connections with extensive-form games.
>
> Currently, we are working on the first steps towards this goal. Specifically, we are developing **gradient-based methods tailored for structured, non-monotone multiplayer games**. A key idea we are exploring is the analogy between *two-sided PŁ-conditions* (for two-player games) and **hypo-monotonicity** in multi-agent operator theory. This connection may allow us to transfer and generalize some of the intuition and techniques from our current setting.
>
> If this optimization foundation holds, we aim to investigate how the core components of our analysis—such as **gradient path bounds** and their link to **network width and depth**—can be extended to games with more than two players. While this direction lies beyond the scope of the current submission, we believe it opens up a rich and technically deep avenue for future work.
>
> Thank you again for raising this forward-looking and foundational question.
>
>
> Once again, thank you for your careful reading and constructive criticism. We hope this clarifies the motivation and broader impact of our work, and we welcome further feedback or suggestions.
>
> ---
> Below we provide a summary of some of the literature on overparameterization conditions derived for the case of minimizing $L_2$ loss function with shallow and deep neural networks (FCNN = Fully Connected Neural Network, $n$ = number of training samples, and $L$ = number of hidden layers).
>
> | Reference  | Model | Depth | Initialization | Activation | Width |
> |--------------------------------|-------------|----------|--------------------|-----------------|-------------------------|
> | Oymak and Soltanolkotabi (2020) | FCNN | Shallow | Standard Gaussian | ReLU | $\Omega(n^2)$ |
> | Zou and Gu (2019) | FCNN | Deep | He | ReLU | $\Omega(n^8 L^{12})$ |
> | Nguyen (2021) | FCNN | Deep | LeCun | ReLU | $\Omega(n^3)$ |
> | Song et al. (2021) | FCNN | Shallow | He/LeCun | Smooth | $\Omega(n^{3/2})$ |
> | Bombari et al. (2022) | FCNN | Deep | He/LeCun | Smooth | $\Omega(\sqrt{n})$ |                 |
>
> **References:**
>
> [1] Oymak and Soltanolkotabi (2020), Toward moderate overparameterization: Global convergence guarantees for training shallow neural networks
>
> [2] Zou and Gou (2019), An improved analysis of training over-parameterized deep neural networks.
>
> [3] Nguyen (2021), On the proof of global convergence of gradient descent for deep relu networks with linear widths
>
> [4] Song et al. (2021), Subquadratic overparameterization for shallow neural networks
>
> [5] Bombari et al. (2022), Memorization and optimization in deep neural networks with minimum overparameterization.

---

> > ### Comment · Area_Chair_iygP · 2025-08-04
> >
> > Dear reviewer,
> >
> > Thank you for your service!
> >
> > The authors have responded to your comments. Could you please take the following actions (if not already):
> >
> > 1- Press the “Mandatory Acknowledgement” button to confirm that your read the other reviews and response by the authors
> > 2- Reply to authors by saying if their response addressed your concerns. You are encouraged to engage in a discussion with the authors in case you disagree with aspects of their response to further clarify them.
> >
> > 3- You are also encouraged to provide your thoughts on the other reviews and response to them.
> >
> >
> > Thank you
> >
> > The AC

---

> > ### Comment · Reviewer_Xm4L · 2025-08-05
> >
> > Thank you for your detailed rebuttal. I appreciate you taking the time to address my concerns and provide additional context for your work.
> >
> > I acknowledge that my initial criticism of the bound was likely too harsh without the context you have provided from the literature. I also appreciate the practical insights you've drawn from your theoretical analysis. That said, a gap between the theory and current practice remains the primary weakness of the submission in its current form.
> > Given your clarifications, I have a more positive view of the paper's contribution. While the practical limitations exist, the theoretical groundwork is valuable. I will raise my score accordingly to reflect that the reasons to accept now outweigh the reasons to reject.

---

### Official Review · Reviewer_F1FN · 2025-07-01

**Clarity:** 3
**Significance:** 4
**Originality:** 4
**Rating:** 5
**Confidence:** 4

**Summary:**

The paper considers hidden convex/concave games parametrized by 2-player neural networks. More precisely, the authors consider a data set $D = (D_x, D_y ) =  ( x_i , y_i )_{i=1}^n$ and  the following min-max game

$sum_{i=1}^n = \ell( G_\theta(x_i,y_i ) , G_w(x_i,y_i )$
where $G_z,G_w$ are the 2-layer NNs.

The authors consider the case where the function $\ell(\cdot,\cdot)$ is convex/concave. In fact they consider a more special case where $\ell(\cdot,\cdot)$ is strongly convex/strongly while admiting an additional bilinear term. The authors establish that *Alternating Gradient Descent/Ascent* converges to a min-max equilibrium $(w^\star,z^\star)$ in polynomial number of steps.

On the technical level the authors leverage a recent result of Yang et al. establishing convergence results for Alternating GDA in zero-sum game satisfying the *two-sided PL condition*. In a nutshell, the authors show that with a suitable initialization of the  weights the two-sided PL condition will be satisfied. Following techniques from the Neural Tangent literature, the authors then establish that the trajectory of Alt GDA remain sufficiently close to the initial weights. In this way they are able to establish the fact that two sided PL condition is always satisfied and derive a final convergence result.

**Questions:**

1. In line 194 you write $I_{1}^{D_F}(F_\theta):= \sum_{i \in [|D_F|]} \ell_{i} (y_i , F_\theta(y_i)) = \sum_{j \in [|D_G|]} \ell_{j} (y_j , F_\theta(y_j))$. Do you mean $\ell(x_i,y_i)$? is it a typo? Otherwise I do not understand why the loss term changes with the example.

2. Also find a bit unintuitive the fact that the data set $D = (D_F,D_G)$ is separated between the two players. Could you provide some more details or an illustrative example on the coupling matrix $A(x_i,x_j,y_i,y_j)$?

3. What if the function $I_2$ is convex/concave instead of bilinear?

4. Do your results transfer in case of noisy gradients?

5. Have you experimentally evaluated your claims in simple hidden convex/concave games e.g. Rock-Paper-Scissors parametrized by two DNNs?

6. I think there is small typo in 304 ($P_0$ two times).

**Ethical Concerns:**

["NO or VERY MINOR ethics concerns only"]

**Final Justification:**

After the author-reviewer discussion i am very confident on the significance of the paper's results and thus i recommend acceptance.

**Limitations:**

Yes

**Quality:**

4

**Strengths And Weaknesses:**

**Strengths**

I think that this a very interesting paper considering a very interesting and important setting. Identifying classes of non-convex/non-concave games that circumvent the intractability results of Daskalakis et al. but at the same time remain relevant to real-world setting is a very important research direction. The paper provides solid convergence results in the class of hidden convex/concave games which I believe is a very interesting first step towards the aforementioned direction. The authors have made an excellent job in providing specific example in Appendix B motivating the class of hidden convex/concave markov games.

At the technical level, the authors transfer the techniques coupled the techniques devellopped in the overparametrization literature with recent results concerning min-max optimization to establish their convergence results. The basic challenge consisted of establishing that along the trajectory of Alt GDA the two-sided PL conditions are satisfied. The paper seems to admit deep technical depth. That being said, I did not check the correctness of the proofs in the appendix.

**Weaknesses**

One of weakness of the papers concerns the presentation and the verification of the theoretical claims. I want to remark that this is to an extend justified due to 9 page limit and the technical nature of the paper. However I really believe that more discussion would be helpful concerning the various set of assumptions presented along the paper. Another probably weak point concerns the fact that the authors consider a bilinear + strongly convex/concave setting limiting the applicability of the results. Finally no experimental evaluations are presented. I believe that the providing some experimental verification of the theoretical claims even in toy examples (e..g RPS played by two NNs) would add value to the paper.

---

> ### Author Rebuttal · Authors · 2025-07-31
>
> We would like to sincerely thank the reviewer for their thoughtful and encouraging comments regarding our research direction, and for the time and care dedicated to reading even the appendix sections. Below we address the points raised.
>
> ### On Experiments
>
> Since the focus of our work is on building the theoretical foundations of training dynamics in hidden convex–concave games, we chose not to burden the 9-page main paper with additional experimental sections. That said, we appreciate the reviewer's interest, and we have conducted new experiments to support our claims. Unfortunately, due to the recent note discouraging inclusion of new content in the rebuttal, we are unable to include them here. We will be excited to incorporate them in the final version, with illustrative ML- and game-friendly examples (e.g., neural networks playing RPS-style games).
>
> We also note that related works in the hidden convex–concave literature and NNs in Zero-sum games (see, e.g., [1–6]) already contain visualizations of this kind, which we view as further support of the framework's relevance.
>
> ### On Assumptions and Applicability
>
>
> Extending our answer to Reviewer `tXT3` (score: 5 - accept), we agree that a more detailed discussion of the assumptions will improve clarity. The table below contrasts our setup with what is typically observed in practice:
>
> |                          | **Our Paper**                                           | **In Practice**                                                                         |
> |--------------------------|---------------------------------------------------------|------------------------------------------------------------------------------------------|
> | **Type of Neural Network** | 1-hidden-layer, fully-connected, smooth activations     | Typically deep networks; not necessarily fully-connected (e.g., ResNets, CNNs)           |
> | **Training Algorithm**     | AltGDA                                                  | Mainly double-loop methods (e.g., approximate best-response oracles); not always AltGDA  |
> | **Initialization**         | Gaussian with variance constraints                      | Similar heuristics (e.g., He, Xavier, or LeCun initializations)                          |
>
> Among these, the assumption on AltGDA is arguably the most benign. Stabilization is essential in nonconvex–nonconcave min-max settings. While double-loop methods are often used in practice, AltGDA provides a simpler, parallelizable single-loop alternative.
>
> Similarly, Gaussian initialization aligns well with common initialization schemes. The main divergence lies in the network architecture: practical models are often very deep with fixed-width layers. Understanding their theoretical behavior remains an open and fascinating question. Recent works have begun to study overparameterization in deep networks for minimization tasks, but our focus here is on a more tractable regime—avoiding the NTK limit—to deliver a non-asymptotic analysis of 1-hidden-layer networks in game-theoretic settings. Relaxing these assumptions remains a rich direction for future work.
>
> Going beyond smooth activations is also possible, though technically more demanding. Extending the analysis to non-smooth activations (e.g., using Clarke subdifferentials or related tools—see [8]) is indeed a compelling path forward.
>
> ---
>
> On the other hand, some properties in our setup are not assumptions but **structural features of the model**. Much of the existing overparameterization literature focuses on $\ell_2$ regression. In contrast, we chose to target a broader family of problems, motivating structural conditions like smoothness, gradient growth, and Lipschitz continuity not as limiting assumptions but as general-enough properties that support a wide class of objectives.
>
> Regarding the **separable min-max setting**, we note that it is the dominant structure in most Euclidean game formulations. While one could generalize to a setting where the $I_1$ and $I_3$ components follow a non-quadratic drift, this would introduce significantly more complexity. We agree that this is a valuable and interesting theoretical direction, and we would be happy to include it among the open problems in the final version.
>
> Lastly, we clarify that the **composition of a strongly convex function with a smooth well-conditioned map** (e.g., KŁ or PŁ structure) is not an assumption but rather a mathematical fact—what we call *hidden convexity*. This underlies many of the observed phenomena in both minimization and min-max settings.
>
> ---
>
> We will revise the main text to better motivate our structural choices and position them within the broader context of min-max optimization and overparameterization literature.
>
> Although our current setting focuses on bilinear + strongly convex–concave games, we view it as a meaningful and foundational step. Our broader aim is to help identify tractable nonconvex–nonconcave games that remain algorithmically relevant in practical machine learning, and we believe our framework contributes in this direction.
>
> Once again, thank you for your engagement and generous feedback. We truly appreciate your support and look forward to further developing this line of research.
>
> ### Additional questions about the manuscript
>
> > 1. In line 194 you write $I_{1}^{D_{F}}(F_{\theta}) := \sum_{i\in[|D_{F}|]} \ell_{i}\bigl(y_{i},F_{\theta}(y_{i})\bigr) = \sum_{j\in[|D_{G}|]} \ell_{j}\bigl(y_{j},F_{\theta}(y_{j})\bigr)$. Do you mean $\ell(x_{i},y_{i})$? Is it a typo? Otherwise I do not understand why the loss term changes with the example.
>
> In Line 194 we wrote the min-max objective in more generality. However, for our analysis we specifically use losses $\ell$ and $\ell'$ for the MIN and MAX players, respectively. In practice as well (as highlighted with examples in Appendix B), we see this being the case. If one were to use different losses for every example (for both the players), one would have to instead work with the minimum PL-modulus of these losses in our analysis.
>
> > 2. I also find it a bit unintuitive that the data set $D=(D_{F},D_{G})$ is split between the two players. Could you provide more details or an illustrative example of the coupling matrix $A(x_{i},x_{j},y_{i},y_{j})$?
>
> Regarding the splitting of the data, we have stated our result assuming an equal split for ease of presentation. However, no step in our analysis is limited or constrained by the fact that the datasets have been split.
>
> For an example of the coupling matrix, consider the example B.7 in Appendix B for input-games. Here, the coupling matrix $A$ is independent of data: $A(x_i,x_j,y_i,y_j) = \delta_{ij}$.
>
> > 3. What happens if the function $I_{2}$ is convex/concave instead of bilinear?
>
> If $I_2$ is convex-concave such that it satisfies the gradient growth bound condition (Assumption 2.1 (iv)) just like how we assumed so for $I_1$ and $I_3$, our results would carry over.
>
> > 4. Do your results still hold in the presence of noisy gradients?
>
> For literature in the *minimization* setting, there are known overparameterization bounds in the presence of noisy gradients (see, e.g., [7, 8]). So, we would speculate that our results which are for the *min-max* case in the presence of `full-feedback` could be extended to the case of `noisy gradients`. We leave this as an interesting direction for future work.
>
> > 6. I think there is a small typo in line 304 ($P_{0}$ appears twice).
>
> Thank you for pointing out the typo.
>
> **References:**
>
> [1] Mladenovic et al., Generalized natural gradient flows in hidden convex-concave games and GANs, ICLR 2022
>
> [2] Vlatakis-Gkaragkounis et al., Solving min-max optimization with hidden structure via gradient descent ascent, NeurIPS 2021
>
> [3] Sakos, et al., Exploiting hidden structures in non-convex games for convergence to Nash equilibrium, NeurIPS 2023
>
> [4] Song et al., Neural-network-based synchronous iteration learning method for multi-player zero-sum games, Neurocomputing 2017
>
> [5] Lei et al., SGD Learns One-Layer Networks in WGANs, ICML 2020
>
> [6] Wu et al., Using CNN for solving two-player zero-sum games, Expert Systems with Applications 2022
>
> [7] Song et al., Subquadratic Overparameterization for Shallow NNs, NeurIPS 2021.
>
> [8] Zou & Gu, Improved Analysis of Overparameterized DNNs, NeurIPS 2019.

---

> > ### Comment · Reviewer_F1FN · 2025-08-02
> > **Reviewwr F1FN response**
> >
> > Thank you very much for detailed response. I keep my positive impression on the paper and I maintain my score.

---

### Official Review · Reviewer_tXT3 · 2025-07-02

**Clarity:** 3
**Significance:** 2
**Originality:** 4
**Rating:** 5
**Confidence:** 3

**Summary:**

The authors consider models for zero-sum games between neural networks, characterizing scenarios such as adversarial training, alignment and robust optimization, with an equilibrium formulation of the stable desirable system behavior. Generally, when the game objective are non-convex non-concave  there is no guarantee that simple gradient methods would converge to the equilibrium, however in practice there have been many cases where such a convergence does occur. This led the researchers to speculate that there is some underlying structure in the system, leading them to build a theoretical framework for analyzing this, considering the system initialization, the dynamics and “hidden convexity”.

**Questions:**

You make several assumptions regarding the neural models and training regime. Could you summerize these is a short form, and discuss which of these are similar to the practical use-case and which of these actually different?

**Ethical Concerns:**

["NO or VERY MINOR ethics concerns only"]

**Final Justification:**

Good paper, I stand by my rating,

**Limitations:**

The work is theoretical in nature (based on previous empirical evidence).

**Paper Formatting Concerns:**

-

**Quality:**

3

**Strengths And Weaknesses:**

The key result in the paper is a characterization of properties that lead to convergence to an equilibrium in a class of such games (with 2 layer neural networks). The characterization is based on path length bounds for alternating gradient descent/ascent.

Overall, I find the topic of the paper very interesting. The overall structure of competing neural nets has received a huge amount of attention in the ML literature, leveraging foundations and inspirations from game theory. However, the theoretical foundations lagged behind the successful empirical results. In short, game theorists were surprised with the success of simple alternative methods, that they expected to simply orbit rather than converge. Overall I think the class of games tackled here require making non-trivial assumptions on the neural nets and training procedure that are still perhaps far from the actual networks used in practice. Hence, this results narrow that gap between theory and practice, but does not eliminate it. Still, a very interesting result overall.

Couple of comments:

Can you add a table of know results regarding convergence properties from the game theory (I know here you tackle 2 player zero sum games, but what is known about broader classes? Also perhaps discuss polynomial time computation of equilibria and why it could not be used here, and processes such as fictitious play). This would better place the paper in the context of background work.

I think a better exposition of PL is warranted; you do address it, but I think this needs to be expanded.

I think a table with examples (and citations) for games where empirical convergence is observed would make for a better presentation of the empirical inspiration for this work. Also, maybe you could have a discussion of cases where such convergence does not occur

Overall, good paper - my suggestions are mostly around presentation.

---

> ### Author Rebuttal · Authors · 2025-07-31
>
> We sincerely thank you for your enthusiastic review and thoughtful presentation-related comments.
>
> While we already provide an extended discussion in the related work section, we agree that a more structured comparison would be helpful. In the final version, we plan to include a summary table covering works on 2-player min-max problems under various structural assumptions—concave, monotone, quasi-strongly monotone, weak Minty, hypomonotone, and cocoercive—under both full and bandit feedback. This will better position our contributions relative to the literature, especially in connection to AltGDA and the two-sided PŁ-condition.
>
> We also plan to clarify our assumption regarding polynomial-time computation. Indeed, one might suggest solving the latent convex–concave game and then inverting the network to recover parameters approximating the optimal strategy. However, this poses two key challenges:
>
> 1. **Inversion of neural networks is NP-complete in the worst case**, making it impractical to expect practitioners to solve the latent convex–concave game and then invert the network to recover the corresponding parameters. While overparameterization may theoretically improve tractability and mitigate the brittleness of the inversion step, our focus is to understand and justify the empirical success of training dynamics (such as AltGDA) in structured nonconvex–nonconcave settings—without requiring explicit solution of the latent game.
>
> 2. **Games with exponentially large action spaces** make exact solution intractable. In these cases, neural networks (or other function approximators) offer a succinct and practical way to learn high-quality strategies.
>
> This modeling perspective strengthens both the theoretical and practical relevance of our framework.
>
> Additionally, in the appendix, we cite empirical studies showing the effectiveness of gradient methods on large neural networks—many of which fall under our model. In the final version, we also aim to include visualizations from our own experiments. For interested reviewers, similar experiments appear in prior works [1–6].
>
> Finally, we acknowledge the theoretical intractability of computing Nash equilibria and the PPAD-hardness even under smoothed complexity. Naturally, examples exist where GDA-based methods fail—several such failure modes in nonconvex–nonconcave min-max settings have been documented in [7–9]. On the other hand, the complexity of unconstrained min-max optimization remains an open question in complexity theory—specifically, whether it is computationally equivalent to solving NASH (PPAD), KKT (CLS), or lies somewhere in between these complexity classes. Clarifying this would yield a more universal understanding of the power and limitations of oracle-based methods like GDA.
>
> **References:**
>
> [1] Mladenovic et al., Generalized natural gradient flows in hidden convex-concave games and GANs, ICLR 2022
>
> [2] Vlatakis-Gkaragkounis et al., Solving min-max optimization with hidden structure via gradient descent ascent, NeurIPS 2021
>
> [3] Sakos, et al., Exploiting hidden structures in non-convex games for convergence to Nash equilibrium, NeurIPS 2023
>
> [4] Song et al., Neural-network-based synchronous iteration learning method for multi-player zero-sum games, Neurocomputing 2017
>
> [5] Lei et al., SGD Learns One-Layer Networks in WGANs, ICML 2020
>
> [6] Wu et al., Using CNN for solving two-player zero-sum games, Expert Systems with Applications 2022
>
> [7] Kalogiannis et al., Towards convergence to Nash equilibria in two-team zero-sum games, ICLR 2023
>
> [8] Jin et al., What is Local Optimality in Nonconvex-Nonconcave Minimax Optimization?, ICML 2020
>
> [9] Yang et al., Global convergence and variance reduction for a class of nonconvex-nonconcave minimax problems, NeurIPS 2020
>
> We greatly appreciate your support for our paper and would be happy to address any further questions or clarifications you may have.
>
> ### Questions:
> > You make several assumptions regarding the neural models and training regime. Could you summarize these in a short form, and discuss which of these are similar to the practical use-case and which of these actually different?
>
> In the table below, we summarize the assumptions regarding the neural networks and training regime:
>
>
> |  | Our paper | In practice |
> | -------- | -------- | -------- |
> | Type of Neural Network | 1-hidden-layer, fully-connected | Typically deep networks, not necessarily fully-connected (e.g. Residual or convolutional layers) |
> | Training Algorithm | AltGDA | Not necessarily AltGDA / Mainly Double-loop |
> | Network Initialization | Gaussian (with variance constraints) | Similar (e.g. He, Xavier, or LeCun initializations)
>
> Among these, the assumption on AltGDA is arguably the most benign. In non-convex/non-concave min-max optimization, stabilization is essential. In practice, double-loop methods (e.g., approximate best-response oracles) are often used for safety, while AltGDA serves as a more parallelizable and simpler single-loop alternative. Similarly, the Gaussian initialization is closely aligned with practical heuristics like He or Xavier. The main gap lies in the architecture: practical models are often very deep with fixed-width layers. Theoretical understanding of why such architectures succeed remains an super interesting open question. While recent work has begun to explore overparameterization in deep networks for minimization tasks, our paper focuses on a more analytically tractable setting—explicitly avoiding the NTK regime to provide a non-asymptotic analysis for 1-hidden-layer networks in a game-theoretic context. We view relaxing and extending these assumptions as a promising direction for future work.

---

> > ### Comment · Reviewer_tXT3 · 2025-08-05
> > **Thank you for the additional details**
> >
> > Thanks for the extra details re complexity classes.

---

### Official Review · Reviewer_Pjht · 2025-07-06

**Clarity:** 2
**Significance:** 4
**Originality:** 3
**Rating:** 5
**Confidence:** 2

**Summary:**

This paper studies the convergence of Alternating Gradient Descent-Ascent algorithm to NEs in min-max games, while both players' strategies are shallow and wide neural networks with randomly initialized network weights.

**Questions:**

1. In line 190, how to interpret $(\mathcal{D} _F, \mathcal{D} _G) = \{(x _i,y _i)\} _{i=1}^n$? My understanding is $\mathcal{D} _F = \{x _i\} _{i=1}^n$ and $\mathcal{D} _G = \{y _i\} _{i=1}^n$. Is my understanding correct?
(Side note: If my understanding is correct, I suggest the authors simplify the notations used in many expressions, e.g., $i\in [|\mathcal{D} _F|]\Rightarrow i\in[n]$, $I _1^{\mathcal{D} _F} \Rightarrow I _1$ since $\mathcal{D} _F$ is a fixed constant in this paper. The current notations are bit complex and redundant.)
2. In line 192, what is $\mathcal{S} _F, \mathcal{S} _G$? In line 196, it suggests that $\mathcal{S} _F, \mathcal{S} _G$ are the (finite) sets of pure strategy of some underlying game, but it should then have that the image of $F _\theta$ is $\Delta(\mathcal{S} _F)$. On other hand, if $\mathcal{S} _F, \mathcal{S} _G$ are the continuous strategy sets (also corresponding to a subset of Euclidean space of some dimension), then how to interpret $\mathbb{R}^{|\mathcal{S} _F|\times|\mathcal{S} _G|}$ in line 196?
3. In line 194, what is the domain of the loss function $l(y,h)$?
4. Assumption 2.1 (i) in line 203 makes me confused. In the expression $l(y, Map _w(x))$, it seems that $Map _w$ is a neural network with weight $w=(W _1, W _2)$ in later analysis in line 219-220. Why does it make sense to directly assume the behavior of function $l$ over network weight $w$? I think a better choice is to assume the upper bound of $W _1, W _2$ and the smoothness of $l(y,h)$ on $h$ separately. The smoothness of $l$ on $w$ might then follows directly.
5. In Assumption 2.1 (iv) in line 210. It seems that $\mathcal{Y}$ (consequently $\mathrm{diam}(\mathcal{Y})$) is fixed, so $A _2 \mathrm{diam}(\mathcal{Y}) + A _3$ can be treated together as a constant. Is my understanding correct?
6. In line 212, it is unclear that $\mathcal{L} _{\mathcal{D}}$ is convex-concave at which sense. It seems that $\mathcal{L} _{\mathcal{D}}(F _\theta, G _\phi)$ is convex in $F _\theta$ by assumption (iii), but it is highly unlikely that $\mathcal{L} _{\mathcal{D}}(F _\theta, G _\phi)$ is convex on $\theta$, since $\theta$ is network weights and there is no condition guaranteeing the convexity of loss function $l$ over network weight $\theta$.
7. Similar to question 4, in line 223. Why does it make sense to assume the smoothness condition of $h(x)$ over network parameters $W _1^{(h)}, W _2^{(h)}$?
8. What is the Hermite norm defined in line 225?
9. What is the meaning of $(W _2^{(h)}) _k$ in line 227?
10. In Fact 2.6 in line 243, why does loss function $\mathcal{L} _{\mathcal{D}}$ satisfy PL condition? This statement is unclear to me, with same reason described as question 6.
11. In line 257, note that $\min _\theta \mathcal{L}(\theta, \phi)$ depends soly on $\phi$, so the expression $\theta \mapsto \min _\theta \mathcal{L}(\theta, \phi)$ does not make sense.
12. In line 291-300, Lemma 3.1 makes sense only when $c < 1$, yet the authors give no further clarifications on the range of $c$.
13. In line 293, what is the expression of $P _0$?

**Ethical Concerns:**

["NO or VERY MINOR ethics concerns only"]

**Final Justification:**

This paper provides theoretical understanding of the hidden convex-concave games of neural networks, showing that with over-parameterization, proper initialization, and a gradient-ascent-descent algorithm, the NE can be approximated within polynomial complexity.
The results provide a theoretical foundation for algorithms that aim at finding an NE of two networks with opposite loss functions. The techniques are solid as far as I can tell. Based on the above points, I recommend an accept.

**Limitations:**

yes

**Quality:**

3

**Strengths And Weaknesses:**

**Strengths**:

The problem is well-motivated and interesting. The technical results are solid. (I have not checked the correctness of proofs in appendix, since I'm not familiar with these theories.)

**Weaknesses**:

I do not find any weaknesses regarding the theoretical results of this paper.
Actually, I feel confused about many notations and assumptions made in this paper, consequently hard to evaluate the value of this paper. These concerns are listed at **Questions** section below. I would consider to increase my evaluation if these concerns could be addressed.

---

> ### Author Rebuttal · Authors · 2025-07-31
>
> First of all, we sincerely thank the reviewer for the helpful comments and questions, which we believe will contribute to a more favorable and clearer evaluation of our work.
>
> ### Questions:
>
> > **Q1.** In line 190, how should we interpret $(D_F, D_G) = ((x_i, y_i))_{i=1}^n$? My understanding is that D_F = (x_i) for i = 1 to n and D_G = (y_i) for i = 1 to n. Is this correct?
> > *Side note:* If so, I suggest simplifying several notations, e.g., i in [size of D_F] becomes simply i in [n], and I_1 superscript D_F becomes I_1, since D_F is fixed throughout the paper. The current notation feels a bit complex and redundant.
>
>
> Thank you for pointing this out—this is an important clarification. Unlike in classical game dynamics (e.g., min-max games like $x^T A y$ where x and y are strategies), here $D_F$ and $D_G$ represent datasets.
> The notation $(D_F, D_G) = ((x_i, y_i)$ for $i = 1$ to $n$) follows machine learning conventions, where each $(x_i, y_i)$ is a data-label pair. Specifically, we partition the dataset into two parts:
> $D_F = ((x_j, y_j)$ for $j=1$ to $n_1)$ and $D_G = ((x_k, y_k)$ for $k=1$ to $n_2)$ with $n_1 + n_2 = n$. Each player receives their own set of training pairs.
>
>
> > **Q2.** About strategy-space notation (lines 192 & 196)
>
> Thank you for pointing out this typo. Indeed, $S_F$ and $S_G$ refer to general strategy sets, which are not necessarily finite—they may be continuous subsets of Euclidean space. Therefore, the expression in Line 196 should be interpreted more generally as a function defined over $S_F \times S_G$, or $\mathbb{R}^{\text{dim}(S_F) \times \text{dim}(S_G)}$. We acknowledge this mismatch and revise the notation accordingly to avoid confusion.
>
> > **Q3.** The domain of the loss function $l(y,h)$?
>
> According to our adopted notation, the domain of the loss function $l(\cdot, \cdot)$ when using a neural network $h$ for predictions is $\mathbb{R} \times \mathbb{R}^{d_2}$.
>
> > **Q4 & Q7.** Why assume smoothness of $l(y, \text{Map}_w(x))$ and $h(x)$ with respect to weights $w = (W_1, W_2)$ directly?
>
> We agree that one could instead assume smoothness in the network output $h = \text{Map}_w(x)$ and separately constrain $W_1, W_2$. However, we adopt the unconstrained formulation to align with the standard setup in overparameterized optimization and to keep the analysis consistent with AltGDA, which is designed for unconstrained min-max problems. Adapting techniques from constrained settings (e.g., [1]) using proximal PL is a promising future direction.
>
> We emphasize that $h_\theta(x)$ is a smooth operator (e.g., a neural network). Consistent with the overparameterization literature (e.g., [1], [2]), we assume differentiable activation functions. Smoothness helps define a trust region around initialization where the Jacobian of $h_\theta$ remains non-singular. This enables us to define a meaningful radius $R = \mu_{\text{Jac}} / (2 \beta_h)$ where optimization dynamics behave well. ReLU networks are non-smooth only on a measure-zero set. Extending to non-smooth activations using tools like Clarke subgradients (e.g., [3]) is a valuable direction.
>
> > **Q5, Q8, Q9, Q11–Q13.**
>
> **Q5.** Yes, your understanding is correct. We stated it this way for completeness.
>
> **Q8.** Using Gaussian random initialization, the Hermite norm helps bound the singular values of $h$'s Jacobian. For $\phi: \mathbb{R} \to \mathbb{R}$, the Hermite norm is
> $ ||\phi|| = \sqrt{ \sum_{i=0}^\infty c_i^2 }$, where
> $c_i = \langle \phi, q_i \rangle = \frac{1}{\sqrt{2\pi}} \int \phi(x) q_i(x) \exp(-x^2/2) dx$, and $q_i$ is the $i$-th Hermite polynomial.
>
> **Q9.** It denotes $W_2$ at iteration $k$ of AltGDA. That is, if $\theta$ is optimized, $\theta_k$ denotes the parameters at step $k$.
>
> **Q11.** We revised to: *$\min_\theta L(\theta, \phi)$ has a non-empty solution set and a finite optimal value (and analogously for fixed $\theta$).*
>
> **Q12.** We elaborate the expression and valid range for $c \in (0,1)$ in Remark D.5. We will move this to the main text for clarity.
>
> **Q13.** We use the same Lyapunov potential $P_t$ as Yang et al. (2020), but now state it clearly:
>
> $P_t := \left( \max_{\phi} \mathcal{L}(\theta_t, \phi) - \mathcal{L}(\theta^\star, \phi^\star) \right) + \lambda \left( \max_{\phi} \mathcal{L}(\theta, \phi_t) - \mathcal{L}(\theta_t, \phi_t) \right)$
>
> with $\lambda = 1/10$, $t = 0$. The choice of $\lambda$ does not affect our conclusion that $\tilde{\Omega}(n^3)$ overparameterization suffices.
>
>
>
> > **Q6 & Q10.** Hidden Neural Games and PL-condition
>
> We agree that $L_D$ is **not** convex–concave in weights $\theta$ and $\phi$. Rather, it is strongly convex–concave in the outputs of $F_\theta$ and $G_\phi$, i.e., **hidden convex–concave**. This is central to our framework: the objective is nonconvex–nonconcave in parameter space but becomes tractable when viewed compositionally. Our analysis exploits this hidden structure.
>
> We will clarify Fact 2.6:
> By Proposition 2 of Fatkhullin et al. [4], if $f(\theta) = F(H(\theta))$ and $F$ is strongly convex while $H$ is a smooth map (e.g., a neural net), then $f$ satisfies the Polyak–Łojasiewicz condition. Thus, hidden strong convexity implies PL, even for nonconvex objectives. Our contribution is to make this connection explicit in the min-max setting.
>
> Moreover, Proposition 4.1 of D’Orazio et al. [5] gives a precise PL constant depending on the smallest singular value of $H$'s Jacobian. If $H$ is close to an invertible, well-conditioned map, the PL modulus is preserved.
>
> Hence, even though $L_D$ is nonconvex in parameters, it satisfies the PL condition due to its compositional nature—an insight at the heart of our analysis.
>
>
> ---
>
> **References:**
>
> [1] Kalogiannis et al., *Solving Zero-Sum Convex Markov Games*, ICML 2025
>
> [2] Song et al., *Subquadratic Overparameterization for Shallow NNs*, NeurIPS 2021
>
> [3] Zou & Gu, *Improved Analysis of Overparameterized DNNs*, NeurIPS 2019
>
> [4] Fatkhullin et al., *Stochastic Optimization under Hidden Convexity*, arXiv:2401.00108
>
> [5] D’Orazio et al., *Hidden Monotone VIs via Surrogate Losses*, ICLR 2025
>
> ---
>
> We thank you for your thoughtful feedback. We hope our clarifications improve the accessibility of the paper.
>
> Having clarified our notation, here we repeat our central message: when two neural networks interact via simple game dynamics, their weights can be trained to find the Nash equilibrium. Our analysis identifies both the scale and strategy of initialization required. To our knowledge, this is the first work to connect overparameterization to min-max optimization in hidden or non-convex domains.
>
> We also chose our abstraction and notation to move beyond standard $\ell_2$-based objectives, aiming to support generalization to broader loss landscapes. We hope this clarifies our intentions and supports a higher evaluation.

---

> ### Comment · Reviewer_Pjht · 2025-08-03
> **Response to Authors**
>
> Thank you for your response.
>
> I appreciate the authors' detailed response very much. After a revisit of this paper and the reference [4][5] with the help of authors' response, I think that I've understood the proof sketch of the results. I believe the technical calculations in appendix are correct (though I've not carefully checked them). As a result, I'm happy to increase my score to 5 (Accept). My re-evaluation is mainly based on technical solidity and valuable implications of this paper's results (solving hidden convex-concave games of neural networks with guaranteed convergence and polynomial complexity).
>
> One remaining weakness is the paper's presentation.
> The current presentation uses many conventional notations particularly from learning theory,  as well as many results of recent works without restating them in the main body. These issues hinder researchers like me who are not familiar with deep learning theory from understanding the technical insights and results.
> However, I do not mean to underrate this paper for this weakness. Since the technical details of this paper seem substantial and the page requirement is limited, I believe the authors have tried to lay out the technical parts in a relatively reasonable way. I hope that the authors could further improve the clarity in the next version, possibly by utilizing the additional page.

---

### Note · Authors · 2025-08-16

We sincerely thank all the reviewers and area chair for deeply engaging with our submission. We are quite happy to see that the reviewers were positive about our proposed framework of neural min-max games and our contributions therein. We take this opportunity to provide a summary of our planned revisions in order to address the reviewers' feedback.

## 1. Regarding presentation and exposition (addressing `Pjht`, `tXT3`, `F1FN`)

The discussion brought to light some typos and a need for some clarifications/remarks regarding:
1. The link between hidden convexity and PŁ-condition.
2. Assumptions on the the neural networks and training regime in our work compared to what's used in practice.
3. Additional summaries of works on min-max problems under various structural assumptions (concave, monotone, quasi-strongly monotone, etc.) under both full and bandit feedback.

We will make these changes to ensure readability and completeness.

## 2. Regarding practical relevance of our theoretical findings and Future Directions (addressing `F1FN`, `Xm4L`, `njLz`)

Another concern shared by some of the reviewers pertains to the experimental evaluations and practical insights for practitioners based on our theoretical results on overparameterization. While the paper was submitted to `THEORY` track, we understand and recognize the importance of bridging theoretical insights with practical implications. Therefore, we will make changes in our submission to provide:
1. A discussion of existing literature exhibiting benefits of larger neural networks in various min-max settings (adversarial training, GANs, DRO, and neural agents) and successful gradient-based training in shallow or large neural zero-sum games.
2. Supporting experiments such as Rock-Papers-Scissors game played between two neural networks.
3. A discussion around interplay between data geometry, non-singularity of neural network Jacobian and exploration of strategy space.

In addition to points mentioned above, we will also revise our submission by adding a discussion about extending our analysis for networks with non-differentiable activation functions (e.g. ReLU), multi-agent settings, and non-zero-sum games.

---
We are hopeful that our revisions will address all reviewers' concerns and feedback resulting in a stronger submission with an improved narrative. We thank the reviewers once again for taking out their valuable time for engaging with our submission and helping us improve the quality of our work.

---

### Decision · Program_Chairs · 2025-09-17

**Decision:**

Accept (spotlight)

**Comment:**

The paper studies the convergence of zero-sum games between neural networks to equilibrium by exploiting a hidden convexity. This is a quite elegant result, knowing that the study of such, a priori,  non-convex non-concave problems, is hard that has many practical implications in machine learning, since it covers adversarial training, alignment and robust optimization.


I agree with the reviewers that this is a strong contribution that is a strong fit for NeurIPS, the authors are encouraged to include the suggestions made by the reviewers to the revised version.